# Reviews and syntheses: Parameter identification in marine planktonic ecosystem modelling

**Schartau Markus[1], Wallhead Philip[2], Hemmings John[3,4], Löptien Ulrike[1], Kriest Iris[1], Krishna Shubham[1], Ward Ben A.[5], Slawig Thomas[6], and Oschlies Andreas[1]**

[1]GEOMAR Helmholtz Centre for Ocean Research Kiel, Germany
[2]NIVA, Norwegian Institute for Water Research, Bergen, Norway
[3]Wessex Environmental Associates, Salisbury, United Kingdom
[4]now at Met Office, Exeter, United Kingdom
[5]University of Bristol, School of Geographical Sciences, Bristol, United Kingdom
[6]Christian-Albrechts-Universität zu Kiel, Department of Computer Science, Kiel, Germany

*Correspondence to:* Markus Schartau (mschartau@geomar.de) and Phil Wallhead (philip.wallhead@niva.no)

**Abstract.** To describe the underlying processes involved in oceanic plankton dynamics is crucial for the determination of energy and mass flux through an ecosystem and for the estimation of biogeochemical element cycling. Many planktonic ecosystem models were developed to resolve major processes so that flux estimates can be derived from numerical simulations. These results depend on the type and number of parameterisations incorporated as model equations. Furthermore, the values assigned to respective parameters specify a model's solution. Representative model results are those that can explain data, therefore data assimilation methods are utilised to yield optimal estimates of parameter values while fitting model results to match data. Central difficulties are 1) planktonic ecosystem models are imperfect and 2) data are often too sparse to constrain all model parameters. In this review we explore how problems in parameter identification are approached in marine planktonic ecosystem modelling.

We provide background information about model uncertainties and estimation methods, and how these are considered for assessing misfits between observations and model results. We explain differences in evaluating uncertainties in parameter estimation, thereby also discussing issues of parameter identifiability. Aspects of model complexity are addressed and we describe how results from cross-validation studies provide much insight in this respect. Moreover, approaches are discussed that consider time and space dependent parameter values. We further discuss the use of dynamical/statistical emulator approaches, and we elucidate issues of parameter identification in global biogeochemical models.

Our review discloses many facets of parameter identification, as we found many commonalities between the objectives of different approaches, but scientific insight differed between studies. To learn more from results of planktonic ecosystem models we recommend finding a good balance in the level of sophistication between mechanistic modelling and statistical data assimilation treatment for parameter estimation.

## 1 Introduction

The growth, decay, and interaction of planktonic organisms drive the transformation and cycling of chemical elements in the ocean. Understanding the interconnected and complex nature of these processes is critical to understanding the ecological and biogeochemical function of the system as a whole. The development of biogeochemical models requires accurate mathematical descriptions of key physiological and ecological processes, and their sensitivity to changes in the chemical and physical environment. Such mathematical descriptions form the basis of integrated dynamical models, typically composed of a set of differential equations that allow credible computations of the flux and transformation of energy (light) and mass (nutrients) within the ecosystem (U.S. Joint Global Ocean Flux Study Planning Report Number 14, *Modeling and Data Assimilation*, 1992).

Generalised mechanistic descriptions of how energy is absorbed and how mass becomes distributed in an ecosystem already exist, such as dynamic energy budget models (Kooij-

man, 1986) or the metabolic theory of ecology (Brown et al., 2004). But these theories still have limitations, and include incompatible assumptions (van der Meer, 2006). So far no fundamental ecophysiological principle has been further exacted beyond the conservation of mass. A consistent theme running through most ecosystem models is the determination of mass flux of certain biologically important elements, such as nitrogen, phosphorus, iron and carbon (N, P, Fe and C). Nonetheless, the precise details of how mass is transformed and allocated within an ecosystem is far from being established. For this reason, we find a large variety of plankton ecosystem models that differ in their number of state variables as well as in their parameterisation of individual physiological and ecological processes.

## 1.1   Mass flux induced by plankton dynamics

Dynamical marine, as well as limnic, ecosystem models usually start from a description of the build-up of biomass by photoautotrophic organisms (phytoplankton) as these take up dissolved nutrients from the water column and exploit light energy by photosynthesis. Phytoplankton biomass, as a product of primary production, is subsequently removed by natural mortality (cell lysis due to starvation, senescence, and viral attack), predation by zooplankton, and vertical export away from surface ocean layers via sinking of single or aggregated cells and of fecal pellets. Parameterisations of these three loss processes can be interlinked e.g. grazing of phytoplankton aggregates by large copepods. Depending on the trophic levels considered in a model, the predation among different zooplankton types (e.g. between herbivores, carnivores or omnivores) can be explicitly parameterised. Mortality and aggregation of phytoplankton cells and the excretion of organic matter (fecal pellets) by zooplankton act as primary sources of dead particulate organic matter (detritus) that can be exported to depth via sinking. Exudation by phytoplankton and bacteria can be a major source of labile dissolved organic matter that represents diverse substrates for remineralisation. The transformation of particulate and dissolved organic matter back to inorganic nutrients is parameterised as hydrolysis and remineralisation processes. Often hydrolysis and remineralisation are assumed to be proportional to the biomass of heterotrophic bacteria, which is considered in many models. Heterotrophic bacteria remain unresolved in some models where microbial remineralisation is parameterised only as a function of concentration and quality of organic substrates.

At some level most models include a parameterisation to account for the net effect of higher trophic levels that are not explicitly resolved. This is usually formulated as a closure flux back to nutrient pools and whose rates simply depend on the biomass of the highest trophic level resolved. These closure assumptions ensure mass conservation while neglecting the actual mass loss to higher trophic levels like fish, which would be subject to fish movements and changes in biomass

on multi-annual rather than seasonal time scales. Every marine planktonic ecosystem model can thus be described as a simplification of the dynamics inherent to a system of nutrients, phytoplankton, zooplankton, detritus, dissolved organic matter, and possibly bacteria.

In many cases marine ecosystem models are embedded in an existing physical ocean model setup that simulates environmental conditions, advection and mixing of the biological and chemical state variables. Feedbacks from the ecosystem model states on physical variables can be relevant (e.g., Murtugudde et al., 2002; Oschlies, 2004; Löptien et al., 2009; Löptien and Meier, 2011) but are rarely considered in current marine biogeochemical studies.

## 1.2   Parameters of plankton ecosystem models

Amongst the most influential model approaches to study the nitrogen flux through such a marine plankton ecosystem at a local site was proposed by Fasham et al. (1990). Their model involves 27 parameters and they stressed the invidious situation of finding a reliable ecosystem model solution by choosing parameter values that are uncertain or unknown. Laboratory measurements, as well as ship-based experiments with field samples, can provide information about the range of typical values for some parameters, for example the maximum growth rate of photo-autotrophs or the maximum ingestion rate of herbivorous plankton. Other model parameters are extremely difficult to measure, like exudation rates of dissolved organic carbon by phytoplankton or by bacteria. Another difficulty is that parameter values from laboratory experiments are often specific with respect to plankton species, temperature, and light conditions. Their values may not be directly applicable for ocean simulations where parameter values need to be representative for a mixture of different plankton species in a continuously varying physical environment. For example, for a natural composition of diverse phytoplankton cells that all differ in their genotypic- and phenotypic characteristics, we may expect values of some model parameters to follow a distribution rather than having a single fixed value.

In practice, there are always some fixed model parameters that need to be assigned values, whether they describe the behaviour of fixed plankton functional types or the distributions of traits in a stochastic community. In the end, it is the choice of these parameter values that determines a specific model solution of any ecological- or biogeochemical model setup.

## 1.3   The vital role of observational data

Model solutions of interest are typically those that can simulate and explain complex data. Model calibration, which can be considered a form of data assimilation (DA), is the process by which model parameter values are inferred from the observational data. Optimal parameter values are regarded

as those that generate model results that match observations (data-model misfit) but are also in accordance to the range of values known e.g. from experiments or from preceding DA studies. To determine optimal parameter estimates we have to account for uncertainties in data and in model dynamics as well, which is specified by an error model. Parameter estimates are thus conditioned by a) the dynamical model equations, b) the data, c) our prior knowledge about the range of possible parameter values, and d) the underlying error model (Evans, 2003).

Situations can occur where model results that are compared with data are insensitive to variations of some parameters. Values of those parameters remain unconstrained by the available data, which is a problem of parameter identifiability. The availability (type and number) of data thus places limitations on the number of model parameters whose values become identifiable, and values of some parameters may never be fully constrained. This in turn sets restrictions on the complexity of plankton interactions that can be unambiguously confined during ecosystem model calibration (Matear, 1995). Choosing appropriate model complexity is ambiguous and is still subject to discussion (e.g., Franks, 2002; Denman, 2003; Fulton et al., 2003; Anderson, 2005; Le Queré, 2006; Friedrichs et al., 2007; Franks, 2009; Kriest et al., 2012; Ward et al., 2013), a situation which sustains large differences in the level of complexity of current plankton models.

## 1.4 Inferences from data assimilation

Much of the literature on DA in oceanography is focussed on state estimation (e.g., Allen et al., 2003; Natvik and Evensen, 2003; Dowd, 2007; Nerger and Gregg, 2008; van Leeuwen, 2010). In these studies, the primary objective is to improve hindcasts, nowcasts, or forecasts of time-dependent variables such as chlorophyll $a$ (Chl$a$). However, many of the DA methods originally developed for state estimation have more recently been adapted to estimate static parameters, especially for stochastic models where random noise is injected into the model dynamics. Stochastic noise offers a plausible way to represent model error, but it should be noted that it can lead to violations of mass conservation unless it is injected in certain ways (e.g. by perturbing growth rate parameters). Deterministic plankton ecosystem models guarantee mass conservation and have a longer tradition in parameter estimation for marine ecosystem models, although they imply a less explicit treatment of model error. To identify and gradually eliminate model deficiencies it can be helpful to analyse model state and flux estimates while mass conservation is imposed as a strong constraint. The optimisation of only parameter values assures that simulation results remain dynamically and ecologically consistent, which is comparable with those DA approaches in physical oceanography that produce dynamically and kinematically consistent

solutions of ocean circulation (e.g., Wunsch and Heimbach, 2007; Wunsch et al., 2009).

Thorough reviews of common DA methods applied in marine biogeochemical modelling are given by Robinson and Lermusiaux (2002) and by Matear and Jones (2011). Dowd et al. (2014) provide a helpful and up-to-date overview of mainly sequential DA approaches where state estimation is combined with parameter estimation. Gregg et al. (2009) and Stow et al. (2009) discuss how the success of DA results of marine ecosystem models have been evaluated in the past and how model performance can be generally assessed. Fundamentals on DA that include aspects relevant to marine ecosystem and biogeochemical modelling are explained in Wikle and Berliner (2007) and in Rayner et al. (2016).

In our review we primarily focus on topics related to parameter identification, thereby including basic aspects of DA. Parameter identification in marine planktonic ecosystem modelling is a wide field and we do not attempt to discuss differences between various DA tools or techniques. We rather put emphasis on models, including parameterisations of ecosystem processes, statistical (error) models, model uncertainties, and structural complexity. We adopt and explain mathematical notation that is often used for DA studies in operational meteorology and oceanography. On the one hand we provide background information that should facilitate intelligibility when studying DA literature. On the other hand we like to elucidate typical objectives and common problems when simulating a marine planktonic system. In this manner we hope to support a mutual understanding between ecologically/biogeochemically and mathematically/statistically motivated studies.

The paper starts with some theoretical background information (Sect. 2), introducing mathematical notation and depicting prevalent assumptions that are typically made for parameter identification analyses and model calibration (Sect. 2.1). We then branch off from DA theory and discuss the parameters typically dealt with in plankton ecosystem models. In Sect. (3) we disentangle major differences between approaches to parameterising photoautotrophic growth and briefly discuss simple but common parameterisations of plankton loss rates. In this context we also address the utilisation of data from laboratory and mesocosm experiments. Error models are described in order to elucidate error assumptions made in previous ecosystem modeling studies (Sect. 4). This is followed by a description of different approaches to specify uncertainties in parameter values (Sect. 5). An example of parameter estimation with simulations of a mesocosm experiment connects aspects of Sect. (3) with the theoretical considerations of Sect. (5). Thereafter, model complexity is jointly addressed together with cross-validation in Sect. (6), followed by a review of space-time variations in marine ecosystem model parameters (Sect. 7). Emulator, or surrogate-based, approaches are briefly explained and exemplified (Sect. 8) before we discuss parameter estimation of large-scale and global biogeochemical ocean circulation

models (Sect. 9). Finally, we summarise the insights that we gained on parameter identification in Sect. (10), and we will briefly address prospects of some marine ecosystem model approaches that could improve parameter identification.

## 2  Theoretical background

The term parameter identification is used broadly to describe parameter estimation problems, including the specification of uncertainties in parameter estimates and model parameterisations. It involves the following procedures:

a) Parameter sensitivity analyses: the evaluation of how model results change with variations of parameter values.

b) Parameter estimation: the calibration of model results by adjusting parameter values in light of the data.

c) Parameter identifiability analyses: the specification of parameter uncertainties in order to reveal structural model deficiencies and shortages in data availability/information.

All three aspects are interrelated and should not be viewed as mutually exclusive procedures. For example, before starting with parameter estimation it is helpful to include information from a preceding sensitivity analysis, e.g. selecting only parameters to which model results are sensitive to. Likewise, an identifiability analysis complements the sensitivity analysis by providing information about error margins and possible ambiguities of optimal parameter estimates.

### 2.1  Statistical model formulation

#### 2.1.1  Model states, parameters, and dynamical model errors

The prognostic dynamical equations of a marine ecosystem model can be expressed as a set of difference equations:

$$\boldsymbol{x}_{i+1} = M\left[\boldsymbol{x}_i, \theta_e, \boldsymbol{f}_i, \boldsymbol{\eta}_i(\theta_\eta)\right] \tag{1}$$

with index $i$ representing a particular time step (i.e. $t_i$). The model state vector $\boldsymbol{x}_i$ has dimension $N_x = N_g \times N_s$ where $N_g$ is the number of spatial grid points and $N_s$ is the number of model state variables (e.g. phytoplankton biomass). The dynamical model operator $M$ is typically at least a nonlinear function of the earlier state $\boldsymbol{x}_i$, a set of ecosystem parameters $\theta_e$ describing rate constants and coefficients in the dynamical model, and a set of time and space dependent forcings and boundary conditions $\boldsymbol{f}_i$. If the ecosystem model is coupled "online" with a physical ocean model, $\boldsymbol{f}_i$ includes both physical model forcings (e.g. wind stress) and ecosystem model forcings (e.g. surface short-wave irradiance). If the physics is coupled "offline", $\boldsymbol{f}_i$ includes ecosystem model forcings and physical model outputs (e.g. seawater temperature).

For stochastic dynamical models, $M$ also depends on random noise variables or dynamical model errors $\boldsymbol{\eta}_i$ while for deterministic models we have $\boldsymbol{\eta}_i = 0$. These errors are described by distributional parameters $\theta_\eta$, e.g. location and scale parameters of a probability density function. Dynamical model errors usually enter the dynamics additively, multiplicatively, or as time/space- dependent corrections to $\boldsymbol{f}$ or $\theta_e$. They may represent the individual or combined effects of errors in forcings, boundary conditions, random variability in model parameters, and structural errors in both the physical transport model (e.g. due to limited spatial resolution) and the biological source-minus-sink terms (e.g. due to aggregation of species into model groups). In the geophysical DA community, error models that explicitly account for dynamical model errors (noise) are often termed *weak constraint* models, while those that assume a deterministic model are termed *strong constraint* (Sasaki, 1970; Bennett, 2002, page 25).

#### 2.1.2  True states and kinematic model errors

To relate the dynamical model output of Eq. (1) to observations, it is helpful to first consider how it may relate to a conceptual and hypothetical true state $\boldsymbol{x}^t$, which is then imperfectly observed. In this respect we must also consider the averaging scales. In marine ecosystem modelling there is almost always a large discrepancy between the spatio-temporal averaging scales of the model, that define the meaning of the "concentrations" in $\boldsymbol{x}$, and the averaging scales of the observations from in-situ sampling or remote sensing. For example, the spatial averaging scale of a model may be defined by a model grid cell of size 10 km in the horizontal and 10 m in the vertical, while the averaging scale of the observations might be the 10 cm scale, e.g. of a Niskin bottle sample. Even with a perfect model, data from finescale observations may diverge from model output due to unresolved sub-grid scale variability induced by fluid structures such as eddies and fronts, forming patches of high next to low concentrations e.g. of nutrients or organic matter.

A general relationship between the true state and model state can be expressed as:

$$\boldsymbol{x}^t = T\left[\boldsymbol{x}, \boldsymbol{\zeta}(\theta_\zeta)\right] \tag{2}$$

where $T$ is a truth operator, and $\boldsymbol{\zeta}$ is a set of random variables described by distributional parameters $\theta_\zeta$. We will refer to the $\boldsymbol{\zeta}$ as *kinematic* model errors because they are associated with the model state, while the *dynamical* model errors $\boldsymbol{\eta}$ in Eq. (1) act to perturb the model dynamics. The true values of the kinematic model errors therefore define the potential discrepancy between the target true state and a hypothetical ideal model output (i.e. with the "true" values of the parameters and, if applicable, also with the "true" values of the dynamical model errors).

How we interpret and specify Eq. (2) depends on the spatio-temporal averaging scales chosen to define the true

state $\boldsymbol{x}^t$, which in turn depends on the objectives of the modelling study. One approach is to define these averaging scales as equal to or larger than the shortest space and time scales that are fully resolved by the model. Kinematic model errors $\zeta$ may then represent the integrated effects of the various dynamical sources of model error, if these are not already accounted for by dynamical model errors $\boldsymbol{\eta}$ in Eq. (1). Alternatively, the true state can be defined over scales smaller than those resolved by the model, possibly at the scales of the observations. This may lead to a simpler model for observational error (see below), but now the $\zeta$ must account for the unresolved scales, in addition to any error effects in the model dynamics otherwise not accounted for. With stochastic dynamical models ($\boldsymbol{\eta} \neq 0$), the true state is usually defined on the scales of the model and assumed to coincide with the model output for some ($\theta_e$, $\boldsymbol{\eta}$), such that no kinematic error model is needed.

### 2.1.3 Data and observational errors

The observation vector **y** can be related to the true state via:

$$\boldsymbol{y} = O\left[\boldsymbol{x}^t, \boldsymbol{\epsilon}(\theta_\epsilon)\right] \tag{3}$$

where $O$ is the generalized observation operator and $\boldsymbol{\epsilon}$ is a set of random *observational* errors described by distributional parameters $\theta_\epsilon$ and accounting for uncertainties associated with the usage and interpretation of the data. These include at least the random measurement error due to, for example, instrument noise. In addition they may include a contribution from *representativeness* error due to finescale variability, if $\boldsymbol{x}^t$ is defined as an average over larger scales than those of the observations (see above). Alternatively, if the observations are preprocessed into estimates on the larger scales of $\boldsymbol{x}^t$, there may be an *undersampling* error component due to inexhaustive coverage of the raw samples. The observation operator $O$ may also contribute to $\boldsymbol{\epsilon}$, for example if the model output needs to be interpolated from the model grid to the data coordinates, or if $O$ includes conversion factors such as chlorophyll *a*-to-nitrogen (Chl*a*:N) ratios.

The simplest possible example of an observational error model assumes additive Gaussian errors. Equation (3) then becomes:

$$\boldsymbol{y} = H\left(\boldsymbol{x}^t\right) + \boldsymbol{\epsilon} \tag{4}$$
$$\longrightarrow \boldsymbol{\epsilon} = \boldsymbol{y} - H\left(\boldsymbol{x}^t\right)$$

where $H$ accounts for interpolation and units conversion and $\boldsymbol{\epsilon} \sim G(0, \mathbf{R})$ is Gaussian distributed with mean zero and covariance matrix $\mathbf{R}$. This may be a reasonable error model for most physical variables and chemical concentrations with ranges well above zero (e.g. dissolved inorganic carbon or total alkalinity in the open ocean). However, many nutrients and plankton biomass variables may vary close to their lower bounds of zero, and display positive skew in their observational errors. For such variables, a lognormal observational

error model may be more appropriate:

$$\boldsymbol{y} = H\left(\boldsymbol{x}^t\right) \circ \exp\left(\tilde{\boldsymbol{\epsilon}} - \frac{\tilde{\boldsymbol{\sigma}}^2}{2}\right) \tag{5}$$
$$\longrightarrow \tilde{\boldsymbol{\epsilon}} = \log(\boldsymbol{y}) - \log\left(H\left(\boldsymbol{x}^t\right)\right) + \frac{\tilde{\boldsymbol{\sigma}}^2}{2}$$

where $\circ$ denotes element-wise multiplication and $\tilde{\boldsymbol{\sigma}}^2$ denotes the variance in logarithmic space. The bias correction term ($\tilde{\boldsymbol{\sigma}}^2 / 2$) ensures unbiased errors, but is frequently neglected in practice. The various options and challenges of defining an appropriate error model are discussed in detail in Sect. (4).

### 2.2 Estimation methods

#### 2.2.1 Basic probabilistic approaches

We now consider how to estimate uncertain parameters $\Theta$ given the data $\boldsymbol{y}$, where $\Theta$ includes all biological parameters $\theta_e$ and possibly distributional parameters ($\theta_\eta, \theta_\zeta, \theta_\epsilon$). There are basically two probabilistic approaches for doing this: Bayesian estimation and maximum likelihood estimation. In the Bayesian approach, we treat the parameters as random variables, and choose parameter values on the basis of their 'posterior probability' i.e. the conditional probability density of the parameter values given the data $p(\Theta \mid \boldsymbol{y})$. The posterior probability is computed using Bayes' theorem:

$$p(\Theta \mid \boldsymbol{y}) = \frac{p(\boldsymbol{y} \mid \Theta) \cdot p(\Theta)}{p(\boldsymbol{y})} \propto p(\boldsymbol{y} \mid \Theta) \cdot p(\Theta) \tag{6}$$

where $p(\boldsymbol{y} \mid \Theta)$ is the likelihood and $p(\Theta)$ is the unconditional or 'prior' distribution of the parameter values. The proportionality follows in Eq. (6) because the probability of the data $p(\boldsymbol{y})$, otherwise known as the "evidence" for the model, is independent of the parameter values.

In general the likelihood can be expressed as an integral over probabilities conditioned on particular values of the model state and true state:

$$p(\boldsymbol{y} \mid \Theta) = \int \int p(\boldsymbol{y} \mid \boldsymbol{x}^t, \Theta) \cdot p(\boldsymbol{x}^t \mid \boldsymbol{x}, \Theta) \cdot p(\boldsymbol{x} \mid \Theta) \, \mathrm{d}\boldsymbol{x}^t \, \mathrm{d}\boldsymbol{x}$$
$$\tag{7}$$

where the conditional probabilities $p(\boldsymbol{y} \mid \boldsymbol{x}^t, \Theta)$, $p(\boldsymbol{x}^t \mid \boldsymbol{x}, \Theta)$, and $p(\boldsymbol{x} \mid \Theta)$ are specified by the chosen models for observational error (Eq. 3), kinematic model error (Eq. 2), and dynamical model error (Eq. 1) respectively. In practice we are unlikely to require such a complex expression for numerical evaluation; aggregation of error terms and redundancy between kinematic and dynamical model error usually allows simplifications.

The Bayesian approach encourages us to explicitly quantify our prior knowledge about the parameter values through the prior $p(\Theta)$. In marine ecosystem modelling, we are unlikely to ever consider cases of complete parameter ignorance, where a parameter value could possibly switch sign

or get incredibly large. Every parameter is expected to have a value that falls into a credible range, otherwise the associated parameterisation would be difficult to defend. In some cases, when broad uniform or "uninformative" priors are assumed, it may not be necessary to specify exact limits of these distributions as the analyses may become insensitive to these limits once the range becomes sufficiently broad. There are inherent difficulties with the concept of "ignorance" priors: for example, a flat prior distribution over $\phi$ will correspond to an informative prior for some function $g(\phi)$ (see Cox and Hinkley, 1974 for further discussion). In any case, trying to minimise the impact of prior distributions is rather defeating the object of Bayesian estimation, which explicitly aims to synthesise information from new data with prior information from previous analyses.

Once the likelihood is formulated and a prior distribution is prescribed, classical Bayes estimates (BEs) may be computed from posterior mean or posterior median values of $\Theta$. Assuming the statistical assumptions are correct, these estimators will minimise the mean square error or mean absolute error respectively of the parameter estimate $\widehat{\Theta}$ (e.g., Young and Smith, 2005). To obtain BEs can be computationally expensive, requiring sophisticated techniques to sample efficiently from the posterior distribution (e.g. by Markov Chain Monte Carlo, MCMC, methods). An alternative Bayesian estimator, very widely used in geosciences, is the joint posterior mode or maximum a posteriori (MAP) estimator (e.g., Kasibhatla, 2000; Bocquet, 2014), given by maximising the posterior probability $p(\Theta \,|\, \boldsymbol{y})$ as a function of $\Theta$. Such estimates are more computationally feasible in large problems where the search for the maximum of the posterior (or the minimisation of its negative logarithm) can be greatly accelerated by techniques such as the variational adjoint (Bennett, 2002, Chapter 4).

In maximum likelihood (ML) estimation we seek the parameter values $\widehat{\Theta}_{\mathrm{ML}}$ that maximise the probability of the data given the parameter set, i.e. $p(\boldsymbol{y} \,|\, \Theta)$. When considered as a function of $\Theta$, this probability is called the *likelihood* of the parameter values $L(\Theta \,|\, \boldsymbol{y})$ because it is strictly a probability of the data, not of the parameter values. Indeed, in ML estimation we do not need to consider the parameter values as random variables at all; rather they are considered as fixed, unknown constants. For this reason the '|'s are sometimes replaced by ';'s to emphasise that, in a non-Bayesian context, the likelihood is not a conditional probability in the sense of one set of random variables dependent on another (e.g., Cox and Hinkley, 1974). In the ML approach, no prior information on the parameter values is used except possibly to define upper or lower plausible limits or allowed ranges for the parameter search (Young and Smith, 2005).

Historically, Bayesian methods (Bayes, 1763; Bayes and Price, 1763) predate ML methods of Fisher (1922) by some margin. Fisher introduced ML methods partly to avoid problems in defining prior ignorance (see above) but also to avoid the noninvariance property of Bayesian estimators (Hald, 1999). This property means that given the BE of one parameter $\widehat{\phi}_{\mathrm{B}}$, the corresponding BE of a nonlinear function of that parameter $g(\phi)$ is not simply given by plugging in the estimate ($\widehat{g}_{\mathrm{B}} \neq g(\widehat{\phi}_{\mathrm{B}})$), while for ML estimates the invariance property does hold ($\widehat{g}_{\mathrm{ML}} = g(\widehat{\phi}_{\mathrm{ML}})$). We will see an example of this in Sect. (2.3).

### 2.2.2 Sequential methods

In some problems, assimilating all the data at once from all available sampling times can be computationally impractical. This is particularly likely for models with stochastic dynamics ($\boldsymbol{\eta} \neq 0$ in Eq. 1), if the data are clustered in time, or if model states need to be repeatedly updated as new data come in. In such cases a sequential approach can be expedient. The basic idea is to break the large integration problem defined by Eq. (7) into a number of smaller problems by sequentially assimilating observations in subsets defined by sampling time. The method comprises a consecutive sequence of two major steps, a forecast- and an analysis step respectively. If the sequential algorithm is accurate, it should approximate the posterior parameter distribution defined by Eqs. (6 and 7) at times where all available data have been assimilated.

To see how this works, suppose we know the probability density $p(\boldsymbol{x}_j^t \,|\, \boldsymbol{y}_{1:j}, \Theta)$ of the true state at sampling time $t_j$ (possibly an initial condition) for a given value of the uncertain parameters $\Theta$ and given all the previously assimilated observations $\boldsymbol{y}_{1:j}$ (possibly null). The probability density at sampling time $t_{j+1}$ is given by the forecast density:

$$p(\boldsymbol{x}_{j+1}^t \,|\, \boldsymbol{y}_{1:j}, \Theta) = \int p(\boldsymbol{x}_{j+1}^t \,|\, \boldsymbol{x}_j^t, \Theta) \cdot p(\boldsymbol{x}_j^t \,|\, \boldsymbol{y}_{1:j}, \Theta) \, \mathrm{d}\boldsymbol{x}_j^t \tag{8}$$

In general this integral can be approximated by an ensemble of Monte Carlo simulations, sampling an initial condition from $p(\boldsymbol{x}_{j+1}^t \,|\, \boldsymbol{y}_{1:j}, \Theta)$ and then running the model to the next sampling time $t_{j+1}$ (possibly including stochastic dynamical noise, and possibly accounting for kinematic model error). Next, in the analysis step, the new observations are assimilated by applying Bayes' theorem:

$$p(\boldsymbol{x}_{j+1}^t \,|\, \boldsymbol{y}_{1:(j+1)}, \Theta) \propto p(\boldsymbol{y}_{j+1} \,|\, \boldsymbol{x}_{j+1}^t, \Theta) \cdot p(\boldsymbol{x}_{j+1}^t \,|\, \boldsymbol{y}_{1:j}, \Theta), \tag{9}$$

which again can be approximated e.g. by Monte Carlo sampling. The forecast and analysis steps can then be repeated until all the data are assimilated. Note that Eq. (9) assumes conditional independence of the observations, allowing us to write $p(\boldsymbol{y}_{j+1} \,|\, \boldsymbol{x}_{j+1}^t, \Theta)$ instead of $p(\boldsymbol{y}_{j+1} \,|\, \boldsymbol{x}_{j+1}^t, \boldsymbol{y}_{1:j}, \Theta)$. This amounts to assuming that the observational errors are independent between sampling times (Evensen, 2009), which may not be strictly true if sampling is frequent and if there is a noticeable contribution from representativeness/undersampling, or from errors in conversion factors (see Sect. 2.1.3).

Once the predictive filtering densities $p(\boldsymbol{x}_{j+1}^t \mid \boldsymbol{y}_{1:j}, \Theta)$ have been approximated for all sampling times ($t_j$ with $j = 1, \ldots, N_t$), these can be used to approximate the likelihood in Eq. (7), since:

$$p(\boldsymbol{y} \mid \Theta) = \prod_{j=1}^{N_t} p(\boldsymbol{y}_j \mid \boldsymbol{y}_{1:j-1}, \Theta) \qquad (10)$$

$$= \prod_{j=1}^{N_t} \int p(\boldsymbol{y}_j \mid \boldsymbol{x}_j^t, \boldsymbol{y}_{1:j-1}, \Theta) \cdot p(\boldsymbol{x}_j^t \mid \boldsymbol{y}_{1:j-1}, \Theta) \, \mathrm{d}\boldsymbol{x}_j^t$$

$$= \prod_{j=1}^{N_t} \int p(\boldsymbol{y}_j \mid \boldsymbol{x}_j^t, \Theta) \cdot p(\boldsymbol{x}_j^t \mid \boldsymbol{y}_{1:(j-1)}, \Theta) \, \mathrm{d}\boldsymbol{x}_j^t$$

For $j=1$ in Eq. (10) we have a set of zero members and $p(\boldsymbol{y}_j \mid \boldsymbol{y}_{1:j-1}, \Theta) = p(\boldsymbol{y}_1 \mid \Theta)$. The third line of Eq. (10) again assumes conditional independence of the observations and the final integral can in general be approximated using the predictive ensembles (see Jones et al., 2010; Dowd, 2011; Dowd et al., 2014). This procedure can be repeated for different values of $\Theta$ and combined with Eq. (6) to assess posterior probability.

Alternatively, $p(\Theta \mid \boldsymbol{y})$ can be calculated from a single application of the filter using a 'state augmentation' approach whereby the parameters $\Theta$ are appended to the vector $\boldsymbol{x}$ as additional state variables with zero dynamics. In practice, random parameter noise may need to be added to avoid filter degeneracy, such that this approach may be considered a separate estimation method (Dowd, 2011). However, if such ad hoc noise can be avoided, or if the parameters are in fact assumed to vary stochastically, then the augmented-state filter at the end of the assimilation interval should approximate the theoretical Bayesian posterior for this time. For other times, a 'smoother' algorithm would be required. A further benefit of the augmented-state filter is that the parameter estimates for intermediate time periods may show temporal pattterns that expose deficiencies in the model formulation and provide useful information for model development (e.g., Losa et al., 2003).

The various types of filter differ essentially in terms of how the integrals in Eqs. (8) and (9) are approximated. Particle filters (van Leeuwen, 2009) use Monte Carlo sampling for both steps while the Ensemble Kalman Filter (Evensen, 2003, 2009) uses Gaussian and linear approximations for the analysis step, enabling the use of smaller ensembles but at the cost of lower accuracy in strongly nonlinear/non-Gaussian problems. The (Extended) Kalman Filter applies when the model dynamics are (quasi-) linear and both model and observational errors are Gaussian. These conditions allow both integrals to be evaluated analytically, but appear to be rarely applicable to parameter estimation in marine ecosystem models. For reviews of sequential approaches the reader is referred to Dowd et al. (2014) for marine biogeochemical modelling and to Bertino et al. (2003) for oceanography in general.

### 2.2.3 Variational methods

At present there appears to be some ambiguity regarding the term "variational" in the context of DA. It is sometimes used to describe approaches explicitly based on control theory or "inverse methods" that may not include explicit assumptions on error distributions and where cost functions are defined a priori, rather than being derived from statistical or probabilistic models. However, a distribution-free approach seems difficult to recommend in general for marine ecosystem model parameter estimation, given the strong nonlinearity, non-Gaussianity, and relatively weak data constraint often encountered in such problems. Within the marine ecosystem modelling community, the term "variational DA" is often used more broadly to refer to all non-sequential methods that involve the minimisation of a cost function, whether or not this is based on a probability model.

In any case, there are some powerful mathematical tools developed for variational DA that can be applied to minimise cost functions. Adjoint methods allow the gradient of the cost function with respect to all fitted parameters to be computed in an extremely efficient manner, see Lawson et al. (1995), and Appendix (C). This is particularly useful when dealing with a large number of fitted parameters (high-dimensional $\Theta$) of computationally expensive models (e.g., Tjiputra et al., 2007). The application of the adjoint method helps reducing the number of model runs to provide access to joint posterior mode and maximum likelihood estimates.

Pelc et al. (2012) provide useful theoretical background for different 4DVar approaches (four-dimensional, in space and time, variational approaches) and show how this adjoint method can be used to estimate ecosystem model parameters jointly with a large number of initial condition parameters. See also Bennett (2002) for an introduction to variational DA and adjoint methods in physical oceanography.

However, it can be disadvantageous to employ a search algorithm that relies too much on local gradients (e.g. from an adjoint model) to minimise the cost function, because this may result in finding a local minimum rather than the global minimum that defines the MAP or ML estimate (Vallino, 2000). This issue appears to be frequently encountered in marine ecosystem modelling applications, and should be expected as a product of strong nonlinearity and weak data/prior constraint. For such cases, a non-local approach such as simulated annealing, following Hurtt and Armstrong (1996, 1999) or a microgenetic algorithm, following Schartau and Oschlies (2003), may be preferable, at least during an initial period of the search before the broader region of the global minimum is located (Ward et al., 2010). The main drawback of these non-local search algorithms is that they tend to require a larger number of model runs (at least order of $10^3$) to have a good chance of accurately locating the global minimum, although they may yet provide meaningful improvements to prior parameter estimates for order of 100 runs (Mattern and Edwards, 2017).

#### 2.2.4   Recent approaches

Much recent interest has focused on combined state and parameter estimation, whereby model parameters $\Theta$ are estimated together with a true state $\boldsymbol{x}^t$ (e.g., Simon and Bertino, 2012; Fiechter et al., 2013; Parslow et al., 2013; Weir et al., 2013; Dowd et al., 2014). In the Bayesian approach, model parameters and system state are both random variables. We can therefore apply Bayes' Theorem to the composite random variable $\Psi = (\Theta, \boldsymbol{x}^t)$ and decompose the prior as $p(\Psi) = p(\boldsymbol{x}^t \mid \Theta) \cdot p(\Theta)$ to obtain an expression for the joint posterior:

$$p(\boldsymbol{x}^t, \Theta \mid \boldsymbol{y}) \propto p(\boldsymbol{y} \mid \boldsymbol{x}^t, \Theta) \cdot p(\boldsymbol{x}^t \mid \Theta) \cdot p(\Theta) \qquad (11)$$

This equation has so far been applied to stochastic dynamic models with no kinematic model error (cf. Fiechter et al., 2013; Parslow et al., 2013). Equation (6) can be recovered from Eq. (11) by integrating (marginalising) both sides over $\boldsymbol{x}^t$.

In some other recent studies emphasis is put on "hierarchical" error models (Zhang and Arhonditsis, 2009; Parslow et al., 2013; Wikle et al., 2013). Here, the traditional model parameters are replaced with stochastic processes over time and/or space, and parameter identification focuses on the *hyperparameters* that describe the stochastic processes (e.g. means, variances, autocorrelation parameters). This is essentially similar to the case of parameter estimation for a stochastic dynamical model (Sect. 2.2.2) and fits into the general formulation in Sect. (2.1), if we treat the stochastic parameters as additional state variables with dynamical model errors $\boldsymbol{\eta}$. The hyperparameters could in principle be estimated by ML, sometimes referred to as an "empirical Bayesian" approach (Cox and Hinkley, 1974), but it appears that computational tractability may favour the "hierarchical Bayesian" approaches (e.g., Zhang and Arhonditsis, 2009), which may also make use of sequential Monte Carlo methods (e.g., Jones et al., 2010; Parslow et al., 2013).

Another important initiative is the estimation of hyperparameters of the kinematic error model along with the ecosystem parameters (Arhonditsis et al., 2008). The posterior of the kinematic model error provides an estimate of the model discrepancy, introduced by Kennedy and O'Hagan (2001) and originally referred to as model inadequacy. The model discrepancy is defined as the model error for the "true" values of the model parameters, i.e. the unknown values of the parameters for which the model best represents $\boldsymbol{x}^t$. Estimates of model discrepancies may thus provide useful diagnostics for model skill assessment and development.

### 2.3   From statistical model to cost function

The choice of a suitable estimation method for marine ecosystem model parameters should be mainly based on the availability of relevant prior information, as well as on the basic error assumptions (Eqs. 1, 2, 3). Once the error model

and estimation method have been chosen, we can derive the probability densities and cost functions that can be used for parameter estimation.

As a simple but common example, consider a deterministic model with no model error and data with additive Gaussian observational errors, Eq. (4), with known covariance matrix $\mathbf{R}$. We wish to use a total of $N_y$ data, summing over all data types, to estimate $N_\Theta$ parameters by Bayesian estimation. A survey of the literature might lead us to model the prior distribution of $\Theta$ as Gaussian with a mean $\Theta^b$ and covariance matrix $\mathbf{B}$. From Eq. (6) the posterior density is proportional to a product of the likelihood and the prior density:

$$p(\Theta \mid \boldsymbol{y}) \propto \frac{1}{\sqrt{(2\pi)^{N_y} \det \mathbf{R}}} \cdot \exp\left[ -\frac{1}{2}\, \boldsymbol{d}^T \mathbf{R}^{-1} \boldsymbol{d} \right]$$
$$\cdot \frac{1}{\sqrt{(2\pi)^{N_\Theta} \det \mathbf{B}}} \cdot \exp\left[ -\frac{1}{2} \boldsymbol{\Delta}_\Theta^T \mathbf{B}^{-1} \boldsymbol{\Delta}_\Theta \right] \qquad (12)$$

where the data-model residual $\boldsymbol{d}$ is defined by $\boldsymbol{d} = \boldsymbol{y} - H(\boldsymbol{x})$ (see $\boldsymbol{\epsilon}$ in Eq. 4). The deviation from the prior is $\boldsymbol{\Delta}_\Theta = \Theta - \Theta^b$. A MAP or joint posterior mode estimate of $\Theta$ can then be obtained by minimising the cost function $J(\Theta) = -2 \log p(\Theta \mid \mathbf{y}) + \text{constant}$, given by:

$$J(\Theta) = \boldsymbol{d}^T \mathbf{R}^{-1} \boldsymbol{d} + \boldsymbol{\Delta}_\Theta^T \mathbf{B}^{-1} \boldsymbol{\Delta}_\Theta \qquad (13)$$

where constant terms (since independent of $\Theta$) have been dropped.

Alternatively, nonnegativity constraints on the variables and parameters may lead us to prefer the lognormal observational error model. Likewise, we can assume lognormal priors for the parameters. In this case the posterior density becomes:

$$p(\Theta \mid \boldsymbol{y}) \propto \frac{1}{\sqrt{(2\pi)^{N_y} \det \tilde{\mathbf{R}} \prod_j y_j}} \cdot \exp\left[ -\frac{1}{2}\, \tilde{\boldsymbol{d}}^T \tilde{\mathbf{R}}^{-1} \tilde{\boldsymbol{d}} \right]$$
$$\cdot \frac{1}{\sqrt{(2\pi)^{N_\Theta} \det \tilde{\mathbf{B}} \prod_l \Theta_l}} \cdot \exp\left[ -\frac{1}{2} \tilde{\boldsymbol{\Delta}}_\Theta^T \tilde{\mathbf{B}}^{-1} \tilde{\boldsymbol{\Delta}}_\Theta \right] \; (14)$$

where the data-model residuals and parameter corrections on the transformed scale are defined by $\tilde{\boldsymbol{d}} = \log(\boldsymbol{y}) - \log(H(\boldsymbol{x})) + \dfrac{\tilde{\boldsymbol{\sigma}}^2}{2}$ and

$\tilde{\boldsymbol{\Delta}}_\Theta = \log(\Theta) - \log(\Theta^b) + \dfrac{(\tilde{\boldsymbol{\sigma}}^b)^2}{2}$. A MAP estimator of $\Theta$ is then obtained by minimising:

$$J(\Theta) = \tilde{\boldsymbol{d}}^T \tilde{\mathbf{R}}^{-1} \tilde{\boldsymbol{d}} + 2 \sum_{l=1}^{N_\Theta} \log(\Theta_l) + \tilde{\boldsymbol{\Delta}}_\Theta^T \tilde{\mathbf{B}}^{-1} \tilde{\boldsymbol{\Delta}}_\Theta \qquad (15)$$

The MAP or posterior mode estimator of $\log(\Theta)$ is equivalent here to the posterior median estimate and is obtained by maximising $p(\log(\Theta) \mid \boldsymbol{y})$. This leads to a cost function given by Eq. (15) without the second term, $2 \sum_{l=1}^{N_\Theta} \log(\Theta_l)$ (cf., Fletcher, 2010). Due to the noninvariance property of

Bayesian estimates, the exponent of the MAP estimator of $\log(\Theta)$ will generally differ from the MAP estimator of $\Theta$. By contrast, ML estimates are obtained by minimising the cost functions without any of the prior terms (second terms in Eq. 13, second and third terms in Eq. 15). In each case the same ML estimator for $\Theta$ is obtained whether we use $\Theta$ or $\log(\Theta)$, as expected from the invariance property of ML estimates.

## 2.4   Remarks on data assimilation terminology

We close this section with some cautionary remarks about different terminology that the reader may encounter the literature. First, many DA papers and textbooks start by assuming a certain cost function, based on variational or optimal control theory, rather than deriving it from a probabilistic treatment as herein (e.g., Le Dimet and Talagrand, 1986; Bennett, 2002; Fletcher, 2010). These studies tend to refer to MAP estimates obtained by minimising cost functions such as Eq. (13) as "weighted least squares estimates". However, any analogy with regression analysis is stretched because these estimates are fundamentally dependent on, and potentially biased by, the assumed prior distributions. Second, many DA papers and textbooks use the term "likelihood" to refer to the posterior probability $p(\Theta \,|\, \boldsymbol{y})$ in Eq. (6), and the term "maximum likelihood estimators" although modifiers such as "(Bayesian)" (Jazwinski, 2007, page 156) or "(posterior)" (Tarantola, 2005, page 40) are sometimes added. This obscures the fact that posterior mode estimators, like all BEs, are dependent on assumed prior distributions. Maximum likelihood avoids this dependence, but in doing so tends to be unsuitable for high-dimensional parameter estimation in the partially-observed systems typically encountered in oceanography and geophysics.

## 3   Typical parameterisations of plankton models and their parameters

Deviant parameter estimates of a model may point towards a deficiency in model structure, forcing, or in boundary conditions. Estimates of the effectively same parameters may turn out to be different within dissimilar plankton ecosystem models, even if those models may have been calibrated with the same data and although they possibly share an identical physical (environmental) setup. To understand why parameter estimates can be different it is helpful to unravel some of the basic differences between major parameterisations that describe growth and loss rates of phytoplankton.

A crucial element of most plankton ecosystem models is the description of phytoplankton growth as a function of light, temperature, and nutrient availability. How growth of algae is parameterised is relevant and the associated parameter values affect timing and intensity, e.g. of a phytoplankton bloom in model solutions.

## 3.1   Differences between maximum carbon fixation and maximum growth rate

The build up of phytoplankton biomass depends on how much of the available nutrients can be utilised and how much energy can be absorbed from sun light. Under nutrient-replete and light-saturated conditions, the carbon fixation (gross primary production, GPP) reaches a (temperature dependent) maximum rate, described as a parameter ($P_m^C$) with unit $\mathrm{d}^{-1}$. For models that do not resolve mass flux of carbon explicitly, $P_m^C$ is substituted by a maximum growth rate ($\mu_m$) to express the phytoplankton's maximum assimilation rate of nitrogen (N), or of phosphorus (P). The maximum GPP and the maximum growth rate are interrelated and in principle one can be derived from the other (Smith, 1980). In reality, maximum C-fixation, maximum N- or P-assimilation, and cell doubling rates are highly variable. This requires at least cellular C, N and Chl$a$ to be explicitly resolved, (linking for example, intracellular nutrient allocation to photoacclimation Shuter, 1979; Laws et al., 1983; Pahlow, 2005; Armstrong, 2006).

In practice an analogy between $P_m^C$ and $\mu_m$ is often assumed in N- or P- based biogeochemical models (assuming fixed stoichiometric elemental C:N:P ratios for algal growth). The parameter $P_m^C$ or $\mu_m$ is typically multiplied with a dimensionless temperature function ($f_T$) (e.g., Arrhenius, 1889; Eppley, 1972), allowing for temperature induced changes of metabolic rates. The actual potential maximum rate ($P_m^C \cdot f_T$ or $\mu_m \cdot f_T$) is then reached at some prefixed reference or optimum temperature accordingly. In early N-based plankton modelling studies (e.g., Evans and Parslow, 1985; Fasham et al., 1990; Doney et al., 1996) the maximum growth rate was mainly adopted from Eppley (1972). In subsequent DA studies this maximum rate was either subject to optimisation (e.g., Fasham and Evans, 1995; Spitz et al., 2001) or it was kept fixed because then parameter values of the limitation functions could be better identified (Matear, 1995; Fennel et al., 2001).

## 3.2   Combining parameterisations of light- and nutrient limitation

In many marine ecosystem models two separate limitation functions are combined: one that expresses the photosynthesis versus light relationship (P-I curve) and another that describes the dependence between ambient nutrient concentrations and nutrient uptake. The two functions are similar in their characteristics, starting from zero (no light or no nutrients) and approaching saturation at some high light and at replete nutrient concentration. Three approaches are generally found in marine ecosystem models to limit algal growth by photosynthesis and nutrient uptake. The first is to apply Blackman's law (Blackman, 1905), assuming that growth is reduced by the most limiting factor, either by light or by nutrient availability (e.g., Hurtt and Armstrong, 1996; Oschlies

and Garçon, 1999; Klausmeier and Litchman, 2001). The second is to multiply both limitation functions (e.g., Evans and Parslow, 1985; Fasham et al., 1990; Follows et al., 2007). The third approach involves combinations of light- and nutrient limitation that resolve interrelations between cell quota, N-uptake and the photoacclimation state of the algae (e.g., Armstrong, 2006, see Sect. 3.5 ). Whether the first, second or third approach is considered can be expected to affect estimates of the associated parameter values.

### 3.3   Photosynthesis as a function of light (P-I curve)

In a P-I curve the level of increase from low to high irradiance is specified by the initial slope parameter (the maximum of the first derivative of the P-I curve with respect to light), also referred to as photosynthetic efficiency ($\alpha_{\text{phot}}$) (Smith, 1936; Jassby and Platt, 1976; Cullen et al., 1992; Baumert, 1996). Photosynthetic efficiencies were derived from P-I measurements, for example by Platt and Jassby (1976), Peterson et al. (1987), and Platt et al. (1992) and their mean values were used for many N-based models (e.g., Fasham et al., 1990; Sarmiento et al., 1993; Doney et al., 1996; Oschlies and Garçon, 1999). Published measurements of $\alpha_{\text{phot}}$ were typically normalised to Chl$a$ concentrations. In case of N- or P-based models careful considerations are then needed with respect to the phytoplankton's cellular Chl$a$ content, which can vary by a factor of ten and more. Values of $\alpha_{\text{phot}}$ were found to vary by a factor of three (Côté and Platt, 1983) during a three month period, which can be attributed to changes in phytoplankton community structure as well as to photoacclimation. Platt and Jassby (1976) reported an even larger variational range over a one year period, from $\alpha_{\text{phot}} = 0.03$ to 0.63 mg C (mg Chl$a$)$^{-1}$ h$^{-1}$ W$^{-1}$ m$^2$ within the upper ten meters.

### 3.4   Algal growth and nutrient limitation

Typical parameterisations of growth limitation by nutrient availability (ambient nutrient concentrations) are expressed with the half-saturation constant ($K_s$) of a classical Monod equation (Monod, 1942, 2012). Another approach is to parameterise limitations of the nutrient uptake rate, described with a parameter referred to as nutrient affinity ($\alpha_{\text{aff}}$) (Aksnes and Egge, 1991). The affinity based parameterisation may also be applied to describe nutrient-limited growth, assuming that the rates of nutrient uptake and growth are balanced. In this case both parameters ($K_s$ and $\alpha_{\text{aff}}$) can be interpreted as being interrelated $\alpha_{\text{aff}} = \mu_m \cdot f_T / K_s$. However, $\alpha_{\text{aff}}$ is derived from mechanistic considerations that are fundamentally different from former interpretations of $K_s$ of a Monod equation (Pahlow, 2005; Armstrong, 2008; Pahlow and Oschlies, 2013; Fiksen et al., 2013). For comparison between estimates of $\alpha_{\text{aff}}$ it is important to know whether this parameter describes limitation of growth or of nutrient uptake. The description of nutrient limited growth with the Monod equa-

tion, thereby retrieving values for $K_s$ from measurements, had been discussed in the past (e.g., Eppley et al., 1969; Falkowski, 1975; Burmaster, 1979; Droop, 1983). This discussion regained attention during recent years and the sole application of the Monod equation is currently viewed as a considerable drawback when simulating plankton growth under transient (unbalanced growth) conditions (Flynn, 2003; Smith et al., 2009; Franks, 2009; Smith et al., 2014, 2015).

### 3.5   Algal growth and intracellular acclimation

More complex growth dependencies are described with models that consider intracellular acclimation dynamics (e.g., Geider et al., 1998; Pahlow, 2005; Armstrong, 2008; Wirtz and Pahlow, 2010). In these models, photoautotrophic growth rates become dependent on cell quota, e.g. usually normalised to carbon biomass (N:C), and the amount of synthesised Chl$a$ per cell. With such approaches, the changes of the mass distribution of phytoplankton C and N, as well as the cellular Chl$a$ content, have to be explicitly resolved in the model. One advantage is that these models are more sensitive to variations in light conditions and nutrient availability. The respective equations involve physiological parameters that are related but not identical to those of classical N- or P-based growth models, which impedes a direct comparison of older estimates of growth parameters with values currently used in models with acclimation processes resolved.

### 3.6   Losses of phytoplankton biomass

Parameterisations of phytoplankton cell losses involve lysis (starvation and/or viral infection), the aggregation of cells together with all other suspended matter, and grazing by zooplankton. Exudation and leakage are processes of organic matter loss that occur while the physiology of the algae is functional. Cell lysis, exudation and leakage are usually expressed as a single rate parameter and this loss of organic matter is assumed to be proportional to the phytoplankton biomass.

Parameterisations of phytoplankton losses due to the process of coagulation and sinking of phytoplankton and detrital aggregates are basically derived from the principle theory of coagulation. The application of coagulation theory to simulate phytoplankton aggregation is well established for models that resolve size classes of particles (of phytoplankton cells and detritus) explicitly (Jackson, 1990). But the representativeness of simplifications (e.g. reduction to two size classes) assumed for model simulations remains an open task (e.g., Ruiz et al., 2002; Burd and Jackson, 2009). Aggregation parameters in marine ecosystem models are often assumed to represent the combination of a collision rate and the probability of two particles sticking together after collision (e.g. stickiness of algal cells). These two parameters, collision rate and stickiness, are multiplied with each other to yield a final aggregation rate. They are therefore difficult to estimate sep-

arately. Unless prior information can be used their estimates are always collinear, which suggests to estimate their product instead (as done in example in Sect. 5.4).

A common problem is to find constraints that allow for a clear distinction between phytoplankton losses due to the export of aggregated cells and the loss because of grazing. Both processes can be responsible for the drawdown of phytoplankton biomass, and data that cover the onset, peak and decline of a bloom are needed for a possible distinction. How the complex nature of predator-prey interaction is parameterised remains a critical element of plankton ecosystem models. Compared to the approaches that describe algal growth an even larger number of different parameterisations exist for grazing (Gentleman et al., 2003). Experimental data of grazing rates and collections of field data of zooplankton abundance are therefore of great value.

Elaborate analyses of meso- and microzooplankton biomass, grazing and mortality rates were done by Buitenhuis et al. (2006, 2010). For their two studies they compiled an extensive database with laboratory and field measurements. With their data syntheses they could derive parameter values for simulations with a global ocean biogeochemical model. Furthermore, independent field data, not used to derive the meso- and microzooplankton parameter values, were considered for assessing the performance of their model on global scale. Their work reflects the large effort that can be dedicated to this topic for achieving reliable simulation results of zooplankton grazing.

The explicit distinction between zooplankton size classes, like meso- and microzooplankton, was bypassed in Pahlow et al. (2008). Their model allows for omnivory within a the zooplankton community, which is resolved by introducing adaptive food preferences. These preferences are treated as trait (property) state variables that adapt to the relative availability of different prey. This reduces the number of parameters needed to describe a variety of different behaviour in grazing responses. Field data from three ocean sites in the North Atlantic were used by Pahlow et al. (2008) for calibrating their plankton model. They conducted a two-step approach for parameter optimisation. First they optimised parameter values so that depths and dates of minimum and maximum observed values become well respresented by their model at all three sites. In a second step they refined their parameter estimates by minimising weighted data-model residuals. After parameter optimisation they identified distinctive complex patterns between zooplankton grazing and plankton composition for the three simulated ocean sites. Besides their phytoplankton grazing losses it turned out that their optimal estimates of photo-acclimation and maximum C-fixation ($\alpha_{\mathrm{phot}}$, $P_m^C$) agree with those values derived from model calibrations with laboratory data.

## 3.7 Constraining simulations of algal growth with laboratory and mesocosm data

Parameter values of acclimation models have typically been adjusted to explain laboratory measurements (Geider et al., 1998; Flynn et al., 2001; Pahlow, 2005; Armstrong, 2006; Smith and Yamanaka, 2007a; Pahlow and Oschlies, 2009; Wirtz and Pahlow, 2010). So far, there is a limited number of experimental studies whose data were used to calibrate these acclimation models (Laws and Bannister, 1980; Terry et al., 1983, 1985; Healey, 1985; Flynn et al., 1994; Anning et al., 2000). Model calibrations were usually done by tuning parameter values so that model solutions provide a qualitative good fit to the laboratory data. In many cases the parameter adjustments relied on the researchers' experience and intuition, sometimes accounting for prior parameter values obtained from preceding model analyses (e.g., Flynn et al., 2001). Analyses of parameter uncertainties of recent acclimation models are often lacking. Most laboratory modelling studies had put emphasis on the physiological mechanistic model behaviour while error assumptions for quantitative data-model comparison were hardly considered.

Explicit error assumptions for parameter optimisations and for comparisons of acclimation model results with laboratory data were introduced by Armstrong (2006) and by Smith and Yamanaka (2007a). In both studies additive uncorrelated Gaussian observational errors were assumed and optimised results of different model versions had been compared. Armstrong (2006) applied a "simulated annealing" algorithm (Metropolis et al., 1953) to fit his optimality-based model version to the data of Laws and Bannister (1980). The same data were used to also fit the model of Geider et al. (1998) and he evaluated the likelihood ratio of the two ML estimates, to discuss and underpin the improved performance of his refined acclimation parameterisations. Smith and Yamanaka (2007a) also compared the performance of two acclimation models, of Geider et al. (1998) and Pahlow (2005) respectively. Optimal parameter values for the two model versions were obtained with the MCMC method, minimising the misfit between model results and data of the Flynn et al. (1994) experiment. Apart from mechanistic considerations, Smith and Yamanaka (2007a) concluded that the models of Pahlow (2005) and Geider et al. (1998) were describing the assimilated data equally well, since both cost function minimum values were comparable. However, the simulated N:C and Chl$a$:N ratios of the model proposed by Pahlow (2005) were in much better agreement with observations during the exponential growth phase, which remained undifferentiated by their error model (assuming C, N and Chl$a$ data to be independent). Different considerations for error models will be addressed hereafter in Sect. (4).

To collect diverse data that fully resolve onset, peak and decline of an algal bloom at ocean sites is difficult to achieve. Data derived from remote sensing, e.g. Chl$a$ concentration and primary production rates, provide limited information to

explain relevant differences between processes described before, like N-utilisation, fixation and release of C, and synthesis and degradation of Chl*a*. Mesocosm experiments that enclose a large volume of a natural plankton and microbial community can be helpful in this respect, if they provide a good temporal resolution of the exponential growth phase as well as of the post-bloom period. Vallino (2000) highlighted the benefits of using mesocosm data to test plankton ecosystem models, as done before by Baretta-Bekker et al. (1994, 1998). One advantage is that mesocosms are, apart from the surface, closed systems and measurements of inorganic nutrients, dissolved and particulate organic matter should, in principle, add up to approximately constant concentrations of total nitrogen and total phosphorus. Total carbon concentrations may only vary due to air-sea gas exchange. By design these experiments often integrate valuable series of joint and parallel measurements, yielding detailed data from various scientist with different expertise (e.g., Williams and Egge, 1998; Riebesell et al., 2008; Guieu et al., 2014). Drawbacks are uncertainties in initial conditions and also the representativeness of mesocosm data to reflect the real dynamics in the ocean is subject to discussion (e.g., Watts and Bigg, 2001). In spite of these limitations, simulations of mesocosms or of enclosures experiments (e.g. with large carboys deployed in the field) have helped to identify credible model parameter values and assess model performance. This is particularly true for tracing microbial dynamics (Van den Meersche et al., 2004; Lignell et al., 2013) or for details in the composition and fate of particulate organic carbon and nitrogen (POC and PON) (Schartau et al., 2007; Joassin et al., 2011).

In contrast to laboratory measurements, data from mesocosm experiments reflect some natural variability of the plankton community, mainly captured by replicate mesocosms. The availability of measurements from replicate mesocosms is also helpful when defining error models that specify the statistical treatment of the data used for parameter estimation.

## 4   Error models

Error models define our assumptions about uncertainties and the statistical relationships between observed data, the true state, model output, model inputs (forcings and initial/boundary conditions), and model parameters. Here we review error models that have been applied to address the various sources of uncertainty in marine ecosystem models and consider their implications for parameter identification. An explicit treatment of each source of uncertainty may not be necessary but we do recommend to reflect on how these uncertainties can be accounted for when modelling plankton dynamics and biogeochemical cycles.

### 4.1   Uncertainty in observations

The simplest and most common models for observational error assume that the observational errors $\epsilon$ are: i) additive normal, ii) constant variance between samples, and iii) independent between samples and variable types. Such models are also commonly used to represent aggregated errors accounting for both observational and kinematic model error (see Sect. 2.1); we will refer to these as *residual* errors.

The additive normal assumption (i) is straightforward but also restricted, as it does not capture three common characteristics of some ecosystem data such as Chl*a* concentrations: 1) larger values tend to have larger errors, 2) values cannot be negative, and 3) the error distribution has positive skew. Characteristic (1) may be captured by scaling the standard error with modelled values (e.g., Hurtt and Armstrong, 1996) or with observed values (e.g., Harmon and Challenor, 1997), while characteristic (2) can be resolved using truncated error distributions (e.g., Hooten et al., 2011). All three characteristics together can be captured by gamma distributions (Dowd, 2007) or power-normal distributions whereby normality is assumed on a power-transformed scale (Freeman and Modarres, 2006). The power-normal family includes lognormal (e.g., Hemmings et al., 2003) and square-root normal models (e.g., Fasham and Evans, 1995).

For power-normal, gamma, or proportional error assumptions we have the difficulty that the variance on the original scale approaches zero at low values. This may be unrealistic, at least in regard to instrumental noise. In normal models this problem can be addressed by adding a constant term to the variance (Schartau et al., 2001; Schartau and Oschlies, 2003) or standard deviation (Vallino, 2000). Another difficulty is that transform-normal models may require unbiasing factors when assuming unbiased errors on the original scale (e.g. $\exp -\tilde{\sigma}^2/2$ for the log-transform). More flexible models may be obtained by e.g. fitting the power transform parameter (Box and Cox, 1964), assuming generalised Gaussian distributions (Tarantola, 1987; Evans, 2003), or using 'anamorphic' transformations (Bertino et al., 2003; Simon and Bertino, 2012). It is yet unclear whether such extra flexibility is generally necessary, but it has been demonstrated that the choice of transformation can strongly affect estimates of plankton ecosystem fluxes (Evans, 2003) and that a good choice can improve parameter estimation in twin experiments (see Fig. 1 and Simon and Bertino, 2012).

The validity of the constant variance assumption (ii) may be improved by a scale transformation, although the transformation that best normalises the error distribution (see above) may not best promote the homogeneity of variance. Spatiotemporal variations in the error variance may naturally occur, for example due to seasonal modulations of the unresolved variability and hence the representativeness error component. Accounting for this variation should improve parameter estimates and uncertainty assessment (cf., Hem-

mings and Challenor, 2012), but in applications this has rarely been attempted (Hemmings et al., 2003; Dowd, 2007).

In some contexts e.g. mesocosms, the error covariance matrix might be estimated from experimental replicates prior to fitting the model (Sect. 5.4). In problems where sampling is sparse and/or when the model error contribution is large, the error variances may not be estimable from data alone (Evans, 2003). Here the variances may instead be parameterised and estimated jointly with the ecosystem model by Bayesian or ML estimation, which has been done in few studies (Hurtt and Armstrong, 1996, 1999; Stock et al., 2005; Malve et al., 2007; Lignell et al., 2013).

The assumption of independent errors between samples and variable types (iii) can be invalidated in cases where contributions from representativeness error or kinematic model error are large, or where the data have been derived by interpolation or application of a regression model. Neglected correlation may result in parameter estimates that are less efficient (higher variance) and more strongly correlated (e.g. see example in Sect. 5.4). Pre-averaging the data is somewhat helpful to promote independence (and normality, via the Central Limit Theorem), but might also remove some of the informative variability. One common *ad hoc* intervention in the cost function is to scale the residual error variance with the sample size of each data type, to avoid biasing the fit in favour of better-sampled variables (e.g., Schartau and Oschlies, 2003; Friedrichs et al., 2007). More formal treatments have fitted parameterisations of the error correlations jointly with the ecosystem model (e.g., Stock et al., 2005; Arhonditsis et al., 2008).

Whatever the assumptions of the observational/residual error model, it is possible to test their validity using the assimilated data, either by analysing the residuals and performing lack-of-fit tests (Bennett, 2002, p43; Stock et al., 2005; Wallhead et al., 2014) or by comparing fit statistics with those obtained under alternative error models (using e.g. likelihood ratio tests, information theoretic or Bayesian criteria, see Sect. (6.2).

Finally, we caution that certain interpolated or derived data may strictly invalidate the observational error model, not only due to error correlation (see above), but also due to the introduction of *smoothing bias*. Data interpolated onto a model grid will tend to systematically underestimate true values where they are high and overestimate them where low; an effect that will be difficult to account for in the observational error model. In this situation parameter estimates can become biased towards values that suppress spatiotemporal variability in plankton dynamics. Similarly, if the data are derived from a regression model, these estimates may also "trim the peaks and fill the valleys", because in a regression model (e.g. $y = a_0 + a_1 p + \epsilon$, where $p$ is some predictor data) there is always some part of the true variability that is included in the error term, and therefore subject to smoothing bias. In principle this could be avoided by including an inverted regression relationship in the operator $O$ and assimilating the "raw" predictor or proxy data instead of the regression-based estimates.

## 4.2 Prior uncertainty in $\Theta$

Prior uncertainty plays an important role in estimating model parameters. Typically, there is not enough information in the assimilated data to constrain all parameters of a biogeochemical model. The results may well be sensitive to the "error model of prior uncertainty". Prior uncertainty can be represented by prior probability densities in Bayesian approaches or plausible ranges in non-Bayesian approaches. To account for nonnegativity constraints, prior distributions typically include lognormal (Parslow et al., 2013), square-root normal (Gunson et al., 1999), or beta distributions (Dowd and Meyer, 2003), although normal distributions may yet be applicable for parameters that are well constrained above zero (Parslow et al., 2013). To our knowledge no application has yet incorporated prior correlations between parameters in $\Theta$ (i.e. off-diagonal terms in matrix $\mathbf{B}$ introduced in Sect. 2.3). This is surprising, given the fact that posterior uncertainty assessments consistently reveal strong correlations (e.g., Matear, 1995; Prunet et al., 1996; Fennel et al., 2001; Faugeras et al., 2003; Kreus and Schartau, 2015).

Quantifying the prior uncertainty in $\Theta$ is often difficult due to: 1) the existing diversity of model structure, functional forms used in the various parameterisations, and definitions of model state variables, and 2) the intrinsic variability between assimilated data sets in terms of taxonomic composition of the plankton community vs. (usually monospecific) laboratory cultures. As a result, it may not be advantageous to simply set the prior uncertainty in $\Theta_l$ as the posterior uncertainty from one previous study. A more common approach is to first gather best estimates of $\Theta_l$ from a series of previous studies that included parameterisations and state variable definitions sufficiently consistent with the present, and then treat these as unbiased data from which a prior distribution or plausible range can be determined.

When posterior uncertainty becomes unacceptably high, it can be reduced by reducing the prior uncertainty in $\Theta$, and there are several strategies for doing this. First, we should incorporate further data, perhaps of a qualitative nature, into the prior constraints. For example, if it is known *a priori* that certain species or functional groups coexist in certain regions at certain times of the year, then any $\Theta$ resulting in competitive exclusion of one of these groups might be ruled out *a priori*. Another possibility within the Bayesian paradigm is to incorporate the subjective opinion of experts (O'Hagan, 2006). A second strategy is to model statistical structure in the prior parameter values, and thereby fill in missing prior parameter estimates for certain species included in the modelled species or groups. Examples here include the use of allometric scaling relationships with cell size (e.g., Edwards et al., 2012) and phylogenetic relationships derived from stochastic modelling of trait evolution (Bruggeman et al.,

2009; Bruggeman, 2011). Third, we may seek to reduce the model complexity in terms of the number of free parameters, thereby removing poorly-constrained parameters and parameter correlations that may act to inflate the posterior uncertainty. This may be achieved using sensitivity analysis (e.g., Friedrichs, 2001; Garcia-Gorriz et al., 2003; Hemmings et al., 2003) or model selection criteria (e.g., Ward et al., 2013). A risk here is that parameter estimates and uncertainty assessment may be compromised if model selection uncertainty is not properly accounted for (Burnham and Anderson, 2002). Fourth, it may be possible to reformulate the model in such a way that the prior parameter uncertainty is reduced. For example, a hierarchical model in which parameters vary randomly over space (Zhang and Arhonditsis, 2009) or time (Parslow et al., 2013) may enable the use of stronger prior constraints on the distributional parameters describing this variability (i.e. the 'hyperparameters'). Similarly, a stochastic trait-based approach (e.g., Follows et al., 2007) may employ distributional parameter values that are better known a priori than values for individual species or functional groups, although such a reduction in prior uncertainty has not yet been clearly demonstrated in the literature.

### 4.3  Uncertainty in initial conditions (ICs)

Dynamical marine ecosystem models are usually specified by differential equations that are first-order in time, and therefore require for solution one initial condition (IC) for each grid cell or spatial location in the model. These inputs are, in general, uncertain, and liable to impact the model output, at least during a transient relaxation period, or indefinitely if the uncertainty spans more than one basin of attraction of the dynamical system or if the model dynamics are chaotic (e.g., Huisman and Weissing, 1999).

In some cases it is possible to neglect IC error because of accurate measurements, or because a steady state (equilibrium or seasonal cycle) that is only sensitive to $\Theta$ can be assumed. Caution is required if neglecting IC uncertainty because initial concentrations are known to be small (e.g. in January); small absolute errors may be large relative errors that can still affect e.g. timing and magnitude of a spring bloom (Evans and Parslow, 1985).

In non-spatial (0D) models, IC errors have been modelled as both fixed parameters (e.g., Vallino, 2000) and as random variables (Bayesian parameters) with specified prior distributions (e.g., Arhonditsis et al., 2008). In mesocosm studies, ICs can play a critical role in determining the model trajectory, and can comprise a large proportion of the fitted parameters (e.g., Lignell et al., 2013). For spatial models, it seems necessary to limit the degrees of freedom of the IC uncertainty (Li et al., 2006), e.g. by using a Bayesian error model with spatial covariance in the prior (Smith et al., 2009; Pelc et al., 2012). To model IC uncertainty, Gaussian distributions are most often employed, often with a log transform to improve realism of the distributional form (see Sect. 4.1). For

systems with strong physical control, it may be possible to limit IC uncertainty to only the physical variables, allowing this to generate biochemical uncertainty over an initial burn-in period (Natvik and Evensen, 2003; Simon et al., 2015).

### 4.4  Uncertainty in forcings and boundary conditions (BCs)

Marine ecosystem models are usually modulated by time and space-dependent environmental drivers (forcings) and boundary conditions that are not predicted by the model dynamics but are necessary inputs to determine the evolution of the model state variables. Studies have demonstrated the sensitivity of biogeochemical variables to errors in bottom-up forcings such as wind stress and vertical mixing (e.g Evans, 1988; Friedrichs et al., 2006; Béal et al., 2010; Sinha et al., 2010) and top-down forcings such as fishing (e.g., Heath, 2012). BC errors may have little impact on variables strongly controlled by internal dynamics at sufficient distance from the boundaries, but they may become critical if they affect internal system constraints such as the supply of limiting nutrients or fluxes of heat/salinity that drive internal circulation and stratification.

There are basically two approaches to modelling the effects of BC/forcing error: 1) to consider individual or net impacts on model dynamics as dynamical model errors ($\eta$ in Eq. 1), thus requiring a stochastic model, or 2) to consider the net impacts on state variables as kinematic model errors ($\zeta$ in Eq. 2), which may permit a deterministic model. The dynamical approach (Eq. 1) is arguably more realistic, more likely to generate realistic temporal correlations and cross-correlations, and accounts for time and parameter-dependent variation in the form and correlation structure of the joint state variable probability density. It also allows individual error sources to be considered separately. However, approaches based on stochastic models can be computationally intensive and methodologically complex, and parameterising all individual sources of BC/forcing error poses a major challenge. Rather than attempting a comprehensive treatment, current approaches tend to restrict the dynamical noise to certain key sources such as the atmospheric forcing (Natvik and Evensen, 2003; Simon and Bertino, 2009) or surface irradiance and background light attenuation (Torres et al., 2006; Ciavatta et al., 2011), and/or they model the net effect of BC/forcing errors and structural errors synthetically as additive (e.g., Losa et al., 2003, 2004) or multiplicative (e.g., Dowd and Meyer, 2003; Weir et al., 2013) perturbations. It may be questioned to what extent the simple parameterisations used to describe these noise processes accurately describe the net or individual error sources, and it can be difficult to constrain the distributional parameters *a priori*, especially if the structural component is important. Hierarchical filtering methods may allow these "hyperparameters" to be estimated jointly with the other parameters (Jones et al.,

2010) but these may incur a computational cost that is prohibitive for spatial models at present.

The kinematic approach (Eq. 2) offers an immediate computational saving because the integral over model error configurations (over $x^t$ in Eq. 7) can usually be performed analytically, such that accounting for model error may amount to simply adding variance and correlation structure to the observational error covariance matrices. However, this may require a more complex parameterization of the error covariance that may still not properly capture seasonal or ecosystem parameter dependence (Stock et al., 2005; Arhonditsis et al., 2008; Zhang and Arhonditsis, 2009). Hemmings and Challenor (2012) demonstrated a Monte Carlo simulation approach to determine the variability of kinematic error variances due to BC/forcing error, but without accounting for correlations or $\theta_e$-dependence. Note that with a deterministic model, and model error treated kinematically, the ecosystem parameters $\theta_e$ will likely be optimised to reproduce the ensemble-mean or ensemble-median behaviour of the true system. This may be convenient for future simulations, but it may also result in biases when using previous parameter estimates from laboratory experiments or stochastic model data assimilations to constrain the prior uncertainty in $\theta_e$ (see Sect. 4.2).

In either case, BC/forcing error models may fall short in describing potential errors in *phase*, like the timing of nutrient depletion. Model solutions that predict the right sequence of events (e.g. a plankton bloom) but with slightly wrong timing or spatial location, perhaps due to phase error in the atmospheric forcing or ocean circulation, may suffer a double penalty due to changes where none occur in the data and no change where the data do vary. DA may then "smooth out" the model variability in order to minimise this double penalty (Wallhead et al., 2006; Ravela et al., 2007). The problem of phase/timing error has received substantial attention in numerical weather forecasting and geophysical DA (e.g., Hoffman et al., 1995; Lawson and Hansen, 2005; Mittermaier, 2007; Ravela et al., 2007; Ziegeler et al., 2012) and has been highlighted as an issue for marine ecosystem models (Schartau and Oschlies, 2003; Friedrichs et al., 2006). A simple remedy is to average the data and model over larger spatio-temporal scales in the data assimilation (e.g., Schartau and Oschlies, 2003), but again this may remove informative variability and result in a $\widehat{\Theta}$ that is only suited to those larger scales. Wallhead et al. (2006) explored a more explicit approach assuming random time lags between the true state and model state i.e. kinematic model errors in phase, which can be expressed as $\zeta(\theta_\zeta)$ in Eq. (2) (see Appendix A). This may improve the bias and variance of ecosystem parameter estimates compared to a simpler approach assuming only additive residual error (Wallhead et al., 2006, Table A1).

For some problems, in particular for chaotic systems, the phase noise may be too intense or ill-defined to allow effective use of a parametric phase lag model. A better approach here might be to use a 'synthetic likelihood' (Wood, 2010), whereby the raw data and model output are replaced with a carefully chosen, informative set of phase-insensitive summary statistics (e.g. means, standard deviations, and lag correlations; cf., Heath, 2012). This approach could incorporate the comparison of modelled vs. observed Fourier spectra and cross-spectra/coherences (e.g., Powell et al., 2006). Whether the statistics e.g. of spectral slopes by themselves provide good constraint on ecological parameters should be tested since it may not be sufficient (Armi and Flament, 1985; Martin, 2003; Franks, 2005).

## 4.5 Uncertainty in model formulation and structure

Even with perfectly-known parameters, forcings and initial/boundary conditions, we would still not expect the modelled fluxes such as primary productivity and grazing to perfectly reproduce the true fluxes, or the state variables to perfectly follow the true variability. Aggregation of species into model functional groups, effects of finite spatial and temporal resolution, and inherent approximations in the flux parameterisations and model structure may all contribute to "structural error" in the model dynamics.

One promising approach to account for structural error is to add stochastic noise (dynamical model errors) to the ecosystem model parameters $\theta_e$ (see Sect. 7). This preserves mass conservation and may allow information on the temporal (e.g. seasonal) variability of species composition within functional groups to be utilised within the stochastic process parameters (e.g., Parslow et al., 2013). However, as with explicit treatments of BC/forcing error (see Sect. 4.4), a comprehensive treatment of all sources seems likely to result in an overparameterised error model and appears to be not yet attempted. An alternative (or complementary) approach is to treat the structural errors as synthetic dynamical or kinematic model errors, with one noise process for each state variable. Here it seems the challenges are to control mass conservation and to find some efficient way to constrain the distributional parameters *a priori* or *a posteriori*.

We note that some structural errors may impose persistent or intermittent biases in the model output that may not be amenable to a simple statistical description. For example, a succession in blooming phytoplankton species might extend or multiply the bloom periods in ways that are not "random" and that are difficult to reconcile with a single model functional group, even with stochastic parameters. Limited spatial resolution can also impose persistent biases that lead to poor extrapolation properties when we try to correct them by adjusting $\theta_e$ (Wallhead et al., 2013). In such cases, rather than elaborating the error models, effort might be better spent improving the explicit biological or spatial resolution of the model, or exploring implicit resolution techniques (e.g., Wirtz and Eckhardt, 1996; Merico et al., 2009; Wallhead et al., 2013).

An alternative approach might be to employ the tools of multimodel inference (Burnham and Anderson, 2002; Link

and Barker, 2006). The idea here is to base inference of target parameters, states, and fluxes on a family of candidate models, each differing in structure and parameterisation, rather than on a single model. For example, we might be fairly certain about the form of the photosynthesis-irradiance (P-I) function in phytoplankton, but much less certain about the appropriate formulation of zooplankton grazing. Multi-model inference would allow the P-I parameter values and their uncertainties to be inferred on the basis of several candidate models, each assuming the same P-I function but different grazing parameterisations. The resulting multimodel estimates and uncertainties would be less likely to be biased by a poor choice of grazing formulation than the inference premised on a single *a priori* formulation.

## 5    Posterior parameter uncertainties

The determination of parameter uncertainties has many facets, getting to the core of discussions of Bayesian and frequentist approaches and interpretations (e.g., Efron, 1986; Cox, 2005; Lele and Dennis, 2009). Depending on the estimator, uncertainties in the combination of parameter values may either disclose a credible region of a random distribution of parameter values (Bayesian interpretation) or they mark a confidence region that should include the true value with a certain nominal probability of e.g. 95% (frequentist interpretation). The latter means that different data sets would yield different confidence regions and e.g. 95% of those regions are expected to include the true "fixed" value.

In general, if we wish to make inference about uncertainties of parameter estimates ($\widehat{\Theta}$) we need some knowledge about the distributional shape of the posterior $p(\widehat{\Theta} \mid \boldsymbol{y})$ or of the likelihood $p(\boldsymbol{y} \mid \widehat{\Theta})$. Likewise, we can gather information about the parameter-cost function manifold in the vicinity of $(\widehat{\Theta}, J(\widehat{\Theta}))$. For this we may consider some threshold offset value $\Delta_J$, which is an upper limit for the deviation from the minimum value $J(\widehat{\Theta})$. Such a limit may identify all cost function values that are insignificantly larger than $J(\widehat{\Theta})$. Large deviations from optimal estimates might be required for some parameters (components of $\widehat{\Theta}$) before the corresponding cost function values reach this threshold, while for other components only small variations are enough. Such tolerance limit defines an uncertainty region in parameter space:

$$\left\{ \Theta : J(\Theta) - J(\widehat{\Theta}) \leq \Delta_J \right\} \tag{16}$$

Typical threshold values are defined as the $\alpha$ quantile of a parametric or nonparametric probability distribution.

For an unbiased ML estimator, the $\chi^2$-distribution with the degree of freedom (df=$N_y - N_\Theta$) has been suggested for deriving a threshold value $\chi^2(\text{df}, \alpha)$ (e.g., Kuczera, 1990; Meeker and Escobar, 1995; Raue et al., 2009, 2011). But for nonlinear models the $\chi^2$-distribution might be inappropriate

and the $\alpha$ quantile of the actual distribution, $J(\Theta) - J(\widehat{\Theta})$, needs to be evaluated by other means (e.g., Raue et al., 2011). Furthermore, the degree of freedom (df) that specifies location and shape of the $\chi^2$-distribution may not be representative. Only if error correlations have been correctly specified in $J$ (see Sect. 4) and the asymptotic approximation (for large $N_y$) is applicable, then the correct degree of freedom is $N_y - N_\Theta$. The effective number of independent observations can be lower and the considered error correlations can be imprecise, for example when measurements like Chl$a$ and carbon dioxide concentrations are negatively correlated during exponential growth but can then become positively correlated shortly after the peak of an algal bloom. We therefore expect the effective degree of freedom to be often lower than $(N_y - N_\Theta)$ and $\chi^2(\text{df}, \alpha)$ would therefore be an optimistic threshold, i.e. likely to underestimate the true range of uncertainty, unless the correct number of degrees of freedom is determined.

### 5.1    Confidence and credible regions

Uncertainty regions in parameter space can be determined basically in two different ways, either based on a Bayesian- or frequentist interpretations. According to the Bayesian interpretation a credible region is specified by conditional probability distribution of the true value given the data. For maximisations of the likelihood $p(\boldsymbol{y} \mid \Theta)$ it is often stated that credible and confidence regions are practically identical. Such interpretation is imprecise since the methods to confine either regions can be very different with respect to the underlying assumptions, e.g. MCMC versus bootstrap approaches.

In case of classical BEs no tolerance limit $\Delta_J$ is explicitly prescribed. Instead, an efficient sampling of $(\Theta, J(\Theta))$, or directly of the posterior $p(\Theta \mid \boldsymbol{y})$, is applied. Sequential methods can provide approximations of the posterior parameter distribution once all data have been assimilated. These approximations differ, depending on how Eqs. (6) and (7) are sampled and evaluated, as discussed in Sect. (2.2.2). A helpful overview with some comprehensible examples (of four different methods and three different ensemble sizes) is given by Weir et al. (2013). BE methods that do not rely on sequential approaches may also be applied and credible regions are then simply inferred from selective (acceptance/rejection) sampling schemes in a MCMC approach, e.g. Metropolis-Hastings algorithm (Metropolis et al., 1953; Hastings, 1970). MCMC methods for the derivation of credible regions are also used for ML estimation problems (e.g., Smith and Yamanaka, 2007a). The main point is that here the data are assumed fixed.

A fundamentally different approach to the BE methods is to repeat parameter optimisations many times but with data subsamples or resample data sets. Large data sets are split up into a series of subsamples that should be as independent as possible. Or many synthetic data sets are created by applying a random number generator to independently draw bootstrap

samples (Efron, 1985; Efron and Tibshirani, 1986). This approach accounts for variable data and it mimics a repetition of an experiment or a repeated sampling at ocean sites. For each bootstrap data set ($\boldsymbol{y}^*$) a corresponding optimum estimate $\widehat{\Theta}^*$ is obtained. A distribution of $\Delta_\Theta = \overline{\Theta^*} - \widehat{\Theta}^*$ can be derived from a series of optimisations with different bootstrap data sets. Furthermore, nonparametric density estimates of all $J(\widehat{\Theta}^*)$ can be derived and the $\alpha$ quantile can then be determined from the cumulative distribution of such probability density. For some situations a bootstrap approach with as few as ten resample data sets may suffice to highlight specific uncertainties in some model parameters (e.g., Schartau et al., 2007). But to ascertain confidence regions, much larger bootstrap sample sizes are typically needed (Efron and Tibshirani, 1986). In the end, both approaches, MCMC and bootstrap methods, require a large number of model evaluations, typically o($10^2$) - o($10^4$). The benefit is that skewed and contorted posteriors can be better resolved.

## 5.2 Profile likelihoods

An alternative to ensemble-based sequential, MCMC, and bootstrap methods for determining uncertainties of parameter estimates is the construction of 1D- or 2D profile likelihoods (Venzon and Moolgavkar, 1988). For a 2D profile likelihood an array of combinations of two parameters $(\Theta_m, \Theta_n)$ is constructed. For every combination of parameter values (elements of the 2D array) a minimisation of $J(\Theta)$ is repeated while varying all other parameters $(\Theta_{l \neq m,n})$. This is done for all arrays with possible combinations of two parameters, which requires a large number of additional optimisations. The advantage is that uncertainty intervals $[\widehat{\Theta}_l - \boldsymbol{u}_l^-, \widehat{\Theta}_l + \boldsymbol{u}_l^+])$ can be well resolved for each component ($l$) of $\Theta$, with lower and upper uncertainty limits possibly being different $(\boldsymbol{u}_l^- \neq \boldsymbol{u}_l^+)$. Unfortunately, the evaluation of a profile likelihood is impracticable for most marine ecosystem model applications, because of the associated computational costs. Parameter identifiability analyses based on profile likelihoods have been applied to problems where fast evaluations of $J(\Theta)$ were possible (e.g., Brun et al., 2001; Raue et al., 2009, 2011). Brun et al. (2001) evaluated confidence regions for three parameters (rate constants of production, respiration and water-air gas exchange) from profile likelihoods and they showed that the error margins of the parameter estimates can be much larger than those derived with e.g. a point-wise approximation of a posterior uncertainty covariance matrix, described in the following.

## 5.3 Point-wise approximations of posterior uncertainty covariance matrix

A single point in parameter space is identified by ML and MAP estimators, i.e $\widehat{\Theta}$ where the posterior $p(\Theta \,|\, \boldsymbol{y})$ has its maximum. Because of the computational costs we often find studies where parameter uncertainties of ecosystem models

had been approximated point-wise in the immediate vicinity of $\widehat{\Theta}$. A common theory for deriving variance information of a ML estimate is based on the inverse of the Fisher information (Fisher, 1922; see also e.g., Fisher, 1934; Efron and Hinkley, 1978; Cao and Spall, 2010). The underlying assumption is that the likelihood $p(\boldsymbol{y} \,|\, \widehat{\Theta})$ is nearly normal shaped nearby its maximum, which is tantamount to a quadratic increase of $J(\Theta)$ as parameter values are varied around the estimate. Series expansions, like Taylor power series, around the estimate $\widehat{\Theta}$ can be applied to derive relevant properties of $J(\Theta)$ that are theoretically attributed to an uncertainty covariance matrix ($\mathbf{U}_\Theta$). Confidence regions for $\widehat{\Theta}$ can then be expressed in terms of approximations of $\mathbf{U}_\Theta$. For example, for some prescribed df an upper critical confidence level can be specified by the $\alpha$ quantile of a F-distribution (Marsili-Libelli et al., 2003):

$$\left\{ \Theta : \left(\Theta - \widehat{\Theta}\right)^T \mathbf{U}_\Theta^{-1} \left(\Theta - \widehat{\Theta}\right) \leq N_\Theta \cdot F_{df}^{1-\alpha} \right\} \qquad (17)$$

Confidence ellipsoids are described with Eq. (17), thus yielding symmetric uncertainty limits around $\widehat{\Theta}$, i.e. $\boldsymbol{u}_l = \boldsymbol{u}_l^- = \boldsymbol{u}_l^+$. With an approximation of $\mathbf{U}_\Theta$ a confidence interval for every single parameter can be described as $[\widehat{\Theta}_l \pm \boldsymbol{u}_l]$. The individual uncertainty limits can be computed as

$$\boldsymbol{u}_l = t_{\mathrm{df}}^{1-\alpha/2} \sqrt{\mathbf{U}_{\Theta_{ll}}}. \qquad (18)$$

where $t_{\mathrm{df}}^{1-\alpha/2}$ is the two-tails Student's t-distribution for prescribed $\alpha$ and df (Marsili-Libelli et al., 2003). Two approaches to point-wise approximations of $\mathbf{U}_\Theta$ are found in ecological and ecosystem modelling studies. One approach uses first derivates of the model's observation vector with respect to the parameters (Jacobian) whereas the other requires calculations of second derivatives of $J(\Theta)$ (Hessian).

### 5.3.1 Uncertainty covariances based on the Jacobian matrix

A first approach considers a linearisation (first order power expansion) of the model's observation vector $H(\boldsymbol{x})$ around the point estimate $\widehat{\Theta}$. As long as $H\left(\boldsymbol{x}(\widehat{\Theta})\right)$ is not subject to strong nonlinearities, its first derivatives (sensitivity) with respect to $\Theta$ can be used to estimate $\mathbf{U}_\Theta$. For an unbiased ML estimator the covariance matrix can be approximated as:

$$\mathbf{U}_\Theta = \frac{J(\Theta)}{\mathrm{df}} \cdot \left(\mathbf{H}_\Theta^T \mathbf{R}^{-1} \mathbf{H}_\Theta\right)^{-1} \qquad (19)$$

with the Jacobian matrix $\mathbf{H}_\Theta(\widehat{\Theta})$, its transpose ($\mathbf{H}_\Theta^T$), and with the observational error covariance matrix $\mathbf{R}$ (e.g., Thacker, 1989; Kuczera, 1990; Omlin and Reichert, 1999; Brun et al., 2001; Omlin et al., 2001). The term $J(\Theta)/\mathrm{df}$ is added as an approximation of the residual variance of $J$, which should be considered unless $H(\boldsymbol{x})$ is in such good

agreement with data so that the minimum of $J(\Theta)$ actually matches the exact degree of freedom, df. The rows of the Jacobian $\mathbf{H}_\Theta$ are the first derivatives with respect to the parameters $\nabla H(\boldsymbol{x})$, with $\nabla = (\partial/\partial\Theta_1, \partial/\partial\Theta_2, \ldots, \partial/\partial\Theta_{N_\Theta})$ being the Napla operator of first partial derivatives.

### 5.3.2 Uncertainty covariances based on the Hessian matrix

Another more common approach for a point-wise approximation of $\mathbf{U}_\Theta$ is derived from a Taylor expansion around $J(\widehat{\Theta})$. Since $\nabla J(\widehat{\Theta}) \approx 0$ in the minimum, the first order term of the Taylor expansion is negligible. The series expansion then approximates the distribution:

$$J(\Theta) - J(\widehat{\Theta}) \approx \frac{1}{2}\left(\Theta - \widehat{\Theta}\right)^T \boldsymbol{\mathcal{H}}_\Theta \left(\Theta - \widehat{\Theta}\right) \tag{20}$$

The matrix $\boldsymbol{\mathcal{H}}_\Theta$ is the Hessian whose elements are second derivatives of $J(\Theta)$ with respect to the parameters (e.g., Tziperman and Thacker, 1989; Matear, 1995):

$$\boldsymbol{\mathcal{H}}_\Theta = \nabla^T \nabla J(\Theta)\Big|_{\Theta=\widehat{\Theta}} \tag{21}$$

With the Taylor expansion in Eq. (20) we obtain an approximation of the local curvature of $J(\Theta)$ at point $\widehat{\Theta}$, also referred to as the *observed* Fisher information. Like in Eq. (19), but instead of using first derivatives of $H(\boldsymbol{x})$, a posterior uncertainty covariance of $\widehat{\Theta}$ is then approximated by computing the inverse of a Hessian matrix:

$$\mathbf{U}_\Theta = \frac{J(\Theta)}{\mathrm{df}} \cdot 2 \cdot \boldsymbol{\mathcal{H}}_\Theta^{-1} \tag{22}$$

Both approximations (Eqs. 19 and 22) yield, in principle, similar results for accurate ML estimates i.e. when the actual minimum of $J(\Theta)$ has been identified by the optimisation algorithm. In practice search algorithms can terminate at some distance from the actual minimum for numerical reasons, e.g. when the minimum is located in a flat valley of $J$ and the imposed convergence criterion makes an algorithm terminate the search in the periphery of the valley. Marsili-Libelli et al. (2003) proposed an approach where the accuracy of parameter estimates can be improved by minimising differences between the results of Eq. (19) and Eq. (22).

### 5.3.3 The Hessian: its approximation and inversion

Hessian matrices have often been approximated with a finite central differences approach for first and second derivatives of $J$ with respect to ecosystem model parameters at the point-estimate $\widehat{\Theta}$ (e.g., Matear, 1995; Kidston et al., 2011; Kreus and Schartau, 2015). A critical issue of finite difference calculations of the Hessian's elements is the choice of an appropriate increment size ($\boldsymbol{\delta}$), which sets the distance of departure from the optimal parameter point estimate $\widehat{\Theta}$. Sometimes a compromise between resolving flat regions around

$(\widehat{\Theta} + \boldsymbol{\delta}, J(\widehat{\Theta} + \boldsymbol{\delta}))$ and numerical precision has to be found (Kreus and Schartau, 2015). To approach a high accuracy of the Hessian approximation it is possible to consider a set of different increment sizes for the central differences approach, as given in Marsili-Libelli et al. (2003).

The problem of increment size reduces if first derivatives of $J$ with respect to the parameters (gradient, $\nabla J$) are readily obtained with an adjoint model, e.g. as used in a variational DA approach (Sect. 2.2.3). Adjoint versions of plankton ecosystem models have been constructed primarily to compute $\nabla J$ for an efficient search with gradient descent algorithms in the parameter-cost function manifold (e.g., Lawson et al., 1996; Fennel et al., 2001; Schartau et al., 2001; Spitz et al., 2001; Friedrichs, 2002; Faugeras et al., 2003; Zhao et al., 2005; Friedrichs et al., 2007; Xiao and Friedrichs, 2014a). To elucidate the nature of adjoint model developments is beyond the scope of this paper, but a brief summary about adjoint model developments is given in the Appendix (C). The advantage is that all elements of the Hessian can be approximated with finite differences of adjoint model results (e.g., Fennel et al., 2001; Friedrichs, 2002; Faugeras et al., 2003; Friedrichs et al., 2007; Kreus and Schartau, 2015).

Computations of the Hessian, Eq. (21), provide valuable identifiability information even if this matrix is not explicitly used to specify confidence regions of parameter estimates. For example, a decomposition of the Hessian matrix into its eigenvalues and the corresponding eigenvectors reveals which parameters are weakly constrained by the data or it helps to identify structural deficiencies of a model. The eigenvectors' components ($l$) represent the components of $\Theta$. Components of those eigenvectors that belong to small eigenvalues indicate parameter combinations that are poorly constrained or cannot be estimated. In contrast, those eigenvectors that correspond with the largest eigenvalues show parameter combinations that are well constrained. The studies of Fennel et al. (2001) and Faugeras et al. (2003) are informative in this respect, because they provide insight into the range of characteristic eigenvalues and eigenvectors of 0D and 1D marine ecosystem models.

Ideally, every eigenvector would exhibit only one single component, meaning that values of every parameter can be estimated independently of the other parameters' values. In practice this is only the case for few parameters of a planktonic ecosystem model. Eigenvectors with two or more distinct components disclose those parameters whose estimated values are correlated and for which correlation coefficients can be explicitly derived (e.g., Matear, 1995; Prunet et al., 1996). Correlations between parameter estimates are referred to as collinearities. A useful collinearity index was introduced by Brun et al. (2001). Their index expresses how a change in $J$ (or in $H(\boldsymbol{x})$), due to a shift in the value of one parameter can be entirely compensated by adjusting the value of another (correlated) parameter.

## 5.4 Parameter collinearities: an example with phytoplankton loss parameters

In Sect. (3.6) we discussed the difficulty of constraining parameters that determine loss rates of phytoplankton biomass due to grazing, aggregation or exudation and leakage or organic matter. With an example we illustrate typical uncertainties and collinearities in the estimation of phytoplankton loss parameters in the absence of explicit zooplankton observations like micro- and mesozooplankton abundance or grazing rates. Three parameters that affect the loss of phytoplankton biomass have been optimised together with other parameters. For this we assimilated five different types of daily mean observations of a mesocosm study (Engel et al., 2005; Delille et al., 2005) into a plankton ecosystem model with optimal nutrient allocation and photo-acclimation (Pahlow, 2005), as mentioned in Sect. (3.5).

Details of the cost functions and the corresponding mapping from model results $x$ to observations $H(x)$ are given in the Appendix (B). In our example we consider two cost functions, with and without covariances respectively (Eqs. B4 and B5). For both cost functions no prior information is included. As an error model we assume additive Gaussian errors, applying Eq. (4) in Sect. (2.1.3). A simulated annealing (SA) algorithm is first used to identify a best parameter estimate in the vicinity of the global cost function minimum. This point estimate is then used to derive error ellipses (confidence regions) according Eq. (22). These point-wise approximations of parameter uncertainties are finally incorporated to initialise the MCMC method that derives a credible region of posterior parameter uncertainties, based on an algorithm provided by Soetaert and Petzoldt (2010).

Figure (2) shows contours of $J(\widehat{\Theta}_m \pm \Delta_m, \widehat{\Theta}_n \pm \Delta_n; m, n = 1, 2, 3)$ around the optimum at $(\widehat{\Theta}_m, \widehat{\Theta}_n, \min(J))$, while all other parameters are fixed to their optimal estimates $(\widehat{\Theta}_{l \neq m,n})$. Each plot is thus a combination of two loss parameters: maximum grazing $(\Theta_1 = g_m)$ and carbon loss rate $(\Theta_2 = \gamma_C)$ on top (1a/b in Fig. 2); $\gamma_C$ and aggregation parameter $(\Theta_3 = \Phi_{\text{agg}})$ in the middle (2a/b); $\Phi_{\text{agg}}$ and $g_m$ on the bottom (3a/b). Results from MCMC (dots and asterisks) reveal similar collinearities between parameter combinations that involve $g_m$ for the two cost functions (1a/b and 3a/b in Fig. (2). It means that $g_m$ can only be estimated in combination with $\Phi_{\text{agg}}$ and $\gamma_C$. Only if $\Phi_{\text{agg}}$ and $\gamma_C$ were known, then $g_m$ could be identified in this mesocosm model setup with these available data types. We do not find such strong collinearity expressed between $\gamma_C$ and $\Phi_{\text{agg}}$ and their estimates seem to be rather independent (2a/b of Fig. 2), given the mesocosm data.

Another peculiarity is that the ranges of the MCMC's posterior indicate larger uncertainties if the cost function without covariance information is applied (right side of Fig. 2), although model and data are identical. This behaviour is also resolved by the 95% confidence regions that are obtained with a point-wise approximation of error ellipses (lines). Furthermore, collinearities according to the error ellipses are smaller for the cost function with covariances compared to the case of independent data. Here, confidence regions of the error ellipses correspond well with the credible regions of the MCMC results. We stress that this may not be the general case and the good correspondence is likely attributable to the low dimension of the example looked at.

Overall, these results exemplify the uncertainty in constraining major loss parameters in the presence of grazing, if no explicit prior information about grazing rates or data of zooplankton biomass are available. Collinearities between grazing parameters and other phytoplankton biomass losses may be reduced by testing model performance against independent data, e.g. as done for the meso- and microzooplankton grazing in Buitenhuis et al. (2010). In cross-validation studies some combinations of parameters that produce indistinguishable solutions for one experiment or for one ocean site are compared with data of another experiment or at another ocean site, which will be addressed in the following Sect. (6).

## 6 Cross-validation and model complexity

Good performance should be attributable to a model capturing the predominant plankton dynamics under varying conditions in different environments. Parameter values are often optimised for local ocean sites, but ideally, parameter estimates from one site should improve model performance at other locations as well. The generality of optimised models can be tested by cross-validating against independent data, providing a direct and effective test of predictive skill (Gregg et al., 2009).

### 6.1 Cross-validation

Parameter optimisations can often improve the fit of a model by selecting unrepresentative parameter values that serve only to compensate for misfits between data and model results. It is therefore essential to check whether the resultant 'optimised' model is giving the right answer for the correct reasons.

Xiao and Friedrichs (2014b), for example, found that while the optimisation of a range of NPZD models to satellite data tended to reduce model-data misfit, this was often achieved through the adoption of extremely unrealistic parameter estimates, sometimes being multiple orders of magnitude higher or lower than their best a priori estimates. The same authors (Xiao and Friedrichs, 2014a) showed that adding synthetic noise to assimilated satellite data led to the introduction of similar errors, and a significant deterioration of one model's predictive skill. The extreme parameter estimates were not representative for the system and the model performance turned out to be poor when the model was tested

against independent data that were not used during the optimisation procedure.

This is the principle of cross-validation, in which an optimised model is tested in terms of its ability to reproduce data that were not included in the calibration phase. This is often achieved by excluding a subset of the original calibration dataset, for later use in model evaluation. For example, in a variational data assimilation exercise for the Arabian Sea, Friedrichs et al. (2006) repeated their optimisation a number of times, each time excluding data from a particular season. The calibrated models were then used to predict the system behaviour during the withheld season, with the resultant model-data misfit labelled the 'predictive cost function'.

The cross-validation approach has the advantage of testing one of the key attributes of marine biogeochemical models, namely their predictive skill. The technique is, however, not without its difficulties. The first issue is that it is important to ensure the test data are truly independent from the training data. In this regard, Friedrichs et al. (2006) took advantage of the highly seasonal nature of the Arabian Sea, but it would perhaps be less appropriate in regions with a less pronounced seasonal cycle, such as at the centre of a subtropical gyre. A potentially more serious problem occurs when researchers simply divide the available data at random, such that highly correlated data appear in the assimilated and the test data. Under such circumstances, the cross-validation would give no indication as to the ability of the model to predict independent data.

The potential to select unrealistic, compensatory, parameter values may not always be obvious, especially if good estimates of the 'true' (or at least sensible) values of the model parameters are not well known a priori. Such errors may, nonetheless, strongly impact the ability of a model to reproduce anything but the assimilated data. This issue appears to be a common theme in simple marine biogeochemical models calibrated to time-series data, as a number of studies (Fennel et al., 2001; Friedrichs et al., 2006; Ward et al., 2010) have found that parameter optimisation resulted in decreased predictive skill, relative to 'off-the-peg', prior parameterisations. A notable counterpoint to those studies is given by Oschlies and Schartau (2005), who found that simultaneous optimisation of an NPZD model at three time-series sites (Schartau and Oschlies, 2003) led to improved performance when the model was applied within a 3D simulation of the North Atlantic. On the one hand, it seems likely that this improvement was dependent on assimilating data from three highly dissimilar North Atlantic locations, which prevented the inclusion of compensatory errors that were highly specific to any one site (see also Xiao and Friedrichs, 2014a). On the other hand, in Schartau and Oschlies (2003) and in Oschlies and Schartau (2005) it is also stressed that the apparent improvement is associated with some ambiguous rapid nitrogen remineralisation pathway in their simple NPZD model, which can be incorrect in either simulations (1D and 3D), but

with the same positive effect on primary production rates in the central North Atlantic.

## 6.2  Model performance as a function of model complexity

Of the many factors that affect the ability of a biogeochemical model to reproduce and predict observations, the appropriate degree of model complexity in any given situation is both one of the most important, and one of the least well defined. This is because there exists a fundamental trade-off between simplicity and complexity. Simple models have the advantage of being easier to understand, and with fewer parameters they should also be better constrained (both before and after optimisation). Nonetheless, simplification requires a degree of abstraction, and it can sometimes be difficult to draw parallels with the complexities of the observed system.

At the other end of the spectrum, a highly complex model can explicitly resolve more processes, allowing more detailed comparison with observations. As models become more complex, the number of degrees of freedom increases, and the calibrated model will generally be able to match the observations better than a simpler model. If insufficient observations are available, the extra degrees of freedom can lead to the introduction of compensatory errors at the assimilation site, which could then increase uncertainty at other locations, as illustrated by Xiao and Friedrichs (2014b). Similarly, for small changes in the assimilated data an extra flexibility may lead to very different model solutions, also leading to increased uncertainty in model predictions (e.g. Xiao and Friedrichs, 2014a) .

A range of statistical techniques are available to assess this trade off, and a useful review is given by Johnson and Omland (2004). One of the most practical (if not the most general) techniques is cross-validation, as described in the previous section (see also Hastie et al., 2009, section 7.10 for an excellent discussion in a general statistical context). By looking at the effects of adding noise to assimilated remote sensing data, Xiao and Friedrichs (2014a) found that the most complex model they evaluated was also the most sensitive to the introduction of synthetic errors in the assimilated data (Fig. 3). They attributed this result to the extra degrees of freedom that could be 'fit to noise'. This is consistent with earlier findings that model predictive skill deteriorates as complex models can become "overfit" to the data (i.e. too many parameters are fit to inadequate data) (Friedrichs et al., 2006, 2007; Ward et al., 2010).

Aside from directly assessing a model's predictive skill using cross-validation, a number of alternative approaches are available to identify the minimum number of model parameters that are supported by the available data. One of the simplest techniques (in terms of its applicability), is the Akaike Information Criterion (AIC, Akaike 1973). The AIC considers two opposing terms corresponding to the maximum log-likelihood of the parameters given the data ($\ln[L(\widehat{\Theta} \mid \boldsymbol{y})]$,

measuring model data misfit) and a bias-correction factor, that increases with the number of free parameters ($N_\Theta$).

$$\text{AIC} = -2 \ln\left[ L\left( \widehat{\Theta}_p \mid \boldsymbol{y} \right) \right] + 2N_\Theta \qquad (23)$$

Note that for a model fitted by least-squares, the log-likelihood can be approximated by the residual sum of squares (RSS), following Johnson and Omland (2004): $\ln[L(\widehat{\Theta}_p \mid \boldsymbol{y})] \approx -N_y/2 \cdot \ln(\text{RSS}/N_y)$, with $N_y$ being the total number of observations. The AIC, and alternative techniques (weighted AIC, or Bayesian Information Criterion, BIC), seek to quantify the trade-off between bias and variance (e.g., Burnham and Anderson, 2004). Of a range of competing models, the one with the lowest AIC has the greatest empirical support.

A perhaps more intuitive approach is given by the Likelihood Ratio Test (LRT) for e.g. comparing so-called nested models, in which the simpler model is a special case of the more complex model, in the sense that $M_p = f_1$ is a special case of $M_{p+1} = f_1 + f_2$ where $f_2 = 0$. Like the AIC, the LRT aims to account for model complexity in the sense that it compares log-likelihoods:

$$\text{LRT} = J(\widehat{\Theta}_p) - J(\widehat{\Theta}_{p+q}) \qquad (24)$$

with $J(\widehat{\Theta}) = -2\ln[L(\widehat{\Theta} \mid \boldsymbol{y})]$ and index $p + q$ indicating the number of free parameters of the full model. An alternative simpler model (with $p$ parameters) that is not significantly worse than the full model (with $p + q$ parameters) can be selected using this ratio. There is a clear analogy to Eq. (16) in Sect. (5). In other words, although having removed individual parameters (going from $\Theta_{p+q}$ to $\Theta_p$) we may still have an increase in the data-model misfit that is tolerable or insignificant within some limit $\Delta_J$. For nested models only, a value for $\Delta_J$ can be derived from a $\chi^2(\text{df} = q, \alpha)$ distribution. The respective degree of freedom (df) is then assumed to be equal to the difference in the number of free parameters between the full and the reduced model, which is $q$. For LRT with non-nested models an empirical, non-parametric distribution needs to be derived by other means instead, for instance using synthetic (or resample) data sets (e.g., Lewis et al., 2011).

The theory mentioned above is well described by Johnson and Omland (2004), and have already been applied in few ecosystem modelling studies (e.g., Crout et al., 2009; McDonald and Urban, 2010; Ward et al., 2013). The techniques for model selection have generally shown that more complex models are more vulnerable to over-tuning than simpler models. This appears to be because the number of uniquely identifiable parameters in marine biogeochemical models is often very low. Studies based on classic NPZD type models have typically found that the inclusion of as few as three to 15 parameters was supported by the assimilated data (Matear, 1995; Friedrichs et al., 2007; Ward et al., 2013; Löptien and Dietze, 2015). It should however be noted that these studies made use of only very limited datasets, and a higher level of complexity would likely be supported with the incorporation of more comprehensive datasets, especially those describing fluxes.

Ward et al. (2013) sequentially removed parameters from a relatively simple 2NPZD model to show that much of the model structure was redundant, with respect to the assimilated data, Fig. (4). They applied an F-score where the relative change in LRT is related to the relative change in parsimony (i.e. difference in the number of free parameters between the reduced and the full model divided by the degrees of freedom of the full model, $\text{df}_{p+q} = N_y - N_{\Theta_{p+q}}$):

$$\text{F} = \left[ \frac{\text{LRT}}{J(\widehat{\Theta}_{p+q})} \right] \cdot \left[ \frac{N_{\Theta_{p+q}} - N_{\Theta_p}}{\text{df}_{p+q}} \right]^{-1} \qquad (25)$$

As model complexity was reduced, model predictive skill was initially very slow to deteriorate, and $J$ remained similarly low. The increased parsimony of the simpler models led to improved performance in terms of the LRT, and the AIC and Bayesian information criterion (BIC). Once all of the redundant components of the model were removed, removal of essential components led to a rapid increase in $J$, with an associated increase in the other metrics. The LRT selects the simplest model with an F-score below a variable threshold value. The AIC and BIC can be used to select a single model with the lowest score, or preferably to provide individual model weightings for multimodel inference (Burnham and Anderson, 2002), although it appears that this latter has so far seen little application to planktonic ecosystem models.

# 7 Space-time variations in model parameters

Theoretical arguments, as well as results from cross-validations, have revealed problems with the portability of locally calibrated models (e.g., Hurtt and Armstrong, 1999; Friedrichs et al., 2007) and raise the question of how representative local estimates are if applied at larger scales. These limitations encourage estimators that allow spatial and/or temporal variations of parameter values.

For spatial or temporal variation to be useful we have to make sure that the corresponding parameter adjustments reflect changes in the actual underlying (real-world) dynamics. To assess whether this condition is met is a particularly challenging problem that has yet to be adequately addressed. Direct comparisons are needed between optimisations that allow variation in posterior parameter vectors and those that do not. In studies where direct comparisons are made, a common finding is a reduction in the model misfit to the assimilated data by allowing these kinds of variations, but this tells us little. A reduction of the cost function is expected, as a direct consequence of an effective increase in the number of adjustable parameters. As pointed out by Gregg et al. (2009), "skill assessment using assimilated data lacks the independence necessary for a comprehensive, objective evaluation".

Studies where cross-validation is performed to test predictive skill are more informative. Switching between different parameter sets in time or for specific regions may not necessarily be a solution *per se* but may indicate where model refinements have to be investigated (Huret et al., 2007). From analyses of spatially- and temporally varying parameter estimates that improve predictive skill we can learn where and when particular model equations are limited in reproducing changes in plankton dynamics with fixed parameter values. Such analyses should provide important feedback information on revising these parameterisations.

## 7.1 Regional differences between parameter estimates

Satellite ocean colour data are widely used to investigate spatial differences in parameter estimates. In many cases, a local calibration method is applied where parameters are optimised separately to fit Chl*a* data for a number of pre-defined sites or regions spanning a domain of interest. For example, parameters of a 3D-NPZ model were optimised by Garcia-Gorriz et al. (2003) for January and June for two regions, the North- and South Adriatic basin in the Mediterranean Sea. They inferred comparable parameter vectors for the two regions during bloom conditions in January but considerable differences between the regionally optimised parameter sets emerged for June. Garcia-Gorriz et al. (2003) attributed this difference to unresolved variations in plankton composition and changes in biomass concentration between the two basins. Huret et al. (2007) performed a similar assimilation experiment for the Loire and Gironde river plumes in the Bay of Biscay. On the one hand, they found some similarities between parameter estimates for the two distinct river plumes for particular conditions during spring, suggesting the possibility of a common set of parameter values for both plume areas. On the other hand, the authors stressed their optimal parameter estimates to be based on data for a specific period and obtained excessively high Chl*a* concentrations in the Bay of Biscay for the entire simulation year when utilising the mean of parameter estimates for the two plume regions.

Pronounced regional and seasonal differences are not restricted to adjacent seas and coastal areas. Large scale studies for the North Atlantic have shown comparably strong regional differences between parameter estimates (Hemmings et al., 2003; Losa et al., 2004; Doron et al., 2013; Kuhn et al., 2015). A set of sites representing distinct latitude bands was considered for a one year calibration of a NPZ and a NPZD model in Hemmings et al. (2003). The annual cycle at locations on a five degree grid was simulated with variable parameter estimates of a NPZD model in Losa et al. (2004) and individual parameter estimates for thirteen provinces in the North Atlantic, pre-defined according to Longhurst (1995), were derived for a six-compartment 3D biogeochemical model in Doron et al. (2013). Kuhn et al. (2015) estimated NPZD model parameters for six $5 \times 10$ degree regions of the central North Atlantic. Despite the fact that these studies used different models, it is possible to compare some optimised parameters that are equivalent or closely related between all studies. However, little obvious consistency is seen in the spatial patterns between their estimates, although Doron et al. (2013) suggested some similarity between their estimates of phytoplankton maximum growth rate and zooplankton maximum grazing rate with those of Losa et al. (2004). Patterns of spatial variation in parameters are not easily validated as most parameters do not have well-observed equivalents in nature. Nevertheless, Losa et al. (2004) were able to document the plausibility of their posterior photosynthesis parameter values for the maximum phytoplankton growth rate ($\mu_m$ in Sect. 3.1) and intial slope of the P-I curve ($\alpha_{\mathrm{phot}}$ in Sect. 3.3) by comparison with observational estimates of Platt et al. (1991). Six parameters were optimised in all and the posterior parameter fields were cross-validated in a 3D version of their model by comparing the output with an independent SeaWiFS chlorophyll data from 1997-2003 (Losa et al., 2006). The spatially-varying parameter set of Losa et al. (2004), obtained by assimilating Coastal Zone Color Scanner (CZCS) data for the period 1979-1985, was interpolated and extrapolated onto the spatial grid of the 3D model as shown for the two parameters relevant for phytoplankton growth, $\mu_m$ and $\alpha_{\mathrm{phot}}$ respectively (Fig. 5). This enabled the model to simulate the seasonal patterns in SeaWiFS data much better than with a fixed prior parameter vector. An important caveat is that the calibration and validation data sets are essentially two realisations of the same emerging spatio-temporal patterns. To demonstrate improved predictive skill attributable to its dynamics the model would be expected to resolve differences between the two independent data sets, given physical forcing data specific to each period.

## 7.2 Combining sites or regions

The presence of parameter variation between sites or regions for which a model was calibrated independently does not refute the existence of a common parameter vector with which the model could achieve similar results. Garcia-Gorriz et al. (2003) and Hemmings et al. (2003) performed alternative experiments in which regions were combined under a uniform parameter vector constraint, but did not include predictive skill tests for direct comparisons of the performance of spatially-varying and uniform parameter solutions. In other studies, sites have been combined without considering the alternative of allowing parameters to vary spatially. By optimising a 13-parameter model for locations of the Ocean Wheather Ship India (OWSI) and of the Bermuda Atlantic Time-series Study (BATS) simultaneously Hurtt and Armstrong (1999) found that it could capture the primary observed characteristics of the annual cycle at both sites, despite being unable to reproduce the cycle at BATS when calibrated at OWSI. As mentioned in the previous section, the approach of data assimilation over multiple sites has since been used by Schartau and Oschlies (2003) with some suc-

cess in improving predictive skill of a 3D North Atlantic simulation (Oschlies and Schartau, 2005) based on a simultaneous three-sites calibration. A relatively complex global model with 45 adjustable parameters was similarly demonstrated to improve the predictive skill after assimilating time series data at five different calibration sites (Kane et al., 2011).

There is a clear advantage of combining sites or regions, in that it makes more data available to constrain parameters. It also creates a representative sample for the domain of interest, reducing the risk of over-fitting. In contrast, when assimilating data at a single site, Friedrichs et al. (2007) found it necessary to limit the number of adjustable parameters (to four or even less) to avoid portability problems. Use of a larger data set representing a wider diversity of ecosystem behaviour should support a greater number of parameters to be constrained, which would allow a model's true flexibility to be more fully exploited. However, there is a potential disadvantage of combining sites or regions, particularly over large spatial scales, in that the resultant parameter vectors may be less suitable for either region than parameter vectors obtained by local calibration.

Hemmings et al. (2004) introduced the idea of allowing provinces that are in a sense optimal for calibration to emerge during the data assimilation process. A sample of sites from the domain of interest is divided into two similarly distributed sets, one for calibration and the other for cross-validation. The objective is to find "the number and geographic scope of parameter vectors which allow the lowest possible cost of the calibrated model, with respect to the stations in the validation set, to be obtained". The method involves first performing a *whole-domain calibration* where parameters are optimised for all calibration sites, then recursively splitting the domain into two geographic provinces to investigate whether a better calibration can be achieved by optimising parameters for each one separately, a procedure referred to as *split-domain calibration*. The relative merits of the calibration procedures are assessed by cross-validating the posterior parameter vector or vectors against sites from the validation set.

Application of the method to the North Atlantic data set used by Hemmings et al. (2003), with the same NPZ model and twelve adjustable parameters, resulted in the discovery of a two-parameter vector solution having a cross-validation misfit cost 25% lower than that for the single vector solution obtained for all calibration sites. The two sub-domains are shown in Fig. (6). The validation cost was also 24% lower than that obtained when the model was calibrated locally using individual sites. This is consistent with subsequent findings of Xiao and Friedrichs (2014b), where combining sites tends to reduce validation costs. Note that the validation scheme used by Hemmings et al. (2004) may not be able to discriminate well between skill associated with the model dynamics and that associated with the ability of the model to interpolate spatio-temporal patterns between the calibration

sites shown in Fig. (6). This could be resolved by comparison with interpolated output from some purely empirical model fitted to the calibration data.

## 7.3 Spatially varying parameter estimates derived with Bayesian hierarchical modelling

Zhang and Arhonditsis (2009) proposed a Bayesian hierarchical formulation for calibrating aquatic biogeochemical models at multiple sites. In this framework, posterior parameter distributions can vary between sites but the sites share common prior distributions. Fiechter et al. (2013) used this approach to estimate parameter distributions for a 1D NPZD-iron model at two sites in the Gulf of Alaska. Noninformative prior distributions were employed for each parameter so the influence of the priors on the solution for each site was fairly weak. In a parallel Bayes' hierarchical modelling study for the same model, Leeds et al. (2013) assimilated satellite chlorophyll data at nine sites using a spatial Gaussian process model for the parameters with an anisotropic correlation matrix to allow for differences between along-shelf and cross-shelf dependence. The methods employed by Leeds et al. (2013) and Fiechter et al. (2013) seem promising because of their potential for rigorous treatment of uncertainty. However, in the absence of cross-validation experiments, their potential for improving the predictive skill of the models is not well evaluated at present.

## 7.4 Time-varying parameters

The idea of representing seasonal variation in part by temporal variations in the parameters has been examined in various studies (Losa et al., 2003; Brasseur et al., 2005; Dowd, 2006; Roy et al., 2012; Mattern et al., 2012, 2013a, 2014; El Jarbi et al., 2013; Melbourne-Thomas et al., 2013). In some cases, parameters are allowed to vary in space and in time (Tjiputra et al., 2007; Fan and Lv, 2009; Doron et al., 2013; Li et al., 2013). Cross-validation tests comparing the merits of varying and non-varying parameter solutions are mostly lacking, which prevents inferences being drawn about the superiority of these parameter variations for improving predictive skill. Temporal variation is handled naturally by adapting widely used sequential state estimation techniques to obtain parameter values along with state estimates.

Losa et al. (2003) applied a SIR particle filter to a model with 15 time-varying parameters in an assimilation of multi-year time series at the BATS site. The model was treated as a weak constraint with an additive system noise term that was uncorrelated between state variables. Mattern et al. (2013a) instead added noise to their two parameters in a 7-compartment 3D biogeochemical model of the Middle Atlantic Bight, with the advantage that the state evolution over each forecast step was true to the model and correlated errors between state variables were represented. In both cases, the error model is highly subjective, yet it can have a major im-

pact on the results. For instance, Losa et al. (2003) found the level of noise to be a critical factor affecting their solution. This motivated subsequent experiments in which additional time-varying parameters representing the noise level for each state variable were optimised (Brasseur et al., 2005). The posterior parameter trajectories thus obtained were not consistent with the earlier results. Despite the subjective characteristics of the system noise, the solution of Losa et al. (2003) improved the model prediction of unassimilated bacteria data. The necessity of time-variation in the parameters for achieving this is unclear, since no alternative results for static parameter solutions were analysed.

In a more recent BATS assimilation study with a simpler NPZD model, El Jarbi et al. (2013) did compare the performance of time-varying and static parameter solutions. Rather than employing a sequential method, they opted to solve the optimal control problem, i.e. to find parameter trajectories that minimise a cost function for the complete time period. An annual periodicity constraint on posterior parameter trajectories was introduced to allow the calibrated model to be also applied for time periods beyond the range of observations. Optimal periodic parameters were obtained using a two-year data set and validated against independent data for the following three-year period. In cross-validation tests, this solution was shown to improve predictive skill over the static parameter solution of Rückelt et al. (2010). Their results suggest that the time-varying parameter model may capture some aspects of the inter-annual variability, which would indicate dynamical skill.

Mattern et al. (2014) compared the predictive skill of versions of their two-parameter model with time-varying and static parameter solutions. Here, the time-varying solution was obtained using an alternative, emulator-assisted sequential data assimilation scheme. Their cross-validation experiments show a modest improvement in the ability to predict the annual cycle with time-varying parameters. Ability to predict the inter-annual variability was not tested and the achievability of similar predictive skill by purely empirical representations of the annual cycle derived from the observational data is not ruled out.

An experiment allowing both time and space variation in biogeochemical parameters that includes cross-validation is presented by Simon et al. (2015). Performance is compared against that of a model with constant spatially uniform parameters specified a priori but not against static and/or uniform parameter solutions to the data assimilation problem. The study employed an Ensemble Kalman Filter approach for combined parameter and state estimation in a coupled model of the North Atlantic and Arctic Oceans. Estimates for 4 model parameters that varied spatially and seasonally over the domain were obtained by assimilating satellite chlorophyll data for 2008 and 2009 and applied to the estimation of chlorophyll in 2010. A slight improvement was seen in 2010 chlorophyll relative to that for the prior parameter simulation. This suggests a small improvement in predictive skill, per-

haps attributable in part to a better representation of persistent patterns in the annual cycle. A comparison of the assimilating run against independent nutrient data at Station 'M' was generally inconclusive with regard to the potential of the final parameter estimates to improve predictive skill for the nutrient fields

## 7.5 Learning from space and time variation in parameter estimates

As shown in this section, a variety of approaches have been explored for DA with parameters varying in space or time or both. We conclude the section by considering what might be learnt from these types of studies. A common finding is that the posterior misfit cost with respect to the assimilated data is reduced by allowing variation, but this provides no evidence in itself to support the case for parameter variation. Allowing parameter variation increases the number of parameter values to be optimised, making it easier to fit a given data set.

Goodness-of-fit statistics that penalise model complexity in terms of number of parameters (e.g. the F-score of Ward et al., 2013, described in Sect. 6.2) could prove more informative, but are not used. Cross-validation can be used to provide a direct demonstration of differences in predictive skill. In the few studies which do use cross-validation to compare uniform and varying parameter solutions (Hemmings et al., 2004; Mattern et al., 2014; El Jarbi et al., 2013), some evidence of predictive skill is seen but the cross-validation schemes are not shown to discriminate reliably between predictive skill associated with model dynamics and that due to interpolation of patterns in space or persistence of an annual cycle. Better cross-validation schemes will be needed before we can convincingly demonstrate real improvements in the models as a result of introducing spatial and/or temporal variation in parameters.

Allowing parameters to vary reduces the extent to which their values can be constrained by a given set of observations, making an already under-determined problem worse. It could therefore be argued that parameter variation is justified only when there is good evidence to infer that a given model cannot adequately represent the observed variability under the uniform parameter vector constraint. The evidence should be statistically robust, taking into account all relevant sources of uncertainty. The consideration of these additional uncertainties, motivated by its potential for improving parameter estimates (Hemmings and Challenor, 2012), may tend to weaken data constraints further and make the introduction of parameter variation less practical, as well as affecting the strength of the evidence in support of it.

Heterogeneity in the parameter vector is most likely to be useful for structurally simple models. Those models may lack the required flexibility to capture some distinct spatial features observed within large domains or they may fail to resolve specific events during a complete annual cycle. Its introduction may be a sensible alternative to increasing struc-

tural complexity as it does not increase the computational demands of 3D simulations. From an ecological point of view, the need to introduce space and time variations in parameter values reflects limitations in resolving physical environmental changes, or deficiencies in physiological or ecological processes, or all of these factors together. For example, variations in plankton elemental stoichiometry, e.g. variable Chl*a*:C and C:N ratios, induce variations in photosynthetic rates that may not be well described by a model's parameterisation of Chl*a* synthesis and assimilation of nutrients (as discussed in Sect. 3.2). It is helpful to consider biological or environmental reasons why space or time variations of parameter values are expected to improve model performance.

If good reasons are found to support the use of parameter variation for model improvement, then the issue of how to benefit from this spatio-temporal information must be addressed. Spatially varying parameters can be applied directly in 3D models (e.g., Losa et al., 2006). This should work well for hindcasts and short-term forecasts where the application is not compromised by large scale ecological changes. For forecasting, climatological trajectories such as those estimated by El Jarbi et al. (2013) are likely to be of advantage, although their direct application to long-term prediction in the context of global change would be difficult to justify. Application of spatially varying parameters to long-term predictions of global change is possible but will be more complicated than their use in short-term forecasting and it may be necessary to find ways of allowing spatial patterns in biogeochemical parameters to evolve with predicted changes in the physical regimes.

## 8 Emulator approaches

Systematic approaches for parameter optimisation that were successfully applied in 0D or 1D set ups, may become too costly as resolution in space is increased and if the time period for integration is prolonged. This is the case when spatially three-dimensional models with high resolution or steady annual cycles (i.e. periodic solutions) are considered. For the computation of a steady annual cycle (or fixed-point) typically thousands years of model time are necessary, which may result in a number of time steps in the order of $o(10^7)$. Since DA usually involves an iterative optimisation process, typically hundreds or more model evaluations are necessary to obtain a satisfactory parameter set. Thus the necessary time steps during procedures of parameter identification can even reach $o(10^{10})$. Recent attempts aim at replacing computationally costly models with approximations that are less expensive; i.e. emulators have the goal to provide an approximation of the model output trajectory $\boldsymbol{x} := (\boldsymbol{x}_i)_{i=0}^{N_t}$, recalling Eq. (1) of Sect. (2.1):

$$\boldsymbol{x}_{i+1} = M[\boldsymbol{x}_i, \Theta, \boldsymbol{f}_i, \boldsymbol{\eta}_i], \quad i = 0, \ldots, N_t - 1, \tag{26}$$

by substituting the original model $M$ by a simpler one, the emulator $(\widetilde{M})$. Here we disregard a stochastic model approach and consider $\boldsymbol{\eta}_i = 0$ for simplicity.

The application of emulators has emerged in many different fields of science and thus the theoretical background is relatively well developed (e.g. Kennedy and O'Hagan, 2000, 2001; Phillips, 2003; Lucia et al., 2004; van der Merwe et al., 2007; Bliznyuk et al., 2008; Conti et al., 2009; Liu and West, 2009; Castelletti et al., 2012). Two distinct approaches to emulation exist, which we refer to as dynamic emulation and statistical emulation. Both approaches are outlined in the following. Note that the terminology in literature may vary somewhat depending on the respective research field.

### 8.1 Dynamic Emulators

A dynamic emulator (or reduced order or surrogate model) is a substitute for the original model $M$. It makes use of the original model equations but is a simpler representation in terms of resolution or details resolved in the dynamics. The term "simple model" refers here to the computational effort needed to evaluate a solution that is a useful approximation of the solution obtained with the full model. A typical number of model evaluations needed for an automised optimisation process can easily reach the order of $10^{10}$. In this case an emulator becomes particularly valuable, because its application should be much faster than the original model, while as much as possible main properties of the original model are retained. Only then an emulator-based DA approach will give satisfactory results.

Dynamical or physical emulators are based on a simplified model version $(\widetilde{M})$, which might be additionally aligned with interim evaluations of the original model. The term "dynamic" refers to the fact that the emulator is still based on dynamical physical or biogeochemical equations. These can be similar to the ones in the original model but might have some reduced complexity, either by neglecting some processes or by simplifying e.g. the forcing $\widetilde{\boldsymbol{f}}$. Another option is the reduction of accuracy in model output by coarsening the spatial or temporal discretization. For instance, the Transport Matrix (TM) method (Khatiwala, 2007) can be interpreted as an emulator approach with a kind of coarse model. The TM is an emulator that simplifies the original model $M$ by using an approximated and averaged forcing $\widetilde{\boldsymbol{f}}$ in Eq. (26) and a linear approximation of the spatial discretisation, compared to nonlinear advection schemes typically used in ocean models. For the case of a spin-up, as mentioned above, a reduction of accuracy can be achieved by introducing a different criterion that specifies when a tolerable steady periodic solution as been approached.

When using dynamic emulators, it is often insufficient to take the output of the faster but less accurate coarse model during optimisation, because the accuracy of the coarse model $\widetilde{M}$ might be too low to effectively support parame-

ter search process. It can be worthwhile or even necessary to gradually enhance (or update) the emulator's accuracy during the optimisation procedure by introducing special alignment or correction operators. To explain their definition, let us assume we have computed state vectors of the original and of the coarse model with a current set of values for the parameter vector $\Theta_\ell$ in the $\ell$-th step of the optimisation run, i.e.

$$
\begin{aligned}
\boldsymbol{x}_{i+1} &= M[\boldsymbol{x}_i, \Theta_\ell, \boldsymbol{f}_i], \\
\widetilde{\boldsymbol{x}}_{i+1} &= \widetilde{M}[\widetilde{\boldsymbol{x}}_i, \Theta_\ell, \widetilde{\boldsymbol{f}}_i], \quad i = 0, \dots, N_t - 1.
\end{aligned}
$$

We recall that the model state vector $\boldsymbol{x}_i$ consists of the values of the $N_x$ state variables. Thus, in a spatially distributed model, $\boldsymbol{x}_i$ is a vector where every element represents the values at a certain spatial grid point. We here assume that the same numbering is used for the coarse model state $\widetilde{\boldsymbol{x}}_i$.

The alignment operator in optimisation step $\ell$ is then defined element-wise for $\boldsymbol{x}_i$ and point-wise in time by

$$
\boldsymbol{A}_{\ell i} \widetilde{M}[\widetilde{\boldsymbol{x}}_i, \Theta_\ell, \widetilde{\boldsymbol{f}}_i] = M[\boldsymbol{x}_i, \Theta_\ell, \boldsymbol{f}_i]. \tag{27}
$$

Thus, every $\boldsymbol{A}_{\ell i}$ is a diagonal matrix. At the current iterate $\Theta_\ell$, the emulator's output equals the output of the original model. For a parameter vector $\Theta$ close to $\Theta_\ell$, the emulator uses the correction of Eq. (27) – being exact at $\Theta_\ell$ – for the coarse model evaluated at $\Theta$, thus giving only an approximation of the original model. The idea of this *response correction method* is that the deviation between both model outputs remains uncritically similar in a vicinity of $\Theta_\ell$. The emulator is thus not just the coarse model $\widetilde{M}$, but an *aligned* one, $\boldsymbol{A}_{\ell i} \widetilde{M}$, that is now locally optimised. The local optimisation process does not require any additional evaluations of the original model, but only of the cheaper, coarse one. When this inner optimisation gives some new parameter vector $\Theta_{\ell+1}$, the original model is evaluated once again, and the procedure in Eq. (27) is repeated, defining the new emulator for the $(\ell+1)$-th outer optimisation step. In the inner optimisation loop no runs of the original model are needed, and the total number of outer iterations is expected to be lower than in an classical direct optimisation using $M$. This type of optimisation procedure fits in the framework of trust region methods, a class of state-of-the-art algorithms for which a mathematical convergence analysis is shown in Conn et al. (2000).

The method was successfully applied for parameter identification of a transient 1D configuration with a NPZD ecosystem model and for periodic states with climatological forcing in a three-dimensional setting in a N-based model with dissolved organic phosphorus (DOP) (Prieß et al., 2013a, b). Therein, a coarser time-stepping and a less accurate computation of the fixed-point (i.e. a shorter spin-up), respectively, was used to construct the simple model $\widetilde{M}$. For this computationally very costly 3D model, it turns out that the most efficient way is to start the optimisation using the emulator- or surrogate-based optimisation procedure (with a very coarse

model), and then increase its accuracy during the outer optimisation (Slawig et al., 2014).

## 8.2 Statistical Emulators

In contrast to a dynamical emulator, statistical emulators relate the input parameters statistically to the model output and thus to $H(\boldsymbol{x})$ regardless of the dynamical model structure. Generally, statistical emulators interpolate the results of a numerical model from a set of training runs with differing parameters. The aim is to approximate the unknown model output for other input parameters, not included in the training parameter set. Common approaches are based on a polynomial fit (of varying degree). Typically, such interpolations are extended by Bayesian techniques to also obtain uncertainty estimates. For this purpose it is commonly assumed that the model outcome can be represented by a Gaussian process and also that the model output changes smoothly as parameter values are varied. Priori assumptions about reliable parameter ranges and their distribution are required. Another prior choice needed is to determine the respective model output of interest, e.g. results required for $H(\boldsymbol{x})$ to determine $p(\Theta \mid \boldsymbol{y})$ or $L(\boldsymbol{y} \mid \Theta)$, Sect. (2.2). Although there are methods available to reduce the dimensionality for multi-dimensional model output (e.g., Higdon et al., 2008; Leeds et al., 2014), it remains practically infeasible to capture the complete output of a 3D-coupled ocean ecosystem model. While the theory for statistical emulation is relatively well described (e.g., Kennedy and O'Hagan, 2000; O'Hagan, 2006; Liu and West, 2009; Conti and O'Hagan, 2010), statistical emulators are so far rarely applied in biogeochemical ocean modelling.

In Fig. (7) an example of a statistical emulator is provided based on a simple NPZD-type box-model. The model setup is adopted from Löptien and Dietze (2015), thereby resolving seasonal variations in photosynthetically available radiation. Since computational costs are low, the chosen example setup would not necessarily require emulation. However, the model is well suited for testing an emulator approach, because it allows us to evaluate a wide range of model solutions. Figure (7) depicts simulated and emulated root mean square (RMS) errors relative to a set of synthetic observations (i.e. with noise added to model results that are obtained for a prescribed set of parameter values). For our example we use the maximum growth rate of phytoplankton and the maximum grazing rate as free model parameters, while all other model parameters remain fixed. The emulation is based on a second order polynomial, following the approach of Kennedy and O'Hagan (2000). The training runs comprise 25 model simulations in a Latin hypercube design, according to (Urban and Fricker, 2010).

Figure (7) shows very similar results for the emulator and for the full model. In particular, the location of the minimum can be well reproduced by the emulator. Thus, the agreement between emulated and simulated model-data misfit is satisfactory and the emulator could be applied for parame-

ter optimisation. The precision might be further enhanced by considering higher order polynomials and/or more trainings data sets. Note, however, that the complexity of the problem increases with the number of free parameters. In particular, the numerous parameter collinearities in biogeochemical models (e.g. Matear, 1995; Kreus and Schartau, 2015; Löptien and Dietze, 2015) can complicate emulation. Increasing the dimension of the model introduces additional difficulties. One suggestion on how to reduce the dimension of a complex model output is given by Hooten et al. (2011). The authors decomposed modelled surface Chl*a* concentrations of a suite of training runs into singular vectors and predicted the leading modes in dependence of a suite of biological and physical model parameters. During a subsequent parameter optimisation with respect to satellite chlorophyll, they identified zooplankton grazing rate and the light response of phytoplankton to be the most influential parameters. In contrast to most other approaches, where variances are estimated based on Bayesian techniques, Hooten et al. (2011) used a Bayesian approach to estimate the mean values. The study of Leeds et al. (2014) applied a similar technique for DA.

Another example for statistical emulation in biogeochemical modelling is presented by Mattern et al. (2012). Their emulator approach was based on polynomial chaos expansion (e.g. Askey and Wilson, 1985; Wan and Karniadakis, 2006). Mattern et al. (2012) emulated simulation results of Chl*a* concentrations as a function of "maximum zooplankton grazing rate" and the Chl*a*:C-ratio in the Middle Atlantic Bight in year 2006. The authors used an emulator instead of the model to minimise the model-data misfit with respect to daily Chl*a* concentrations observed from remote sensing. They optimised time-constant as well as time-varying parameter estimates. Both approaches improved the overall model performance with respect to Chl*a*. While the original time-varying estimates disregard the actual state of the system, the use of the polynomial chaos method formed the basis of an updated, more reliable method in the study of Mattern et al. (2014) previously discussed in Sect. (7).

Another study of Mattern et al. (2013b) analysed the uncertainty of modelled hypoxia for the Texas-Louisiana shelf based on statistical emulators. The authors investigated the uncertainty due to initial and boundary conditions of biological variables as well as river nutrient loads and phytoplankton growth rate. Additionally, physical factors like river runoff, wind forcing and ocean mixing coefficients were taken into account. The authors revealed considerable uncertainties as their estimates for the hypoxic area varied by more than 40%, when considering reasonable uncertainties in freshwater runoff. Such an extensive analysis would not have been possible without taking advantage of emulators. Furthermore, the use of emulators opens up the possibility of new approaches to exploring the parameter space. One emulator-based technique referred to as "history matching" (Craig et al., 1996), now well-established in other fields and recently applied to the constraint of coupled ocean-atmosphere

model parameters (Williamson et al., 2013), seems a particularly promising approach for parameter identification in marine ecosystem modelling. This relatively simple method uses Bayesian inference to rule out areas of parameter space as implausible, given some set of observations. Estimated uncertainties in both the observations (with respect to the truth) and the emulator (with respect to the model) can be taken into account. The method can be applied iteratively with different observation sets to reduce the size of the plausible region at each stage, either as a precursor to more formal model calibration or as a parameter identification method in its own right.

## 8.3 Combining dynamical and statistical approaches

While emulations based on statistical approaches are comparatively fast, such methods rely on sufficiently large sets of training data (i.e. full model simulations). To generate such training data can be costly, especially for 3D models with high spatial resolutions. To overcome this problem one might consider a combination of statistical and dynamical emulators.

A two stage emulation process is suggested by Hemmings et al. (2015). Their idea is to use a set of 1D models as a dynamical emulator that describes the evolution of the 3D model at representative sites. This Stage 1 emulator allows large ensemble simulations to be run, providing output that could be used as training data for construction of a statistical emulator (Stage 2). The dynamical emulator of Hemmings et al. (2015) is not used in an inner optimisation loop but is used instead to predict 3D model output for arbitrary parameter vectors. It is thus used more like a statistical emulator. In fact, a particular innovation in their study was to quantify uncertainty in the emulator outputs for inference purposes. Another innovation was the inclusion of biogeochemical perturbations associated with lateral advection that are typically ignored in 1D calibration studies. These were derived by averaging 3D model diagnostics over a 10-member ensemble simulation based on a sample of parameter vectors from the search space. Accounting for the lateral flux information was helpful, contributing strongly to the emulator accuracy. The emulator with uncertainty estimates gave robust results for the surface Chl*a* concentration of an ecosystem model of intermediate complexity, considering variation in 8 parameters.

The ultimate aim of the two stage procedure would be to use a sufficiently large number of state estimates of the model based on a (sufficiently precise) dynamical emulator, which can then be used for the construction of a statistical emulator for a cost function or similar metric. The dynamical emulator would effectively bridge the gap between a small reference ensemble that is practical to generate with the full 3D model and the statistical emulator that requires a relatively large training set. The respective metric must incorporate an error model that takes into account all sources of uncertainty in the statistical emulation of the full model. Thus, the uncer-

tainty estimates obtained when training the statistical emulator must be inflated by combining them with the dynamical emulator's own uncertainty estimates. Stage 1 emulation results suggest that it may be important to first extend the latter to include temporal covariance estimates for the parametric uncertainty associated with the averaged 3D model output used. Another important consideration is that global 3D models require long spin-up times to overcome an initial model drift (see Sect. 9.1). The application of dynamical emulation techniques for accelerated spin-up, such as the TM method (Khatiwala, 2007) mentioned in Sect. (8.1), could help to provide a better representation of the parametric variation by increasing the practical length of the spin-up period.

# 9   Parameter estimation of large-scale and global biogeochemical ocean circulation models

Global biogeochemical ocean models are commonly used to investigate the mutual interactions between ocean biota and climate change, a famous example being coupled Earth system models (ESMs) applied in the fifth assessment of the Intergovernmental Panel on Climate Change (IPCC, 2014) and those models that are evaluated as part of the Coupled Model Intercomparison Project (CMIP5; Taylor et al., 2012). Besides individual evaluations of biogeochemical ocean model components (e.g., Ilyina et al., 2013; Tjiputra et al., 2013), global ocean biogeochemical simulation results are often specifically evaluated in terms of their representations of the carbon cycle (e.g., Schwinger et al., 2016). More recent studies also focus on analysing the spread of oxygen minimum zones (e.g., Cocco et al., 2013; Cabre et al., 2015).

## 9.1   Consistency between tracer distribution and ocean circulation field

A major challenge in calibrating biogeochemical models on global scale is that the simulations require many millennia until tracer distributions are in equilibrium with the given circulation field and the biogeochemical processes (Wunsch and Heimbach, 2008). Equilibrium solutions are usually achieved by integrating tracer fields for several thousand years in a so-called model spin-up, based on some seasonally cycling climatological circulation fields. Convergence to steady state conditions depends on the region, tracer type, and form of boundary condition (Wunsch and Heimbach, 2008; Primeau and Deleersnijder, 2009; Siberlin and Wunsch, 2011). It also depends on the values assigned to the parameters of the biogeochemical model, and it is not necessarily a monotonic function of time, but can exhibit inflection points that reflect the interaction of diverse processes happening on different time scales (Kriest and Oschlies, 2015). For parameter optimisation it is meaningful to exclude from a cost function those transient model solutions that involve continuing trends in the redistribution of tracers (see also Séférian et al., 2016).

To attain some equilibrated biogeochemical cycling requires considerable computational time, which makes it particularly difficult to employ methods that exploit the parameter-cost function manifold with a large ensemble of model runs like the MCMC method. The derivation and application of emulators, as described in Sect. (8), is therefore of great value for parameter optimisation of global biogeochemical ocean models. An alternative approach to accelerate the spin-up time is to apply Newton-Krylov methods, by iteratively solving the dynamical system for steady state (e.g., Khatiwala, 2008; Li and Primeau, 2008; Piwonski and Slawig, 2016).

Some speed up of long-term model simulations can also be achieved with an appropriate balance between a model's spatial resolution and the complexity of biogeochemical tracer dynamics, as approached by Ridgwell et al. (2007). Using a coarse grid and a time step of 0.05 yr ($\approx$ 18 days), they could apply an Ensemble Kalman Filter for estimating parameters of their relatively "abstract" biogeochemical component of an ESM of intermediate complexity, building on a DA setup of Annan et al. (2005). Another option is to decrease the number of model runs by applying the variational adjoint method for parameter optimisation (Sect. 2.2.3). Results of an adjoint global biogeochemical model were used by Tjiputra et al. (2007) to determine first derivatives of a cost function with respect to the parameters, see also Appendix (C). However, because of local minima or flat regions in the cost function optimal estimates may then depend on the initial guess of parameter values, as discussed in Sect. (2.2.3).

Some DA applications may not require equilibrated tracer dynamics to maintain steady seasonal cycles, e.g. when applying sequential DA approaches with recurrent analyses steps and corrections of the simulated state variables. An example is the study of Simon et al. (2015) who introduced an ensemble-based DA method for a large-scale biogeochemical model of the North Atlantic and Arctic Ocean. The focus of their study was to estimate spatial and temporal variations of phytoplankton and zooplankton loss rate parameters as well as model states, in order to establish an operational system for hind- and forecasts of Chl$a$ concentrations. Their model was initialised with climatological data of nutrients and oxygen and initial values of the other biogeochemical state variables were set to low constant values. Prior to the DA period (2007-2010) their model was integrated for a six year period, starting in year 2000. This simulation period is much shorter than the few hundreds of years typically needed to equilibrate tracer distribution and ocean circulation in the North Atlantic and Arctic Ocean (e.g., Wunsch and Heimbach, 2008) and the optimised hindcast simulations may therefore not be expected to represent detrended seasonal cycles of biogeochemical tracer distributions and mass flux.

In summary, various procedures for calibrating large-scale and global biogeochemical ocean circulation models exist, but are presently challenged by overcoming limitations in

computational time to approach equilibrated steady cycles in biogeochemical tracer distributions. Data availability on global scale introduces additional limitations to act as constraints for parameter identification of global biogeochemical models.

## 9.2 Data for parameter estimation and calibration in global ocean biogeochemical models

In regard to the ocean's key role in global carbon cycling and hence for the climate system, four different types of data are typically considered for assessing and calibrating global biogeochemical ocean models: i) data of dissolved inorganic tracers, e.g. distributions of nutrients, oxygen, alkalinity and dissolved inorganic carbon, ii) data products derived from remote sensing measurements, e.g. of chlorophyll *a*, or plankton primary- or net community production, iii) in situ measurements or composite data of organic and inorganic matter concentrations, fluxes and rates e.g. at different time-series stations, and iv) observations of the gravitational flux of organic particles to the ocean interior, transporting particulate organic matter through the water column.

For the calibration and assessment of large-scale or global biogeochemical models many studies resort to using climatological data sets, e.g. of nutrients and oxygen, components of the carbonate system (e.g., Watanabe et al., 2011; Tjiputra et al., 2013). Also common is the additional or exclusive use of observational estimates that were derived from remote sensing measurements, like primary production rates and surface concentrations of Chl*a* (e.g., Carr et al., 2006; Tjiputra et al., 2013; Nevison et al., 2015; Simon et al., 2015). Given the often high level of structural complexity of ocean biogeochemical models we find only few studies that involved more elaborate data such as organism groups or fluxes of organic matter. Examples can be found in Gehlen et al. (2006), who compared simulated and observed particle fluxes, or Aumont et al. (2015), who compared simulated and observed dissolved iron concentrations and nitrogen fixation rates. Likewise, Ward et al. (2012) considered satellite based estimates of surface Chl*a* concentrations of different taxonomic groups as specified in Hirata et al. (2011).

One reason for the fallback to rather basic data types such as climatological nutrient concentrations for global model evaluation is the sparse distribution of open ocean, in situ observations. One example is the scarcity of global microzooplankton biomass observations in the ocean, as depicted in Buitenhuis et al. (2010). Direct, in situ, open ocean ship-based observations are sparse in space and time mainly for logistic reasons (and costs) and we therefore find available sets of situ data to be noticeably biased towards certain areas and periods (e.g. towards coastal areas, summer season in the high latitudes, and the northern hemisphere, Kriest et al., 2010).

Ocean measurements of rates are particularly valuable, but these may not be straightforward to accomplish, e.g. isotopic measurements on a research vessel. Some rate measurements may also suffer from large methodological uncertainties, e.g. measurements of nitrogen fixation. Of similar value, comparable to rate measurements, are observations of oceanic particle flux, as obtained from sediment traps or from optical methods (e.g., Gardner, 2000; Buesseler, 1991). These data provide only patchy information about the particle flux in the world ocean. Their analysis and interpretation are also difficult, since particles produced at the surface are subject to horizontal transport by advection, hampering the establishment of correlations between surface and deep fluxes, particularly for slowly sinking particles (e.g. a meter per day) in energetic current fields (e.g. a meter per second) (e.g., Siegel et al., 2008; Frigstad et al., 2015). Attempts to calibrate global models against individual observations of particle flux have not yet revealed any unique "best" model solution (Gehlen et al., 2006; Kriest and Oschlies, 2013). To establish a consistent linkage between surface primary production rates, e.g. as derived from remote sensing, and observed in-situ measurements of particle flux remains a major challenge. This requires a close look at parameters that link production the euphotic zone to deep carbon export. Parameters that specify vertical flux and remineralisation of organic matter ultimately determine carbon storage (Kwon et al., 2009).

## 9.3 Parameters relevant for global ocean biogeochemical modelling

The joint effect of particle flux and remineralisation is often described by one or two parameters in global models. Early models referred to an exponential function of remineralisation with depth (Bacastow and Maier-Reimer, 1991), which - in equilibrium - would correspond to a constant particle sinking velocity and constant remineralisation. Another, common description of particle flux (and hence of subsequent remineralisation) is the consideration of a power law of depth: $F(z) \, z^{-b}$, where $b$ is usually set to $b = 0.858$, representing the open-ocean composite value derived by Martin et al. (1987) from sediment traps (e.g., Maier-Reimer, 1993). Empirical fits to various observations of particle flux suggest that $b$ may vary between 0.3-1.4 (Martin et al., 1987; Berelson, 2001; Van Mooy et al., 2002; Buesseler et al., 2007). This typical range of variation of $b$ has been used and tested in global biogeochemical models e.g. analysing how its value affects dissolved tracer concentrations in the ocean (Kwon and Primeau, 2006, 2008; Kriest and Oschlies, 2013). Kwon et al. (2009) coupled a simple global biogeochemical model with a one-box atmosphere and found a large effect of this parameter on atmospheric $pCO_2$, highlighting the relevance of this parameterisation in ESM simulations. Since this parameterisation is widely used (e.g., Kwon and Primeau, 2006, 2008; Najjar et al., 2007; Parekh et al., 2005) we will have a closer look at its implicit assumptions in the following and discuss potential constraints for the estimation of respective parameters.

Under steady state conditions $b$ can be interpreted as being equal to a constant remineralisation rate $r$ divided by a particle sinking speed $a$ that increases with depth: $b = r/a$ (Kriest and Oschlies, 2008). The associated potential mechanisms that may lead to a vertical increase in sinking speed are selective export of large and fast particles to deeper layers, or repackaging of small particles into larger ones by zooplankton egestion. An alternative interpretation is to assume the sinking speed to be constant while the remineralisation rate decreases with depth. This implies that particles may become more refractory and less susceptible to bacterial degradation, or that bacterial activity is reduced by the decrease in temperature at depth. Other parameterisations of particle flux profiles have been applied in global models, e.g., constant sinking and remineralisation (leading to an exponential flux curve; e.g., Bacastow and Maier-Reimer, 1991), or models that explicitly simulate different groups of particles with different size and properties (e.g., Gehlen et al., 2006; Schwinger et al., 2016). Cabre et al. (2015) provide an excellent overview about different parameterisations for models applied in CMIP5.

So far few attempts have been made to systematically calibrate parameterisations of particle export and remineralisation in global biogeochemical models. Kwon and Primeau (2006) assimilated annual mean phosphate data into a simple global ocean biogeochemical circulation model to optimise globally uniform $b$. Their study shows that the value of of $b \approx 1$ can be well identified for their model when using global climatological data. According to their approach, the tracer distributions are dynamically consistent with their solution of ocean circulation. Such consistency is relevant and $b$ may not be derivable by applying any simulated circulation field to climatological data, e.g. of phosphate (Wilson et al., 2015). Furthermore, Wilson et al. (2015) also discussed how the identification of $b$ is affected by uncertainties in the transport and remineralisation of dissolved organic matter.

In a recent study of Kriest et al. (2017) the export parameter $b$ turned out to be well identifiable, with an optimal value of $\approx 1.3$, based on annual mean climatologies of dissolved nutrients and oxygen. As in Kwon and Primeau (2006) their biogeochemical model explicitly resolves seasonal cycles. Plankton parameters that act on seasonal scale within the upper, near surface layers are more difficult to identify, if annual mean climatological data are used. Figure (8) exemplifies this difficulty, based on results from Kriest et al. (2017) who optimised six biogeochemical parameters in total. The example reveals differences in the sensitivity of the cost function with respect to variations of two contrasting parameters, the zooplankton mortality ($\kappa_{zoo}$) and $b$ respectively. These differences can be visualised from projections of the parameter-cost function manifold ($\Theta, J(\Theta)$), as obtained during parameter optimisation (Schartau and Oschlies, 2003; Ward et al., 2010). To better illustrate the discrepancy between the two parameters in Fig. (8) we defined two arbitrary cost function threshold limits $\Delta_J = J(\Theta)/J(\widehat{\Theta}) - 1$ and $\Delta_J = 0.01$ and

$\Delta_J = 0.001$ (see Eq. 16 in Sect. 5). The projected pattern of the zooplankton mortality reveals a much smaller sensitivity of the cost function (larger uncertainty), compared to the robust (nearly quadratic) pattern of the export parameter $b$. Furthermore, for $\kappa_{zoo}$ some bimodal structure exists within $\Delta_J \leq 0.01$, which impedes parameter identification. Clearly, annual mean climatologies of dissolved inorganic tracers provide only little information on plankton dynamics in the upper layers, while particle export dynamics (which integrates over large spatial and temporal scales) are well constrained by the large-scale distribution of dissolved inorganic tracers. Thus, simulated tracer concentrations at great depth do not critically depend on every parameter that specifies growth and mortality of the plankton.

In the presence of very diverse time and space scales, which is typical in global biogeochemical ocean modelling, the selection of data sets and the definition of the error model strongly affect parameter identification. We also stress that parameter estimates of global biogeochemical modelling studies are conditioned by the applied circulation, which can have a large impact on simulated tracer fields (Najjar et al., 2007), and by the boundary conditions of e.g. of organic matter burial at the sea floor (Kriest and Oschlies, 2013). To date, it remains unclear whether parameters optimized for a given circulation field will improve model simulations in a different setting, e.g. with a different circulation or forcing, as induced by climate change scenarios.

## 9.4 Impact of parameter uncertainties on climate model projections into the future

A typical large-scale application of marine biogeochemical models is their use in ESMs from which projections of future climate change can be derived for different emission and land-use scenarios. Output of such models helps to inform scientists, but also society and policymakers about possible consequences of human action on the climate system. A key example is the most recent assessment report of the IPCC that featured ESMs with fully interactive carbon cycles (IPCC, 2014). An appropriate treatment of the uncertainties contained in the applied scenarios and employed models is crucial for correctly interpreting model projections, informing the societal debate about climate policies and thus strengthening the base for developing relevant measures. A full treatment of uncertainties in the projections of ESM is beyond the scope of our review and we can only address this topic here briefly.

A comprehensive attempt to account for uncertainties in the models when determining likelihoods of reaching certain climate goals, like the politically widely accepted $2°C$ warming goal, was presented by Steinacher et al. (2013) and Steinacher and Joos (2016). Employing a somewhat simplified ESM of intermediate complexity, they ran perturbed parameter ensembles with some ad hoc assumptions about prior probability distributions of the model parameters. The

skill of individual ensemble members was then measured by comparison of model hindcasts with available observations of the current state of the Earth system. A single, pragmatic skill score was used in the assessment and led to an improved posterior estimate of parameter probability distributions. The model dynamics then mapped the parametric uncertainty onto the model projections. From the large ensemble of model solutions that were, in hindcast mode, not inconsistent with the observational constraints, the authors could then successfully derive likelihoods of reaching various climate goals.

Note that reproducing the current climate state is merely a necessary condition for model skill, but may not constrain the model's ability to correctly simulate the sensitivity to natural or anthropogenic environmental change. Observational information on past climate change, such as glacial-interglacial changes may help to better constrain the models' sensitivity to changing environmental conditions, even though no historical analog of the current anthropogenic perturbation is known in terms of the rapid rate of change. Still, any information about model sensitivities to applied perturbations is extremely valuable, be it derived from lab or mesocosm experiments or from historical information. DA is a promising tool to combine such information on very different space and time scales and to develop an improved understanding of how the earth system works and may respond to ongoing environmental change.

## 10 Summary and perspectives

The survey of Arhonditsis and Brett (2004) revealed that relatively few aquatic biogeochemical modelling studies a) considered parameter optimisation (8.5%), b) provided values of data-model misfit (30 %), or c) performed quantitative parameter sensitivity analyses (28%). Since then there has been a vast increase in the number of those studies where the assimilation of biological and chemical data into planktonic ecosystem models is described. Likewise, we now find a wide field of different studies that address problems of parameter identification. Although positive, this development has also brought up diverse approaches whose contexts and connections are sometimes difficult to understand. Furthermore, we face a variety in terminology and notation, which makes it even more arduous to comprehend the various studies and the significance of their findings. With this review we aim to provide support to readers.

The theoretical backbone for studies of parameter estimation and uncertainty builds first of all on how model errors and observational errors are treated. Specifying the error model is an essential first step in the workflow of parameter identification, enabling the subsequent derivation of conditional probabilities and cost functions. Our review shows that there is no ultimate standard error model or procedure but a meaningful practice is to become explicit about these errors

and to reconsider the underlying assumptions for discussions of parameter estimates and model results. Whether the DA approach conserves mass and/or energy is relevant in this respect, depending on the scientific problem addressed. Some ecosystem model applications may not critically depend on mass conservation, e.g. when simulating plankton growth to act as food source in regional simulations of fish stock size and recruitment. In biogeochemical applications the conservation of mass can be essential, in particular for large-scale or global ocean applications.

As in many other fields of science, the basic estimation methods considered in plankton ecosystem DA studies are Bayesian estimation and Maximum Likelihood. Their major differences are how prior information enters the DA approach and how estimates and uncertainties are evaluated. The consideration of prior parameter values from preceding studies is meaningful and likely alleviates parameter identification problems. A drawback then is that asymptotic (point-wise) approximations of posterior uncertainty covariance matrices, as described herein, may not apply. But when the model parameters in question have been estimated before in a number of comparable settings, it may seem a tragic waste of effort and information to pursue an ML approach without prior information. A similar issue arises in specifying an "ignorance" prior, and the choice of using BEs when no prior information is available can also be questioned.

We included a section on typical basic parameterisations of plankton models, mainly to stress that the treatment of light- and nutrient limitation may differ between modelling studies. Furthermore, we touched on the problem of resolving phytoplankton losses specified by e.g. grazing and aggregation parameters. Latest plankton growth models account for physiological acclimation effects, responsible for variations between carbon fixation, cellular allocation of nitrogen and phosphorus, and Chl$a$ synthesis. Those variations are relevant for DA, in particular if flux estimates of carbon (e.g. $CO_2$ utilisation and respiration) are of primary concern. It is thus worthwhile to discuss some of the underlying dynamics that can be resolved with the plankton ecosystem model rather than treating it as a "black box" for simulating Chl$a$ concentrations.

Many acclimation or optimality-based models have been qualitatively calibrated with data from laboratory experiments. DA approaches for parameter estimation were only done in a few of these studies. Going from laboratory data to the assimilation of data from mesocosm experiments can be a useful intermediate step for testing e.g. acclimation or adaptive models and for assessing uncertainty ranges of parameter values. In this respect, parameter estimates of one experiment can be used for cross-validation with data of another independent mesocosm experiment. On the one hand, simulations of the physical environment of mesocosms are easier to implement, compared e.g. to setting up a 1D model for an ocean site. On the other hand, parameter estimates obtained from the assimilation of mesocosm data might not be

representative for ocean simulations. Although more diffi-
cult, model cross-validations between different ocean sites
or regions provide valuable insight, eventually specifying a
model's predictive skill under oceanic conditions.

Some studies have shown that an increase in model com-
plexity may not automatically improve predictive skill. This
can be partially attributed to over-fitting, which can yield
parameter estimates that improve model-data misfits at one
site but induce unreasonable model results at other ocean
sites. Such results illustrate the vital role played by well-
designed cross- validation experiments. A critical element of
cross-validation is whether the assimilated data are truly in-
dependent from the data used for testing model skill. This
is, for instance, not typically the case if observations from
different years but of the same characteristic region are used
unless inter-annual variability dominates over the repeating
seasonal dynamics. Regional differences between parameter
estimates are informative and have the potential to reveal a
model's limitations in a way that can suggest improvements.

Parameter identification becomes more difficult as we go
from local and regional scale to large-scale and global model
simulations. Algorithms for parameter optimisation require
multiple model evaluations, which can be computationally
expensive for global biogeochemical models. The procedure
for optimising parameter values can be accelerated with the
application of an emulator. We discussed the use of dynam-
ical and statistical emulators. The dynamical emulator is a
simpler representation of a full model operator that is com-
putionally expensive, thereby approximating the underlying
model dynamics. A statistical emulator interpolates model
output from a set of training runs with different values as-
signed to the parameter vector. Based on the derived statis-
tics it can be applied to approximate unknown model out-
put for other input parameters. Both emulator approaches
have been shown to efficiently support the search for op-
timal parameter values. The development and use of emu-
lators of biogeochemical models will likely gain in impor-
tance along with improved computer performance. A promis-
ing approach is to apply models with coarser resolution or a
series of 1D models (distributed over ocean regions) as dy-
namical emulators for 3D global biogeochemical model sim-
ulations. Studies have shown that sufficient accuracy of the
emulator can be achieved with repeated intermediate align-
ments of the dynamical emulator. Alternatively, differences
between 1D- and 3D results can be statistically quantified as
emulator uncertainty, impacting on the parameter search pro-
cess and used to modify the emulator-based cost function.

Parameter identification in global marine biogeochemi-
cal circulation models is still in its infancy, due to the high
computational requirements, the huge range of spatial and
temporal scales to be covered, and the comparatively sparse
spatial-temporal distribution of data in the ocean. In contrast
to local optimisations, the consideration of all relevant spatial
and temporal scales has one major advantage in that it pro-
vides the opportunity to rigorously test and benchmark bio-

geochemical models. In addition to tasks and complications
mentioned in our review, care must be taken in the selection
of appropriate data sets, assuring their relevance (or poten-
tial) for answering the questions posed. Moreover a critical
evaluation of the respective roles of physics, biogeochem-
istry, exchanges across the model's boundaries and, possibly,
ecology is an as yet unresolved task.

A recurring problem associated with parameter optimisa-
tion is that marine biogeochemical models are often unreal-
istically simplified, while at the same time remaining uncon-
strained by data. Ideally, models should be developed to min-
imise the number of uncertain parameters yet maintain a level
of complexity that is suited to their intended use in answering
specific questions (e.g., Denman, 2003). To accomplish this
we may not only think of new model approaches, but also of
collecting respective data that can help to constrain solutions
of these models.

## 10.1  Modelling prospects

A commonality of new model formulations is to focus on
principles, e.g. by considering the adaptation of traits towards
optimal trade-offs (e.g., Wirtz and Pahlow, 2010; Dutkiewicz
et al., 2009; Smith et al., 2015), or by accounting for allomet-
ric relationships in growth and plankton interaction (e.g., Ba-
nas, 2011; Acevedo-Trejos et al., 2015), or by using micro-
bial traits in a functional gene approach (Reed et al., 2014).
Recent studies have begun to simulate ecosystem complex-
ity and allow the model to "self-organise" according to a rel-
atively simple set of ecological and physiological rules or
"trade-offs" (Bruggeman and Kooijman, 2007; Follows et al.,
2007). A major advantage of this approach is that the mod-
els are able to resolve greater ecological diversity with fewer
specified parameters whose values can be assumed to be spa-
tially invariant. This diversity allows the simulated plankton
community to reorganise across broad environmental (e.g.
spatial) gradients. But the identification of the most impor-
tant trade-offs governing competition between organisms re-
mains a major challenge (Tilman, 1990; Litchman et al.,
2007, 2012).

Perhaps one of the most remarkable developments is the
revival of thermodynamically inspired ecosystem theories
for modelling biogeochemical cycling in the oceans (e.g.,
Vallino, 2011). In the review of Vallino and Algar (2016) the
concept and potential of the maximum entropy production
principle are addressed. In this modelling approach life in the
ocean is perceived as units of e.g. covalent bonded chains of
carbon atoms that create disequilibria of energy and mass be-
tween organisms. These disequilibria lead to different func-
tional pathways in biogeochemical cycling, accompanied by
a flexible evolution of structural dependencies between nu-
trient or substrate availability, plankton and other organisms.
Such novel or revised approaches are expedient and help to
create new ideas in terms of how to design models and mea-

surement strategies that may alleviate the problems of parameter identification.

## 10.2 Examples of recent advances in data availability

The use of previously underexploited data sets (for example those linking organism size to key ecophysiological rates; Baird and Suthers, 2007; Banas, 2011; Ward et al., 2012) have the potential to bring new constraints on model behaviour, and may go some way to alleviating the degree of underdetermination that is typically associated with parameter estimation. New data sources, such as the Bio-Argo profiling floats, should also advance our understanding, e.g. by documenting seasonal variations of deep Chl*a* maxima in remote oligotrophic regions (Mignot et al., 2014). These Bio-Argo profile data have the advantage that they resolve biogeochemical properties with a relatively high frequency of five to ten days over a sampling period of up to two years.

A substantial fraction of recent fluourescence measurements from Bio-Argo platforms has already been included in a new global Chl*a* database described and provided by Sauzède et al. (2015b). Their quality-controlled data comprise profiles of total Chl*a* concentration together with some additional estimates of the relative contributions from pico-, nano-, and micro phytoplankton. The employed relationship between the relative size distributions and total Chl*a* concentration was derived from an extensive analysis of High- Performance Liquid Chromatography pigment data in combination with Chl*a* fluorescence measurements (Sauzède et al., 2015a). The consideration of these profile data will possibly facilitate the estimation of photoacclimation parameters in particular, and of phytoplankton growth parameters in general.

Data products from remote sensing measurements are continueously improved and new empirical relationships between photosynthesis and respiration are derived to estimate net community production (NCP) on the global scale (e.g., Westberry et al., 2012; Tilstone et al., 2015). These spatially resolved estimates may help to constrain parameters of plankton respiration and remineralisation rates. In spite of large uncertainties, the assimilation of NCP estimates from remote sensing into biogeochemical models may impose additional constraints on parameters that affect solutions of air-sea exchange of $CO_2$ and of organic matter export. In this respect we also stress that upgrades and analyses of time-series data are more then ever essential to make inference about organic matter flux and ecosystem functioning (e.g., Emerson, 2014), which may introduce additional constraints for identifying values of a larger number of parameters of plankton ecosystem models. Finally, we point to latest products from compilations and syntheses of oceanic and atmospheric $CO_2$ data collected by a large international community (Rödenbeck et al., 2015; Bakker et al., 2016). Data products like air-sea $CO_2$ flux of specified ocean regions (biomes), as derived in (Rödenbeck et al., 2015), in combination with data

of nutrient concentrations and $O_2$ will likely put new light on those parameters that determine variations of the elemental stoichiometry ($C:N:P:O_2$) in model results of inorganic and organic matter cycling.

## 10.3 Harmonising research foci in marine ecosystem modelling and data assimilation

The application of DA methods has become standard for calibrating marine ecosystem- and biogeochemical models. But scientific insight can differ between DA studies considerably. In the literature we find that there is often an imbalance between level of sophistication of the ecosystem model used and the DA method employed. This is likely due to the fact that marine ecosystem-/biogeochemical modelling studies integrate knowledge from different scientific fields, of which each has its own foci, objectives, and expertise i.e. plankton ecology, physical oceanography, marine geochemistry, and mathematics and statistics. It is difficult to track major advancements in marine ecosystem modelling when considering the different views from each of these research fields. Furthermore, the design of experimental studies and the collection of field data are often achieved without harmonising the needs of biologists with the modelers' exigencies (Flynn, 2010).

Facets of parameter identification in biological modelling disclose major commonalities and disparities between the objectives expressed in the different research fields. Discussions on parameter identification are therefore helpful to achieve a common understanding and to promote communication between observers, modelers, and statisticians. Problems of parameter identification may thus be well addressed by pooling expertise across multiple disciplines, without losing sight of scientific objectives. Such joint efforts should help planktonic ecosystem models to fulfil their potential as quantitative tools for aquatic sciences.

*Author contributions.* Individual sections of our review were written by one or more lead author(s), with contributions from the other authors (Phil Wallhead, PW; John Hemmings, JH; Ben Ward, BW; Ulrike Löptien, UL; Thomas Slawig, TS; Iris Kriest, IK; Andreas Oschlies, OA, and Markus Schartau, MS). All authors were involved in mutual revisions of the individual sections. The sections' lead authors are: 1. Introduction (MS), 2. Theoretical background (PW, MS, and JH), 3. Typical parameterisations of plankton models (MS), 4. Error models (PW), 5. Parameter uncertainties (MS), 6. Cross-validation and model complexity (BW), 7. Space-time variations in model parameters (JH), 8. Emulator approaches (UL and TS), 9. Parameter estimation of large-scale biogeochemical ocean circulation models (IK, AO, and MS), 10. Summary and perspectives (MS), Appendix A (PW), Appendix B (MS), and Appendix C (MS and TS). Shubham Krishna performed parameter optimisations, MCMC computations of the mesocosm modelling example, as well as calculations of the 2D parameter arrays.

*Acknowledgements.* We gratefully acknowledge the support from the International Space Science Institute (ISSI). This publication is an outcome of the ISSI's Working Group on "Carbon Cycle Data Assimilation: How to consistently assimilate multiple data streams". We like to thank four anonymous referees who provided constructive and helpful comments. The time and effort they spend on our manuscript is much appreciated. The examples of mesocosm data assimilation are based on the mesocosm modelling environment designed for the large integrated projects Surface Ocean Processes in the Anthropocene (SOPRAN, 03F0662A) and BIOACID (03F0728A), both funded by the German Federal Ministry of Education and Research (BMBF). Contributions from Iris Kriest, Ulrike Löptien, and Thomas Slawig were supported by the BMBF funded PalMod - Paleo Modelling: A national paleo climate modelling initiative.

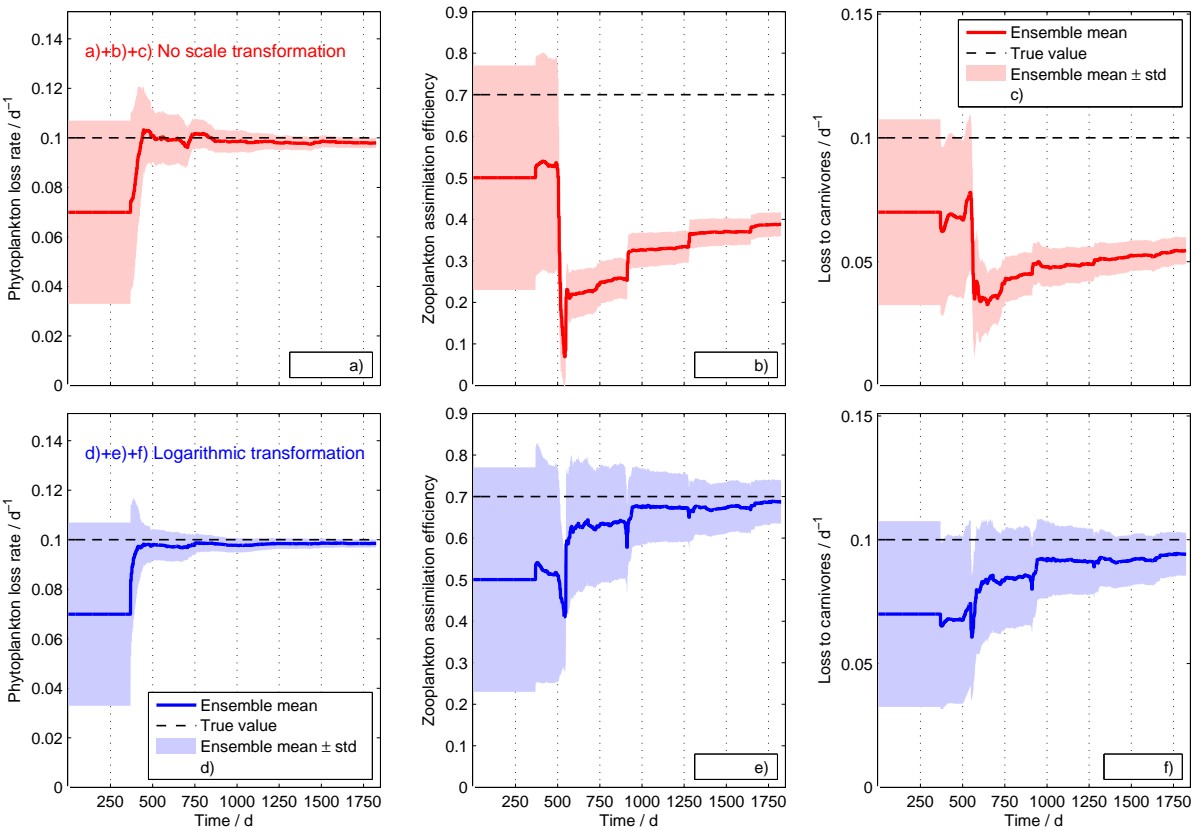

**Figure 1.** Time evolution of parameter estimates in a simulation test of an Ensemble Kalman Filter using untransformed data (a-c, top row) and using logarithmic transformed data (d-f, bottom row) (Simon and Bertino, 2012, Fig. 3). Solid lines and shading show ensemble means and standard deviations averaged over 20 simulation experiments, while dashed lines show the true parameter values. The data were generated using Gamma-distributed observational errors with standard deviation 30% (see Simon and Bertino, 2012). A transformation can significantly reduce the bias of parameter estimates by the end of the assimilation period. Figure was redrawn from results provided by Ehouarn Simon, with permission from Elsevier. Copyright of figure content by Elsevier.

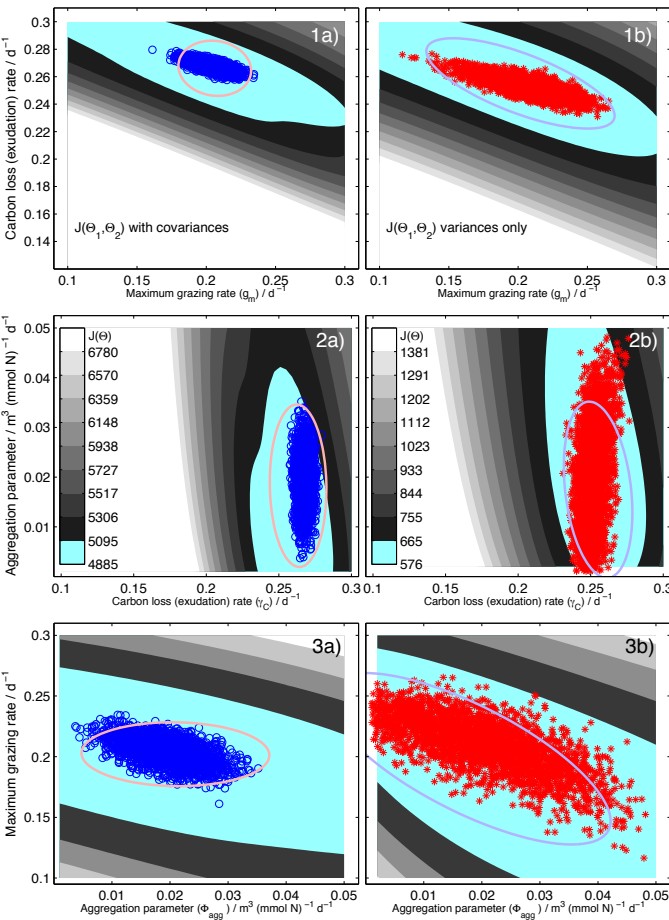

**Figure 2.** Cost function contours when varying values of a combination of two parameters $J(\widehat{\Theta}_m \pm \Delta_m, \widehat{\Theta}_n \pm \Delta_n)$ around the optimum estimate at $(\widehat{\Theta}_m, \widehat{\Theta}_n, \min(J))$, while values of all other parameters remain fixed. Each plot resolves a pairwise combination out of three parameters that all specify phytoplankton biomass losses. The two columns reveal differences in error margins due to different cost functions with same data for the same model: a) with covariances explicitly regarded and b) all data are assumed to be independent. First row (1a and 1b): combination of maximum grazing rate ($\Theta_1 = g_m$) and carbon exudation rate ($\Theta_2 = \gamma_C$). Second row (2a and 2b): combination of the aggregation parameter ($\Theta_3 = \Phi_{agg}$) and $\gamma_C$. Third row (3a and 3b): combination of $g_m$ and $\Phi_{agg}$. Markers show credible regions of parameter estimates obtained with Markov Chain Monte-Carlo (MCMC) method (dots for $J$ with covariances, asterisks for $J$ with variances only). Error ellipses (lines) depict point-wise 95% confidence regions derived from an approximated and inverted Hessian matrix, according Eq. (22). The cyan colored region embeds all cost function values that are lower than an upper threshold $\triangle J^*(\alpha = 0.05)$, derived from a distribution of $J(\widehat{\Theta}) - J^*(\widehat{\Theta})$, where $J^*(\widehat{\Theta})$ are cost function values at $\widehat{\Theta}$ using resampled data (Fig. B1 in Appendix).

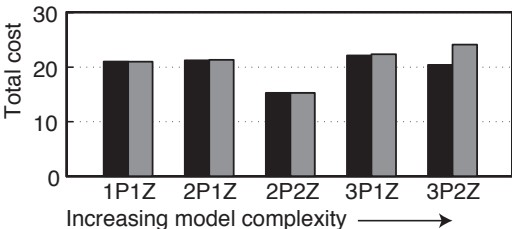

**Figure 3.** Predictive skill for five ecosystem models of different complexity, after assimilation of satellite data (black) and after assimilation of satellite data with 20% added noise (grey) (Xiao and Friedrichs, 2014a). The most complex model appear to be the most sensitive to errors in the data, in terms of its cross-validated predictive skill.

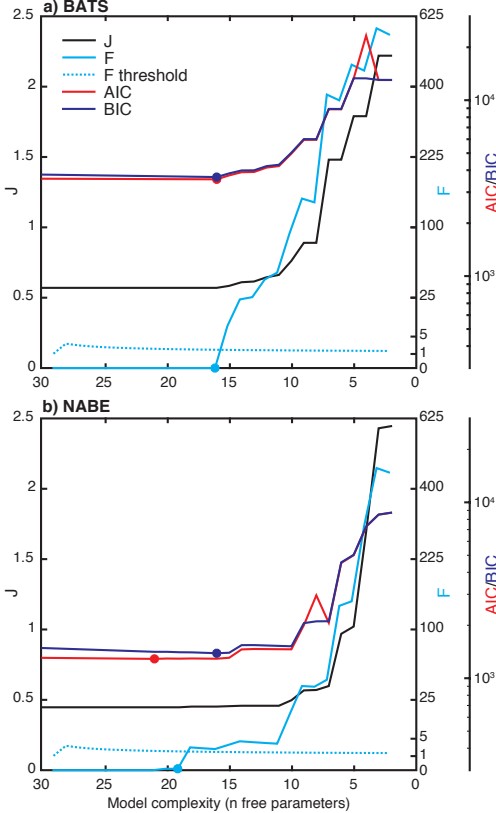

**Figure 4.** Model selection metrics at the Bermuda Atlantic Time-series Study (BATS) and the North Atlantic Bloom Experiment (NABE), as a function of complexity across a suite of nested ocean biogeochemical models (Ward et al., 2013). The least-squares misfit, J (left-hand axis), increases monotonically with decreasing complexity, as it does not penalise model complexity. The likelihood ratio test, F (first right-hand axis), compares each reduced model to the full model, and selects the simplest that is not significantly worse than the full model (F<F threshold). The AIC and BIC (second right-hand axis) both contain terms that account for model data misfit and complexity, and the optimal model is the one with the lowest score. In each case, the optimal model is indicated by a dot.

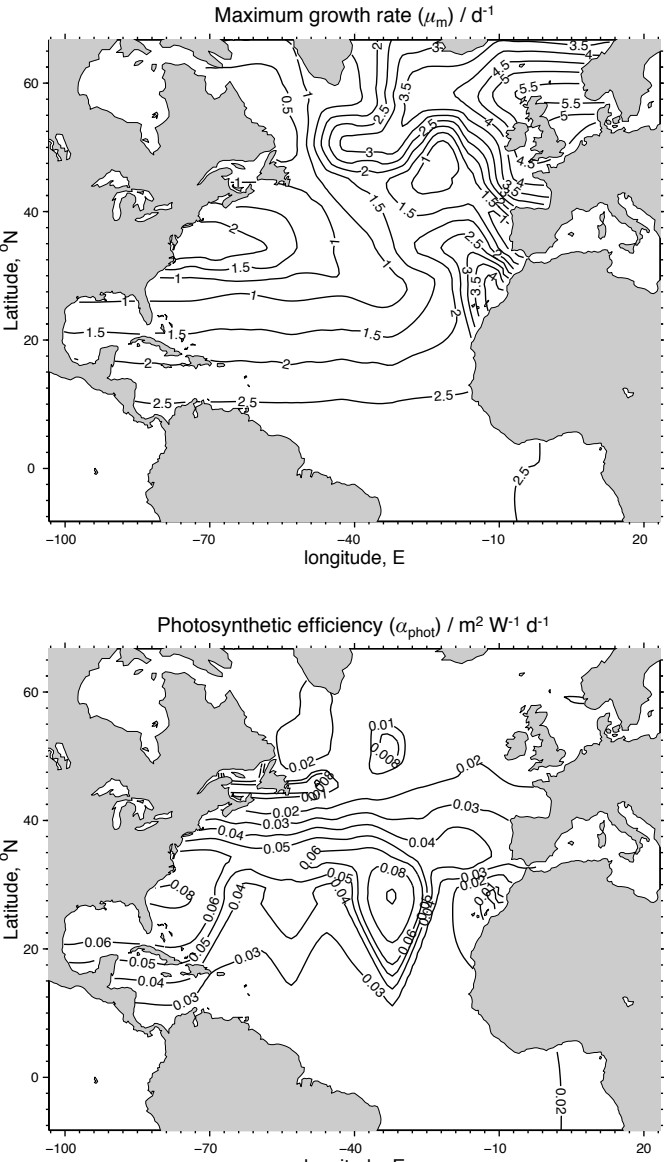

**Figure 5.** Spatially varying estimates for the phytoplankton maximum growth rate ($\mu_m$ in unit $d^{-1}$) and photosynthetic efficiency ($\alpha_{phot}$, in $m^2\ W^{-1}\ d^{-1}$) used in a 3D modelling study of the North Atlantic (Losa et al., 2006). The parameter estimates are based on those obtained in a previous assimilation of satellite chlorophyll data (Losa et al., 2004). Permission to include Fig. (2) from Losa et al., (2006) was granted by the authors. Figure is used with permission from Elsevier. Copyright of original figure by Elsevier.

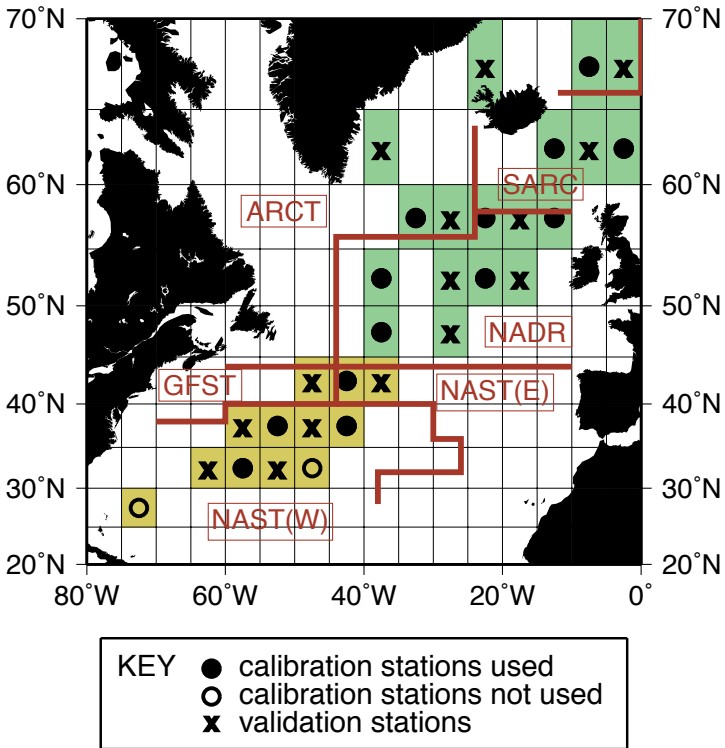

**Figure 6.** Geographic extent of the two sub-domains giving the optimal calibration in the split-domain calibration study of Hemmings et al. (2004), shown here in yellow and green. Also shown are the distributions of the sites used from the calibration set to obtain the parameter vectors for each sub-domain and the sites used for cross-validation. Biogeochemical provinces defined by Longhurst (1998) are shown for reference. ARCT: Atlantic Arctic Province; SARC: Atlantic Subarctic Province; NADR: North Atlantic Drift Province; GFST: Gulf Stream Province; NAST: North Atlantic Subtropical Gyral Province. Figure (6a) of Hemmings et al., (2004) is shown with permission from Elsevier. Copyright of original figure by Elsevier.

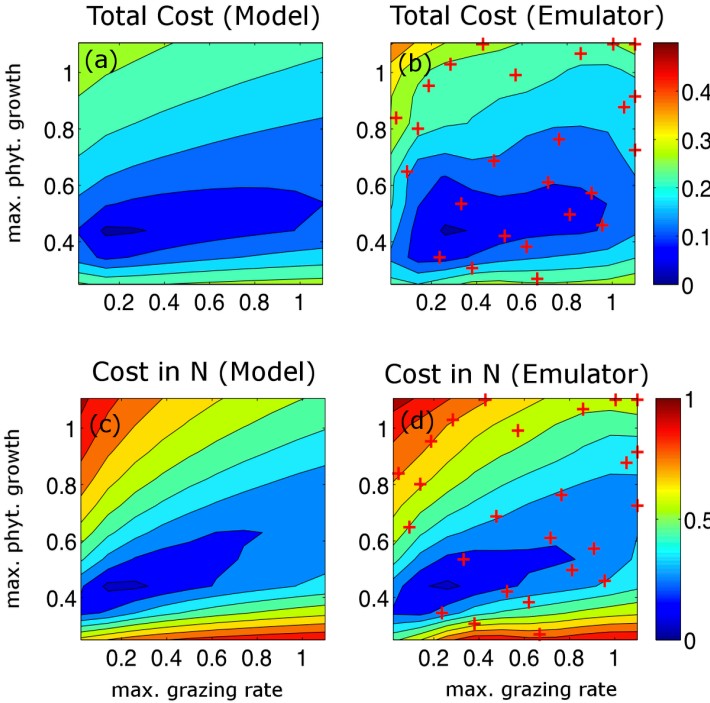

**Figure 7.** Simulated (a,c) and emulated (b,d) RMS (root mean square) error depending on the maximum growth rate of phytoplankton and the maximum grazing rate. Simulated and emulated RMS-errors are provided relative to "synthetic observations", based on a simulation for a given parameter set (HI=15 W m$^{-2}$; m=0.06 d$^{-1}$; $\mu_{\text{max}}$=0.51 d$^{-1}$; H$_n$=0.8 mmol N m$^{-3}$; $m_{\text{PD}}$=0.1 d$^{-1}$; $m_{\text{DN}}$=0.1 d$^{-1}$; $H_Z$=0.9 mmol N m$^{-6}$; $m_{\text{ZN}}$=0.01 d$^{-1}$; $m_{\text{ZD}}$=0.01 d$^{-1}$; $g_{max}$=0.21 d$^{-1}$), which is disrupted by reddish noise (AR(3)-process) with a standard deviation of 0.09 mmol N m$^{-3}$. (Notation after Löptien and Dietze, 2015). Sub panels (a,b) are based on all prognostic variables, while the RMS error in (c,d) is based on nitrate (NO$_3^-$ ) only (c,d). Red crosses mark the training data.

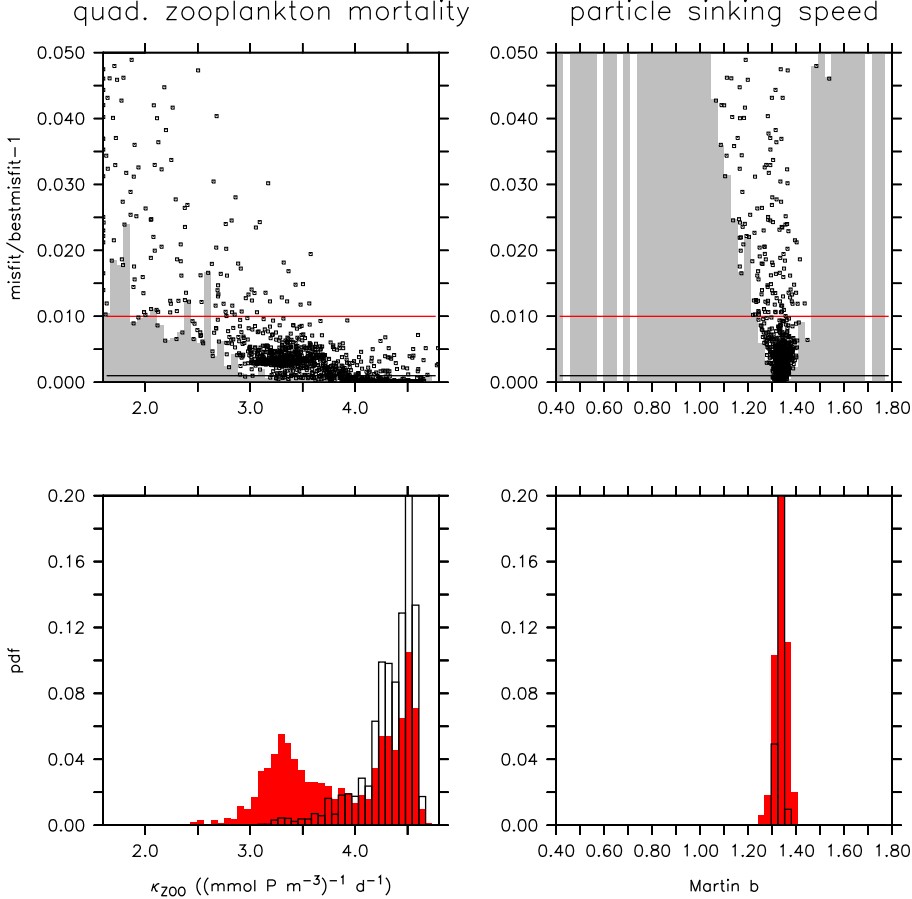

**Figure 8.** Projections from parameter-cost function manifold $(\widehat{\Theta}_l, J(\Theta))$ as obtained during the optimisation of six biogeochemical parameters. Parameters shown are quadratic zooplankton mortality $\kappa_{zoo}$ (left panels) and rate of vertical increase of particle sinking speed, $a$, expressed as quotient $b = r/a$, where $r$ is particle remineralization rate (right panels). Upper panels: cost function (volume-weighted root-mean square error, divided by global mean concentration of each tracer) expressed as its deviation from the minimum. Parameters of all model simulations in the optimisation trajectory were grouped into 50 classes. Grey bars show minimum cost within each class. Red and black horizontal lines indicate deviation from minimum cost of 1% and 0.1%, respectively. Squares show the cost of each individual. Note that the y-axis only extends to 5% above minimum cost at $(\widehat{\Theta}, J(\widehat{\Theta}))$. Lower panels: parameter distribution (PDF) of all model simulations, whose cost do not exceed a threshold limit of $\triangle_J = 1.01 \cdot J(\widehat{\Theta})$ (1%, red bars) or $\triangle_J = 1.001 \cdot J(\widehat{\Theta})$ (0.1%, open bars) 0.1% (open bars) of the minimum cost, see Eq. (16 and text).

## Appendix A:  The Variable Lag Fit with unknown error variances (Sect. 4.4)

In a Variable Lag Fit (VLF), we assume that the truth at time $t_i$ is related to the model output by a kinematic model error ($\zeta$) in phase or time lag $\tau_i$. Equation (2) becomes:

$$\mathbf{x}^t(t_i) = \mathbf{x}(t_i + \tau_i) \tag{A1}$$

A notable feature of this model error representation is that it introduces unknowns $\tau$ that can be conditionally optimised by searching forwards and backwards in time within saved model output, i.e. *without rerunning the dynamical model*. For the demonstration in Fig. (A1) we assumed that the time lag errors are normal and independent: $\tau_i \sim N(0, \sigma_\tau^2)$. This independence assumption may seem restrictive; for example, a misplaced eddy might be expected to impose some correlation between the $\tau_i$ for a set of cruise data. Nevertheless, we find that the method is somewhat robust to neglected lag correlation. Moreover, this formal neglect enables a large computational simplification since the lags can then be optimised one by one, see Wallhead et al. (2006).

For the observational error in Fig. (A1) we assumed lognormal errors with no interpolation or conversion factors, and that all measured variables were sampled simultaneously. Equation (3) becomes:

$$y_{ij} = x_{ij}^t \cdot \exp\left(\epsilon_{ij} - \frac{\sigma_j^2}{2}\right) \tag{A2}$$

at each measurement time $t_i$ and for each measured variable $j$ (Nutrient, Phytoplankton and Zooplankton). For simplicity we further assumed that the observational errors were independent between measurements and data types, hence $\epsilon_{ij} \sim N(0, \sigma_j^2)$. Note that the $\epsilon$ may be considered to include a component of kinematic model error ($\zeta$) without affecting the parameter estimation, hence we refer to them as *residual* errors below. Assuming that the ecosystem parameters $\theta_e$, time lags $\tau$, time lag variance $\sigma_\tau$ and observational error variances $\sigma$ are all unknown, a joint posterior mode estimate of $\Theta = (\theta_e, \tau, \sigma_\tau, \sigma)$ is obtained by maximising the posterior density $p(\Theta \mid \boldsymbol{y})$, equivalent to minimising the following cost function:

$$J(\Theta) = n \log \sigma_\tau^2 + \sum_i \frac{\tau_i^2}{\sigma_\tau^2} + n \sum_j \log \sigma_j^2$$
$$+ \sum_{ij} \frac{(\log y_{ij} - \log x_j(t_i + \tau_i) + 0.5\sigma_j^2)^2}{\sigma_j^2} \tag{A3}$$

To test this cost function, we simulated data from the NPZD model of Oschlies and Garçon (1999) in a 0D setting using the parameters values and sine-squared forcing function from Wallhead et al. (2013). Three years of simultaneous weekly samples of $N$, $P$, and $Z$ were simulated assuming independent normal time lag errors with standard deviation $\sigma_\tau = 10$ days and independent normal residual errors $\sigma_{\log N} = 0.1$, $\sigma_{\log P} = 0.2$, $\sigma_{\log Z} = 0.3$. The data were assimilated into the same NPZD model by one of two methods. In the 'standard fit', no time lag error was assumed and search parameters $\Theta = \{\theta_e, \sigma_{\log N}, \sigma_{\log P}, \sigma_{\log Z}\}$ were estimated by minimising only the final two terms in A3 with $\tau_i = 0$ for all $i$. In the VLF, $\Theta = \{\theta_e, \tau, \sigma_\tau, \sigma_{\log N}, \sigma_{\log P}, \sigma_{\log Z}\}$ was estimated by minimising Eq. (A3). In both cases, we assume uncertainty in only two of the 15 biological parameters, namely the phytoplankton maximum uptake rate $V_m$ and the zooplankton maximum grazing rate

$g$ (hence $\theta_e = (V_m, g)$). For all search parameters, allowed ranges were $\pm 50\%$ about the true values, equivalent to unbiased uniform priors with 29% prior uncertainty. Initial values of the search parameters were chosen at random from this prior, and optimisations were repeated over 10 random restarts to avoid local minima. The experiment was repeated over 20 simulated data sets to obtain the statistics in Table (A1).

Caution must be exercised here regarding the estimation of $\sigma_\tau$. If the prior for $\sigma_\tau$ permits very low or zero values then the MAP estimation will push the estimate of $\sigma_\tau$ towards zero irrespective of its true value. This is because, unlike the fourth term in Eq. (A3), the second term can be made exactly zero with $\tau = \mathbf{0}$ as long as $\sigma_\tau^2 > 0$, in which case the negative contribution of $n \log \sigma_\tau^2$ may produce a spurious, deeper minimum of $J$ near to $\sigma_\tau = 0$. We have found that this spurious minimum need not influence estimation as long as the sample size and the lower limit of the allowed range or rectangular prior for $\sigma_\tau$ are sufficiently large, Fig. (A2). An alternative solution may be to assume a prior that drops smoothly to zero as $\sigma_\tau^2 \to 0$, such as an inverse gamma distribution (cf., Kavetski et al., 2006).

To investigate estimation of the time lag variance parameter $\sigma_\tau$ we obtained cost function profiles by fitting the same data set using a range of fixed values of $\sigma_\tau$, Fig. (A2). We see that with three years of weekly NPZ sampling the cost function function has a strong minimum close to the true value of 10 days, and this minimum should be approached even if the allowed range (prior uncertainty) for $\sigma_\tau$ reaches as low as 1 day. However, if we decrease the number of sampled years, or especially the number of sampled variables, the minimum becomes weaker and a spurious minimum close to $\sigma_\tau = 0$ starts to encroach on the profile. A sufficiently low minimum allowed value $\sigma_\tau^{(\min)}$ may then lead to estimates converging to this spurious minimum.

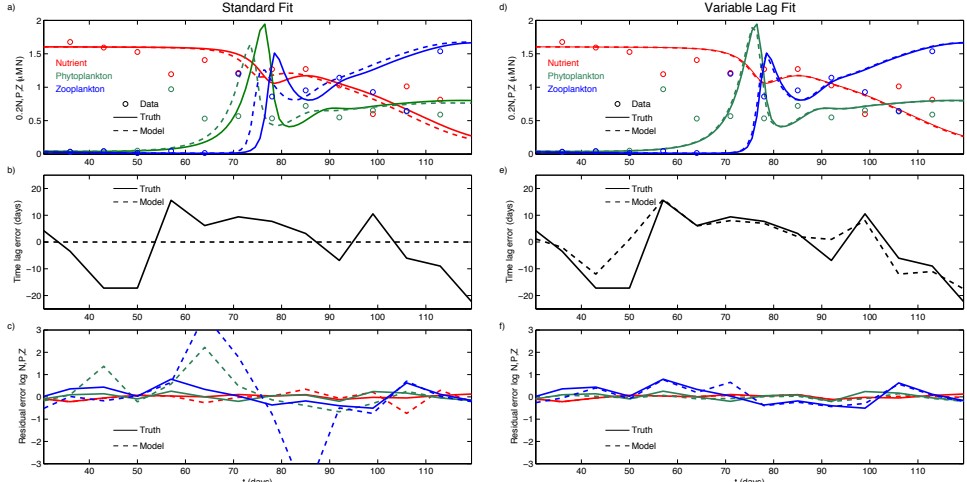

**Figure A1.** Demonstration of the Variable Lag Fit (VLF) applied to a simulated data set. a) shows the system trajectory with the true parameter values (solid lines), the data (dots) simulated assuming normal and independent time lag errors ($\sigma_\tau = 10$ days) and residual errors ($\sigma_{\log N,P,Z} = 0.1, 0.2, 0.3$, see Table A1), and the system trajectory with the VLF parameter estimates (dashed lines, overlapping with solid). b) compares the true time lags (solid) with those estimated from the VLF (dashed). c) compares the true residual errors with those estimated by the VLF (dashed, same colour code as in a)). Three years of data were assimilated but only the initial and post-bloom period of the first year is shown for clarity.

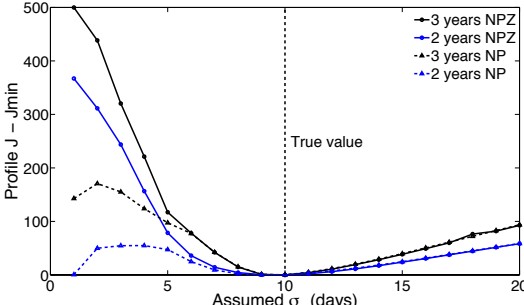

**Figure A2.** Profiles of the Variable Lag Fit cost function (-2 × posterior density) relative to the minimum value for a range of assumed values of the time lag error standard deviation $\sigma_\tau$. For each $\sigma_\tau$, Eq. (A3) was minimised over $(\theta, \tau, \sigma_{\log N,P,Z})$ for the same data set. Different curves correspond to different scenarios for the number of sampled years (at weekly sampling frequency) and number of simultaneously sampled variables (black = 3 years, blue = 2 years, solid lines with circles = Nutrient-Phytoplankton-Zooplankton sampling, dashed lines with triangles = Nutrient-Phytoplankton sampling). The extent to which each curve has a deep minimum close to the true value $\sigma_\tau(\text{true}) = 10$ days indicates the feasibility of estimating $\sigma_\tau$ for the corresponding sampling plan.

**Table A1.** True parameter values and means $\pm$ 1 SD of estimates over 20 simulated data sets, using a standard fit method and a variable lag fit method (see Eq. A3). Three years of weekly NPZ data were simulated using the true values (first row) for the maximum nutrient uptake rate $V_m$, zooplankton grazing rate $g$, residual standard deviations $\sigma_{\log N, P, Z}$, and time lag standard deviation $\sigma_\tau$ (for experiments with lags imposed). With no time lags, the standard fit accurately recovers the true parameter values (third row), but with time lags (fourth row) the standard grazing rate estimates are biased and imprecise, while the residual variances have strong positive bias as they are forced to account for the time lag errors. The variable lag fit avoids these biases and accurately partitions the variance between residual error and time lag error (fifth row).

| | Lags? | $V_m$ (day$^{-1}$) | $g$ (day$^{-1}$) | $\sigma_{\log N}$ | $\sigma_{\log P}$ | $\sigma_{\log Z}$ | $\sigma_\tau$ (days) |
|---|---|---|---|---|---|---|---|
| True values | — | 0.66 | 2.00 | 0.10 | 0.20 | 0.30 | 10.0 |
| First guesses | — | $0.66 \pm 0.19$ | $2.00 \pm 0.58$ | $0.10 \pm 0.03$ | $0.20 \pm 0.06$ | $0.30 \pm 0.09$ | $10.0 \pm 2.9$ |
| Standard fit | No | $0.66 \pm 0.00$ | $2.03 \pm 0.07$ | $0.10 \pm 0.01$ | $0.20 \pm 0.01$ | $0.31 \pm 0.01$ | — |
| Standard fit | Yes | $0.68 \pm 0.03$ | $2.61 \pm 0.44$ | $0.27 \pm 0.02$ | $0.46 \pm 0.07$ | $0.75 \pm 0.14$ | — |
| Variable Lag Fit | Yes | $0.67 \pm 0.01$ | $2.03 \pm 0.19$ | $0.07 \pm 0.01$ | $0.18 \pm 0.02$ | $0.29 \pm 0.02$ | $9.2 \pm 0.7$ |

## Appendix B: Mesocosm example (Sect. 5.4)

For our example we account for six different types of measurements from mesocosms of the Pelagic Ecosystem $CO_2$ Enrichment Study (PeECE I, Engel et al., 2005; Delille et al., 2005): 1) dissolved inorganic carbon (DIC, mmol m$^{-3}$), 2) nitrate (NO$_3^-$, mmol m$^{-3}$), 3) nitrite (NO$_2^-$, mmol m$^{-3}$), 4) Chl$a$ (mg m$^{-3}$), 5) PON (mmol m$^{-3}$), 6) POC (mmol m$^{-3}$). Concentrations of NO$_3^-$ and NO$_2^-$ are not explicitly resolved by the model and therefore these measurements are combined. We refer to their sum as dissolved inorganic nitrogen (DIN). Thus, the number of components of the observation vector is $\mathbf{y}$ is $N_y = 5$. Observations are available on a daily basis over a period of 23 days ($N_t = 23$). The vector includes daily means of nine mesocosms at $t_i$, i = 1, ... , $N_t$. The dynamical model equations determine twelve state variables ($N_x = 12$). The corresponding vector of model counterparts to observations is $H_i(\mathbf{x})$, with carbon and nitrogen biomass concentrations of phytoplankton (PhyN & PhyC), of zooplankton (ZooN & ZooC), of detritus (DetN & DetC), and carbon concentration of (particulate) macrogels (GelC). The data-model residual vector is:

$$
\begin{aligned}
\mathbf{d}_i \quad &= \mathbf{y}_i - H_i(\mathbf{x}) \quad (\text{B1}) \\
&= \underbrace{\begin{pmatrix} \text{DIC}_i \\ \text{DIN}_i \\ \text{Chl}a_i \\ \text{PON}_i \\ \text{POC}_i \end{pmatrix}}_{\text{obs}} - \underbrace{\begin{pmatrix} \text{DIC}_i \\ \text{DIN}_i \\ \theta_i^{\text{Chl:C}} \cdot \text{PhyC}_i \\ (\text{PhyN} + \text{ZooN} + \text{DetN})_i \\ (\text{PhyC} + \text{ZooC} + \text{DetC} + \text{GelC})_i \end{pmatrix}}_{\text{model}}
\end{aligned}
$$

As an error model we assume additive Gaussian errors applying Eq. (4) in Sect. (2.1.3). The standard errors ($\boldsymbol{\sigma}_i$) represent the observed variability between the nine mesocosms, based on daily measurements. Residual error covariance matrices can thus be derived for every sampling day: $\mathbf{R}_i = \mathbf{S}_i \, \mathbf{C}_{(\mathbf{y})} \, \mathbf{S}_i$. The matrices $\mathbf{S}_i$ include diagonal elements with $\boldsymbol{\sigma}_i$ at date $t_i$, while off-diagonal elements are zero. The elements of matrix $\mathbf{C}_{(\mathbf{y})}$ represent correlations between the different types of observations, which were determined for two time intervals: exponential growth and post-bloom period. The distinction between periods of bloom buildup and post-bloom can be particularly meaningful when C and N (or P) data are assimilated. Correlations can switch sign and thus the sign of the data-model residual $\mathbf{d}_i = \mathbf{y}_i - H_i(\mathbf{x})$ matters. For example, PON and dissolved inorganic carbon (DIC) are strongly negatively correlated during the exponential growth phase. During the post-bloom period DIC may still decrease at times when PON concentration declines as well, which yields a weak but positive correlation. The standard errors ($\sigma_i$) can be written in matrix notation with off-diagonal elements being zero:

$$
\mathbf{S}_i = \begin{pmatrix} \sigma_i^{(\text{DIC})} & 0 & \cdots & 0 \\ 0 & \sigma_i^{(\text{DIN})} & \cdots & \vdots \\ \vdots & \vdots & \ddots & 0 \\ 0 & \cdots & 0 & \sigma_i^{(\text{POC})} \end{pmatrix} \quad (\text{B2})
$$

Correlations during exponential gowth ($t_i$; i = 1, ... , 13) / and during post bloom period ($t_i$; i = 14, ... , 22):

$$
\mathbf{C}_{(\mathbf{y})} = \begin{pmatrix}
\text{DIC} & \text{DIN} & \text{Chl}a & \text{PON} & \text{POC} \\
1 & 0.96/0.2 & -0.95/-0.22 & -0.97/\mathbf{0.20} & -0.97/-0.64 \\
. & 1 & -0.96/-0.37 & -0.95/-0.26 & -0.95/\mathbf{0.16} \\
. & . & 1 & 0.96/0.63 & 0.92/-\mathbf{0.26} \\
. & . & . & 1 & 0.94/-\mathbf{0.55} \\
. & . & . & . & 1
\end{pmatrix}
$$

$$(\text{B3})$$

For days with some missing observations (e.g. no PON measurements), the dimension of the vectors $H_i(\mathbf{x})$ and $\mathbf{y}_i$ and matrices $\mathbf{S}_{(\mathbf{y}_i)}$ and $\mathbf{C}_{(\mathbf{y})}$ have to be adjusted for that date accordingly. We disregard any prior information and the cost function (Eq. (13) in Sect. 2.3) reduces to:

$$
J(\Theta) = \sum_{i=1}^{N_t} (\mathbf{y}_i - H_i(\mathbf{x}))^T \mathbf{R}_i^{-1} (\mathbf{y}_i - H_i(\mathbf{x})) \quad (\text{B4})
$$

For our second cost function we assume all data to be independent (i.e. all off-diagonals of $\mathbf{C}_{(\mathbf{y})}$ are zero) and Eq. (B4) can be further simplified to a sum over all individual vector components (indexed with $j$):

$$
J(\Theta) = \sum_{i=1}^{N_t} \sum_{j=1}^{N_y} \frac{(y_{ij} - H_{ij}(\mathbf{x}))^2}{\sigma_{ij}^2} \quad (\text{B5})
$$

The mesocosm model environment was coded in FORTRAN and compiled as shared library so that we could use R as free software environment for statistical computations. For parameter optimisation (simulated anneadling) and for the analysis of the posterior (Markov chain Monte Carlo method) we applied the R package FME of Soetaert and Petzoldt (2010).

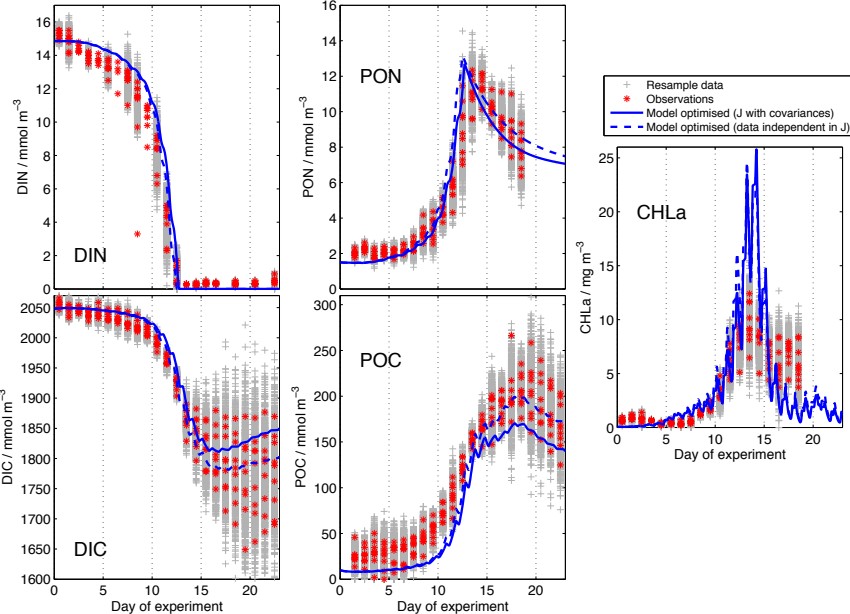

**Figure B1.** Observations of nine mesocosms (red asterisks), resampled data (gray markers) and optimised simulation results (blue lines): Dissolved inorganic nitrogen and carbon (DIN and DIC), particulate nitrogen and carbon (PON and POC), and chlorophyll *a* concentration (CHLa).

## Appendix C: Development of an adjoint model (Sect. 5.3.3)

Adjoint models can be used to efficiently compute the derivative (or gradient) of the cost function $J$. In a parameter identification problem, $J$ depends on $\Theta$ both indirectly via the state variable $\boldsymbol{x}$ and also directly if prior information is incorporated. The optimisation problem can thus be written as

$$\min_{\Theta} J(\boldsymbol{x}(\Theta), \Theta), \tag{C1}$$

where $\boldsymbol{x} = (\boldsymbol{x}_i)_{i=0}^{N_t}$ summarize all time instances of the model variables. To evaluate the derivative of the cost w.r.t. the parameters $\Theta$, we may apply the chain rule and obtain

$$\frac{dJ}{d\Theta} = \sum_{i=0}^{N_t} \frac{\partial J}{\partial \boldsymbol{x}_i} \frac{d\boldsymbol{x}_i}{d\Theta} + \frac{\partial J}{\partial \Theta}, \tag{C2}$$

where we omitted the arguments $\boldsymbol{x}(\Theta)$ and $\Theta$ for brevity.

The needed derivatives of the model variables $\boldsymbol{x}_i$ w.r.t. the parameters $\Theta$ can be obtained by taking the total derivative w.r.t. $\Theta$ of the equations of the dynamical model, Eq. (1):

$$\frac{d\boldsymbol{x}_{i+1}}{d\Theta} = \frac{\partial M}{\partial \boldsymbol{x}_i} \frac{d\boldsymbol{x}_i}{d\Theta} + \frac{\partial M}{\partial \Theta}, \quad i = 0, \dots, N_t - 1. \tag{C3}$$

This time propagation scheme for the derivatives is often called the tangent linear model.

The idea behind adjoint models is to avoid this direct computation, whose effort grows linear with the number of parameters $\Theta$. For this purpose, we re-formulate Eq. (C1), treat both arguments of $J$ independently and use the model equation as a constraint in the optimisation process. This can be expressed as

$$\min_{(\boldsymbol{x}, \Theta)} J(\boldsymbol{x}, \Theta) \text{ s. t. } \boldsymbol{x}_{i+1} = M[\boldsymbol{x}_i, \theta_e, \boldsymbol{f}], i = 0, \dots, N_t - 1. \tag{C4}$$

A useful overview of adjoint model construction and applications is given in Kasibhatla (2000). An established approach to construct an adjoint model is to generate adjoint code directly from the numerical code of a model, based on algorithms that implement the chain rule for automatic differentiation (Griewank, 1989, 2003). According to the description of Giering and Kaminski (1998), a numerical model can be treated as a composition of differentiable functions, where each function represents a statement in the numerical code. The differentiation of such composition can be automated by highly sophisticated tools that yield tangent linear and adjoint FORTRAN code (e.g., Faure and Papegay, 1997; Giering and Kaminski, 1998). The application of adjoint construction tools (e.g., Tangent linear and Adjoint Model compiler, TAMC, of Giering and Kaminski, 1998) have been shown to perform well for studies with large-scale ocean general circulation models that include even complicated boundary conditions (e.g., Stammer et al., 1997; Marotzke et al., 1999; Wunsch and Heimbach, 2007; Heimbach et al., 2011).

Another approach is based on a discretised extended Lagrange equation. Under certain mathematical assumptions, a solution of Eq. (C4) corresponds to a saddle-point $(\boldsymbol{x}, \Theta, \boldsymbol{\lambda})$ of the Lagrangian

$$\mathcal{L}(\boldsymbol{x}, \Theta, \boldsymbol{\lambda}) = J(\boldsymbol{x}, \Theta) + \sum_{i=0}^{N_t-1} \boldsymbol{\lambda}_i^\top (M[\boldsymbol{x}_i, \theta_e, \boldsymbol{f}] - \boldsymbol{x}_{i+1}). \tag{C5}$$

The vector $\boldsymbol{\lambda} = (\boldsymbol{\lambda}_i)_{i=0}^{N_t-1}$ contains the Lagrange multipliers $\boldsymbol{\lambda}_i$, each of which corresponds to one time step in the model. A saddle-point of $\mathcal{L}$ satisfies the conditions

$$0 = \frac{\partial \mathcal{L}}{\partial \boldsymbol{x}_i} = \frac{\partial J}{\partial \boldsymbol{x}_i} + \boldsymbol{\lambda}_i^\top \frac{\partial M}{\partial \boldsymbol{x}_i} - \boldsymbol{\lambda}_{i-1}^\top, \qquad i = 1, \dots, N_t \tag{C6}$$

$$0 = \frac{\partial \mathcal{L}}{\partial \Theta} = \frac{\partial J}{\partial \Theta} + \sum_{i=0}^{N_t-1} \boldsymbol{\lambda}_i^\top \frac{\partial M}{\partial \Theta} \tag{C7}$$

$$0 = \frac{\partial \mathcal{L}}{\partial \boldsymbol{\lambda}} \tag{C8}$$

Here, we again omitted the arguments, and set $\lambda_{N_t} = 0$ in the first equation to keep the compact notation. Note that all derivatives are partial ones since the idea is to decouple $\boldsymbol{x}$ and $\Theta$ and realize their dependency by implying the constraint in Eq. (C4). For simplicity we neglect additional parameter bounds which otherwise would affect Eq. (C7). Taking the derivative in Eq. (C8) for each $\boldsymbol{\lambda}_i$ separately results in the model equations (Eq. 1) again. From (C6) we deduce

$$\frac{\partial J}{\partial \boldsymbol{x}_i} \frac{d\boldsymbol{x}_i}{d\Theta} = \boldsymbol{\lambda}_{i-1}^\top \frac{d\boldsymbol{x}_i}{d\Theta} - \boldsymbol{\lambda}_i^\top \frac{\partial M}{\partial \boldsymbol{x}_i} \frac{d\boldsymbol{x}_i}{d\Theta}, \quad i = 1, \dots, N_t$$

and apply Eq. (C3) to obtain

$$\frac{\partial J}{\partial \boldsymbol{x}_i} \frac{d\boldsymbol{x}_i}{d\Theta} = \boldsymbol{\lambda}_{i-1}^\top \frac{d\boldsymbol{x}_i}{d\Theta} - \boldsymbol{\lambda}_i^\top \left( \frac{d\boldsymbol{x}_{i+1}}{d\Theta} - \frac{\partial M}{\partial \Theta} \right), i = 1, \dots, N_t$$

where $\lambda_{N_t} = 0$ as above. Summing up gives

$$\sum_{i=1}^{N_t} \frac{\partial J}{\partial \boldsymbol{x}_i} \frac{d\boldsymbol{x}_i}{d\Theta} = \boldsymbol{\lambda}_0^\top \frac{d\boldsymbol{x}_1}{d\Theta} + \sum_{i=1}^{N_t} \boldsymbol{\lambda}_i^\top \frac{\partial M}{\partial \Theta}$$

$$= \boldsymbol{\lambda}_0^\top \frac{\partial M}{\partial \boldsymbol{x}_0} \frac{d\boldsymbol{x}_0}{d\Theta} + \sum_{i=0}^{N_t} \boldsymbol{\lambda}_i^\top \frac{\partial M}{\partial \Theta}$$

where we used again Eq. (C3) for $i = 1$. The first term includes the derivative of the initial values $\boldsymbol{x}_0$ w.r.t. the paraemters and in many cases will be zero. As result, the derivative of the cost can be computed from Eq. (C2) using the multiplier vector $\boldsymbol{\lambda}$, but without the tangent linear model. Note that the derivative of the model w.r.t. $\Theta$ in the sum is a *partial* derivative only, thus it does not include the derivaitive of the model variables, but only those of the model equations w.r.t. $\Theta$.

The multipliers $\boldsymbol{\lambda}_i$ satsify a time-stepping scheme themselves, but in reverse direction. Using the transposed form of (C6), we obtain

$$\boldsymbol{\lambda}_{i-1} = \left( \frac{\partial M}{\partial \boldsymbol{x}_i} \right)^\top \boldsymbol{\lambda}_i + \left( \frac{\partial J}{\partial \boldsymbol{x}_i} \right)^\top, \quad i = N_t, \dots, 1, \tag{C9}$$

with $\lambda_{N_t} = 0$ (see above) as starting point of the computation. Since here the transposed (or adjoint) of the linearisation of the model operator $M$ occurs, these equations are referred to as the adjoint equations or the adjoint model. Accordingly, the multipliers $\boldsymbol{\lambda}$ are also referred to as adjoint variables or adjoints. Given a model trajectory $\boldsymbol{x}$ and using Eq. (C9), the trajectory of the adjoints $\boldsymbol{\lambda}$ can be computed. It is crucial to note that both time-stepping schemes, for the variables $\boldsymbol{x}$ and the adjoints $\boldsymbol{\lambda}$, have opposite directions. This requires – except for the case of a linear model $M$ – the complete model trajectory to be stored or recomputed in order to compute $\boldsymbol{\lambda}$.

The adjoint model construction starting from a discretised extended Lagrange equation, Eq. (C5) can easily become extensive,

in particular when discretisations of advection and mixing are included in the model dynamics. Furthermore, even small changes in the equations can entail considerable additional efforts in updating the adjoint model equations. The application of automatic differentiation tools may therefore be better suited for cases where the ecosystem dynamical model is subject to regular modifications.

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
