# Peer review of "Reviews and syntheses: Parameter identification in marine planktonic ecosystem modelling"

_Biogeosciences, 2016_

## Referee Comment (RC1) · Anonymous Referee #1 · 28 Jul 2016

**Review comments for the manuscript: "Reviews and syntheses: Parameter identification in marine planktonic ecosystem modelling (bg-2016-242)" by Markus et al.**

General comments

The authors provide comprehensive reviews on parameter identification in marine ecosystem modeling with theoretical background, specifications of varied sources of uncertainty and examples of parameter identifications from literature. This paper also discusses the trade-offs between model simplicity and complexity coming from the number of represented processes, a number of parameters and variability of those parameters in space and time, as well as other important topics in modern ocean bio-

geochemical research. This paper is well-written and very informative. I recommend publication with only a few minor revisions.

Specific comments

In section 2.3, the authors provide two forms of probability densities and cost functions based on the statistical properties of errors. When one consider the lognormal distribution for errors, the error covariances in (11) and (12) should represent the uncertainty in logarithm space and be different from those in (9) and (10) (e.g., Fletcher, 2010; Song et al., 2012). If I am not wrong, the optimal solution for (12) represents the mode while that for (12) without $2\Sigma_{l=1}^{N\Theta} \log(\Theta_l)$ represents the median in lognormal probability density function. If this is right, I think that it would be useful if you mention mode and median in the sentence after (12).

Technical corrections

- page 2, line 28: there are two "be".

- page 3, line 22: typo for phytoplankton

- page 3, line 32: typo for assimilation

- page 8, line 32: typo for maximum

- page 13 line 2: typo for performing

- page 13 line 16: there needs space between the and "error model

- page 16 line 27: "An" alternative

- page 17 line 9: remove a space after "obtained"

- page 17 line 9: typo for bootstrap

- page 19 line 6: first order power "expansion"

- page 23 line 22: typo for described

- page 23 line 25: typo for retrieving

- page 24 line 9: typo for usually

- page 24 line 12: typo for usually

- page 24 line 24: typo for simulated

- page 25 line 3: typo for expressed

- page 25 line 23: within "the" zooplankton

- page 26 line 11: typo for mesozooplankton

- page 26 line 23: can thus "be" derived

- page 27 line 17: typo for correspondence

- page 30 line 17: typo for simpler

- page 30 line 19: "was" supported

- page 35 line 18: "has" been explored

- page 42 line 30: typo for artifacts

- page 44 line 24: typo for additional

- page 46 line 16: typo for approach

– page 46 line 18: typo for Bayesian

– page 48 line 11: "are" addressed

– page 50 Figure 1: The figure labels are not consistent wit the description in the caption.

**References**

Fletcher, S. J., 2010. Mixed Gaussian-lognormal four-dimensional data assimilation. Tellus A 62, 266–287.

Song, H., Edwards, C. A., Moore, A. M., Fiechter, J., 2012. Incremental four-dimensional variational data assimilation of positive-definite oceanic variables using a logarithm transformation. Ocean Modell. 54–55, 1–17.

---

## Referee Comment (RC2) · Anonymous Referee #2 · 4 Aug 2016

The manuscript discusses aspects of parameter identification, in particular the estimation of parameters in marine ecosystem models using data assimilation. Provided is a wide overview of topics like different error sources, variational estimation methods and the corresponding construction of cost functions, error models, and posterior parameter uncertainties. Further, typical model parameterizations are discussed as well as the aspect of cross-validation and model complexity, and space-time variability of parameters. As a further methodological aspect, emulator approaches are discussed. The manuscript also includes a discussion on some aspects of parameter estimation in large-scale biogeochemical ocean models. Overall, the manuscript considers a vast amount of topics and represents a quite comprehensive review of parameter estimation. The manuscript should fit well into the focus of the journal Biogeosciences. Despite the length o the manuscript, many topics are held very short. In this respect the

manuscript represents more an overview paper than a review. Nonetheless, it should be well suited as an entry point to the topic. However, there are several weaknesses in the presentation and the review of the methods is lacking the inclusion of ensemble-based methods. These issues are described in more detail below and should be resolved before a publication of the manuscript.

A major issue of the manuscript is the strongly varying degree of detail in the review of methods and results. While many findings are simply mentioned by citing the corresponding study, others are worked out as examples in more detail. For the examples, for which also figures are included, it is not clear why these cases have been selected. From the text it is not evident that the examples are scientifically particularly relevant. Given that the chosen examples in the different sections appear to be the scientific work of the lead authors of the respective section, I have the impression that the examples are merely chosen to promote the section lead-authors' work. As such the manuscript leaves an odd impression in that the authors most prominently point out their own work, while keeping results from other studies on a shorter descriptive level or even reducing it to a sole citation of a paper. This varying degree of detail should be corrected. For the readers it will be most helpful, if the examples are clearly chosen to illustrate the scientific most relevant aspects.

With respect to varying detail and the provision of examples, Section 9 is a particular case. One the one hand, the section sticks out in the manuscript because it is written in a rather lengthly style compared to the other sections. Here, I see a good potential to be more concise. On the other other hand, the abstract suggests that Section 9 contains a "survey ... to studies that approached parameter identification and global biogeochemical modeling" and that "Parameter estimation results will exemplify some of the advantages and remaining problems in optimizing global biogeochemical models". Unfortunately, this is only partly true. The "survey" is mainly reduced to 13 lines of text in the upper part of page 41. Directly afterwards the text discusses the general issue that the models need long integration times to reach equilibrium (Sec. 9.1) and

then focuses on the special topic of deep particle flux, which is started in Sec. 9.1 and worked out in quite some detail in Sec. 9.2. This is not marked as an example, but again given the many references to papers of the lead authors of this Section, one has the impression that the authors promote their own work instead of reviewing the state of the research. This is even more pronounced by the fact that from about line 13 of page 44 to the end of subsection 9.2 results from the not yet submitted study by Kriest et al. are discussed. Given that the described work is not yet published and hence not yet reviewed (and that I can not review it based on the incomplete description provided in the text) I can only recommend to reduce this part to one or two sentences stating the general findingd. The full length of the description in the manuscript is scientifically not justifiable until the study itself is published. Overall, I expected from the description in the abstract a comprehensive overview of results from parameter estimation in large-scale models. Unfortunately, this is clearly missing in section 9. Rewriting Section 9 in this respect would significantly improve the manuscript.

With regard to the methods, the study misses ensemble-based schemes, even though page 4, lines 26-27 state that an "overview of major DA aspects concerning parameter identification" is provided. While several studies that used ensemble-based methods are cited throughout the manuscript, the methodology is only shortly described in Section 7.1.3 on time-varying parameters where the method of "Sequential Importance Resampling (SIR)" (nowadays usually called "particle filter") is shortly explained. A manuscript with the motivation to provide a comprehensive overview of parameter estimation in ecosystem models is clearly incomplete when ensemble schemes are left out.

I have the impression that the authors intentionally left out a methodological description of ensemble-based methods (next to the particle filter also including methods based on the Kalman filters) because the authors do not use these methods and because the methods cannot ensure mass conservation. The mass conservation is already mentioned in lines 20-25 on page. The argumentation in the text that filtering methods

are "infringing" mass conservation, that mass conservation is relevant, and that one hence has to use methods that ensure mass conservation in the data assimilation, which "harmonise well" with corresponding methods in ocean state estimation, is part of a very old discussion, which is apparently followed emotionally (which is consistent with the words "infringing" and "harmonise" chosen by the authors). I'm not aware of any study that shows that the change of mass induces errors in the estimation of parameters or issues in the interpretation of the results. Even more, the methods could be used to estimate parameters alone, hence not changing the state directly such that the mass is conserved. My recommendation is that the authors simply avoid this discussion (unless they can provide scientific evidence) and revise the text accordingly.

While the main part of Section 10 summarizes the discussions of the manuscript, the sub-sections 10.1. and 10.2 do not fit with the main part. These sub-sections are not summaries a parts of the main text, nor do they show clear perspectives for the manuscripts' topic of parameter estimation. These aspects would better fit into the introduction section in order to discuss the different aspects of model parameterizations and the interplay of measurements, modeling and data assimilation.

Some more detailed Comments

Abstract; last sentence: I cannot see that the recommendation to find "...a good balance in the level of sophistication between mechanistic modelling and statistical data assimilation treatment..." is a result of the study. Either the authors should remove the statement or revise the text so that this statement results from reviewing the methods and application studies.

Page 6, lines 10-12: I have the impression that "weak constraint" and "strong constraint" are not general expressions used "In the geophysical community", but only in connection with data assimilation. Please consider changing the statement (Unfortunately, I cannot check the two cited books, a I don't have an easy access to them). BTW: Please cite books with providing a chapter or page range.

Page 10, line 1: Here "model discrepancy" and "model inadequacy" are mentioned. For readers it would be very helpful if the text could actually explain what these quantities are. The text states that this is part of an "important initiative" (page 9 line 32), but the description is not really more than mentioning the expressions and referring to two papers.

Page 17, line 31-32: The text states that "The prominance of MCMC methods for data assimilation is described by Rayner et al. (submitted)." Actually, while doing data assimilation for quite a while, I'm not aware of any "prominance" (BTW: this is a typo, it should be "prominence") of MCMC in this field. As data assimilation is usually concerned with high-dimensional models, the application of MCMC is not feasible. Please correct your statement.

Page 19, lines 1-4: Here, it is mentioned that "Two approaches to point-wise approximation of U are found in ... modeling studies" followed by mentioning the approaches. Unfortunately, references are missing for this statement. While in the following subsections some references are provided for methods based on the Jacobian, no paper is cited for the Hessian-based methods.

Page 29, lines 27 and 33: Please provide a reference for the "Akaike Information Criterion" as well as for the "weighted AIC" and the "Bayesian Information Criterion"

Page 35, line 15: It is stated: "It's flexibility could equally well be increased by increasing the size of the parameter vector, rather than allowing it to vary in time". It is unclear whether this statement is a result of some study (which would require a reference), or whether it is just speculation?

Page 42, lines 20-24: Here, the text states that "... three types of data are considered essential for model assessment and calibration" and then lists the data types. I wonder what is the scientific basis for this claim? Unfortunately no paper is cited. Please provide references to support this claim.

Page 47, line 20: The text mentions "dynamical and statistical emulators". Given that most readers are not familiar with these emulators, it would be helpful if each type is shortly explained.

For completeness of the review, please also consider the recent paper Simon et al. J. Mar. Syst. 152 (2015) 1-17, which is also concerned with parameter estimation in an ecosystem model.

---

## Referee Comment (RC3) · Anonymous Referee #3 · 5 Aug 2016

The authors present a summary review of parameter identification for biogeochemical ocean models. This is an important topic, and the manuscript provides a lot of material and some good description of the current state of research and outstanding issues. Yet, while some techniques and issues are discussed in great detail, very little information is provided about others. A less selective, more balanced view is needed to make the manuscript a proper synthesis of parameter identification in marine planktonic ecosystem modelling. My reservations are detailed below.

general comments:

The focus of the manuscript appears to be quite selective and sometimes arbitrary. Some methodologies are described in great detail, while only brief descriptions are given for others. DA techniques that are commonly used are not mentioned at all, for

example Kalman filter-based techniques. If the authors want to provide a synthesis of the current state of the research, these techniques need to be mentioned.

The authors seem to be very focused on mass conservation (is that the reason for not including many ensemble-based techniques?). I do not agree that this is the "one straw that biogeochemical modelers grasp at". In regional models, river inputs routinely break mass conservation, so why should DA techniques not be allowed to create updates to the mass inside the model domain if the data provides evidence for this? At the very least, the authors need to acknowledge that their view on mass conservation is not shared in the entire modelling community.

Given that model complexity and parameter identifiability play an important role in the manuscript (and rightly so), I wonder why there is not more focus on alternatives to the functional group approach. Some approaches, like the "optimal trade-off" are mentioned in section 10.1. Yet there are others which do not require manual parameter selection, like self-selective models (Follows et al., 2007) or the gene-centric approach (Reed et al., 2014), which groups plankton groups based on genetic information. These alternatives to the functional group approach should at least be mentioned, and mentioned earlier than section 10.1.

More focus should be given to the role different data types play in identifying model parameters. Sometimes the manuscript seems to suggest that all is needed is more data in order to identify more parameters, for example in the abstract: "data are often too sparse to constrain all model parameters". Yet more satellite chlorophyll data is probably not helpful in identifying many parameters, other data types and subsurface data are important as well. This is not just true for large-scale models (the issue of different data types is finally discussed in section 9.1 but only in regard to global models). In this context, the authors may also want to discuss the Bio-Argo program which could provide some much needed biogeochemical data products in the near future.

specific comments:

p2 l24: "of fecal pellets" perhaps change to "attached to fecal pellets"

p3 l4: "trophic levels like fish, which would be subject to changes in biomass on multi-annual rather than seasonal time scales": I would argue that the greater challenge with modelling fish is their behaviour and ability to swim, making it impossible to realistically simulate them by tracer variables.

p3 l5: "Every marine planktonic ecosystem model can thus be described as a simplification of the dynamics inherent to a system of nutrients, phytoplankton, zooplankton, detritus, dissolved organic matter, and bacteria". Apart from the fact that some phytoplankton are bacteria, I am wondering why they are listed here if (as stated above) they are often not resolved in models.

p3 l8: "Feedbacks from the ecosystem model ..." Maybe mention that feedback from physical to ecosystem model are essential.

p3 l16: "and the simulated N cycle was shown to already depend on the value assigned to a single parameter, namely the sinking velocity of detritus.": this sentence is not very clear, all other parameters do not affect the nitrogen cycle at all?

p4 l11: "availability of data thus places limitations on the number of model parameters whose values become identifiable." It is not just a numbers game, certain types of parameters may never be fully constrained by certain types of data, even if the model contains just a few parameters.

p4: l18: "Novel DA methods are predominantly devised for improving forecasts ..." While forecasting skill is often used to judge the quality of an assimilation system, many systems are used for hindcasting and creating reanalyses.

p10 l24: "It means that actually the cost function as given by..." this sentence is not very clear

p23 l5: "The third approach leads to more complex representations of growth limitation, as they..." Something may be missing here, the third approach is not described well and

the "they" should be an "it".

p25 l3: the summary of the loss terms here "Cell lysis, excretion and leakage are usually espressed ..." does not agree very well with the summary previously (p2 l22) "... removed by natural mortality (cell lysis due to starvation, senescence, and viral attack)..."

p27 l8: "right and left sides of top and bottom row" it would be useful to have labels (a) - (f) in Figure 2.

p27 l8: "It means that g_m can only be estimated in combination with". It means that estimates of g_m are dependent on the values of the other parameters. If we are certain what the values of phi_agg and gamma_C are, we can still estimate g_m.

p27 l9: "If g_m remains fixed, we do not find such strong collinearity expressed between gamma_C and phi_agg": I would rephrase this, since in this particular experiment only two parameters are varied, i.e. remove the "If g_m remains fixed".

p40 l29: "To account for the lateral flux information was helpful contributed strongly to the emulator accuracy.": something is missing here

Section 9: After discussing methodology, why are global biogeochemical ocean models introduced now? I would move most of this section to the introduction.

technical corrections:

Both "Fig." and "Figure" are used to reference figures.

p15 l17: close parentheses.

p18 l23: "(Fisher, 1922) see also (e.g., Fisher, 1934 ..." move "see also" into parentheses as well.

p36 l30: Citations are now ordered by name, before it was by date.

---

## Referee Comment (RC4) · Anonymous Referee #4 · 8 Aug 2016

In this manuscript the authors provide an overview of various topics of consideration for the implementation and evaluation of marine planktonic ecosystem models, focused primarily on assessment of errors and uncertainty, techniques for optimization of model parameters, and model development considerations.

The overall effect of this manuscript is a thorough discussion of the diverse approaches to parameterization, describing mainstream tactics, recent advancements, and possible new directions. Marine ecosystem model parameterization is a wide and still rapidly developing field, and this synthesis of major methodologies addresses a growing need for such resources for both veterans and those new to the field. The manuscript is well written and directly relevant to the scope of Biogeosciences, and I recommend publication with some revision.

[Figure]

General Comments:

One of the main objectives of this paper is to "provide support to readers" who strive to understand the diversity of parameter estimation approaches despite the variety of terminologies and notations used. Because of the broad range of topics covered throughout even this article, I feel that including a glossary of symbols (with or without short descriptions and page references) used in all equations, perhaps as an appendix, would be a helpful guide to readers.

To make the manuscript more understandable by novices in the field, the authors may want to begin the review with Section 5 "Typical parameterisations of plankton models and their parameters" rather than starting with error models. Given that the title of this article is parameter identification in plankton modeling, it seems strange that theoretical discussion of error models comes before any discussion of actual ecosystem parameterization attempts.

The manuscript is quite long and certain sections contain significantly more detail than others. As a result, the manuscript would be improved if the sections could be made more consistent in terms of their degree of detail.

Specific Comments:

In certain places, e.g. section 2, the article assumes a familiarity of terms, which may benefit from greater introductory explanation. In particular, I found the description of "kinematic model errors" and "dynamical model errors" on page 6 somewhat lacking. If these are the terms commonly used, I would suggest including reference to a more thorough explanation, otherwise I would suggest providing a greater introductory explanation since you use these terms repeatedly. Likewise, at the beginning of section 4.1, a few introductory remarks describing the authors' specific intended meanings of "confidence" and "credible" would be helpful when transitioning into this section.

Section 2.2 would likely benefit from being split into a few subsections.

[Figure]

Should section 4.2 be in section 4? Should section 4.3 come before 4.2?

Page 26, line 2: Can you mention one or two advantages of mesocosm data to complement your mention of drawbacks in the next sentence?

Page 40, sect 8.3: Can this section include a mention of where the authors think the use of combined emulators is headed? What are the hurdles to overcome?

Page 45, line 18: After stating that an appropriate treatment of uncertainties for Earth system models is critical, it seems as if this section is going to go into depth on that subject, but it is a cursory overview. I would suggest mentioning here that a full treatment of uncertainties in Earth system models is beyond the scope of this article.

Page 48, section 10.1: It seems that this paragraph is not focused on what the section title states. Half of the paragraph is about a novel thermodynamically inspired ecosystem model. Can the title be reworded, or the text reworded as to relate more to "keeping number of free parameters low". Also, in the title, what does "grasping" mean in reference to complexity?

Technical corrections:

Page 1, lines 9 - 16: Tense consistency: these lines begin in the present tense (e.g. "we explore how. . .") and then use the future tense (e.g. "complexity will be covered. . .")

Page 3, line 15 "such a marine plankton ecosystem"

Page 4, line 29: "we like to"

Page 5, line 1: no hyphen in "branch off"

Page 5, line 8: comma after "Sect. 10"

Page 7, line 25: sigma, used in eq. 5, should be defined here.

Page 8, line 32: spelling of "maximum"

Page 9, line 13: "We will see an example of this below". Can you please indicate where

the example is shown?

Page 11, lines 5 - 15: great discussion of terminology.

Page 12, line 21: "In many cases when sampling..."

Page 14, line 17-18: Can the sentence beginning with "One must only..." be reworded? It is unclear as it currently stands.

Page 15, line 35: Which NPZ model is this?

Page 16, line 10" "...since they may not be sufficient"

Page 16, line 31: Should this read "Multimodel inference would allow the grazing parameters..." instead of "...allow the P-I parameter values"?

Page 17, line 13: comma after "ML estimator"

Page 18, lines 5: I would suggest changing "can be" to "are" for consistency with the next sentence.

Page 18, line 28: "level", not plural

Page 18, eq. 14: Is 'T' the truth operator from eq. 2? If not, can you state what this represents.

Page 19, line 12: "so that" should be "that"

Page 22, line 18: replace "it"

Page 23, line 10 - 19: check verb tense.

Page 23, line 11-12: Include "for example", since alphas have been derived from many other studies as well.

Page 23, line 26: "has" instead of "had"

Page 23, line 27: When did this discussion regain importance?

Page 25, line 23: "a" instead of "a the"

Page 26, line 23 "...can thus be derived..."

Page 28, line 25: no comma after "compensatory"

Page 29, line 3: "simulation", not plural

Page 29, line 16: "Similarly, an [any?] extra flexibility?"

Page 30, line 15: "has" instead of "have"

Page 32, line 12: "...with an independent..."

Page 36, line 20: "...thousands of years..."

Page 37, line 8: Can this sentence be reworded or removed? It is unclear how this connects with the previous sentence.

Page 37, line 11: "to the original"

Page 37, line 17: "has" instead of "as"

Page 40, line 29-30: This is a sentence fragment.

Page 41, line 22: Is "close to the heart" appropriate?

Page 42, line 2: "...it remains"; no comma after "investigated"

Page 42, line 25: "." instead of ":"

Page 43, line 6: no comma after "value"; "the" instead of "that"

Page 43, line 7: "or" instead of "and"

Page 43, line 9: "which" instead of "that"

Page 44, line 26: extra comma

Page 46, line 5: extra space after "30"

Page 46, line 24: comma after "'ignorance' prior"

Page 47, line 25: are these quotes necessary around "emulator"?

Page 47, line 31: "and" instead of comma after data sets

―――――――――――――――――――

---

## Author Comment (AC1) · 29 Oct 2016

**Specific comments by Referee #1**

***Comment* 1**: In section 2.3, the authors provide two forms of probability densities and cost functions based on the statistical properties of errors. When one consider the lognormal distribution for errors, the error covariances in (11) and (12) should represent the uncertainty in logarithm space and be different from those in (9) and (10) (e.g., Fletcher, 2010; Song et al., 2012). If I am not wrong, the optimal solution for (12) represents the mode while that for (12) without $2 \sum_{l=1}^{N_{\Theta}} \log(\Theta_l)$ represents the median in lognormal probability density function. If this is right, I think that it would be useful if you mention mode and median in the sentence after (12).

[Figure]

*Author's response*: We have corrected the notation to also include tildes on residuals
and covariances evaluated on the transformed scale. Because we introduced addi-
tional equations to the Theoretical Background section, the equation numbers have
changed (equation 11 now is 14 and equation 12 is 15). The lognormal equations now
read:

"Alternatively, nonnegativity constraints on the variables and parameters may lead us
to prefer the lognormal observational error model. Likewise, we can assume lognormal
priors for the parameters. In this case the posterior density becomes:

$$p(\Theta \mid \vec{y}) \propto \frac{1}{\sqrt{(2\pi)^{N_y} \det \tilde{\mathbf{R}}} \prod_j y_j} \cdot \exp\left[-\frac{1}{2} \tilde{\vec{d}}^T \tilde{\mathbf{R}}^{-1} \tilde{\vec{d}}\right]$$

$$\cdot \frac{1}{\sqrt{(2\pi)^{N_\Theta} \det \tilde{\mathbf{B}}} \prod_l \Theta_l} \cdot \exp\left[-\frac{1}{2} \tilde{\boldsymbol{\Delta}}_\Theta^T \tilde{\mathbf{B}}^{-1} \tilde{\boldsymbol{\Delta}}_\Theta\right] \tag{14}$$

where the data-model residuals and parameter corrections on the transformed scale
are defined by $\tilde{\vec{d}} = \log(\vec{y}) - \log\left(H\left(\vec{x}\right)\right) + \dfrac{\tilde{\boldsymbol{\sigma}}^2}{2}$ and $\tilde{\boldsymbol{\Delta}}_\Theta = \log(\Theta) - \log(\Theta^b) + \dfrac{(\tilde{\boldsymbol{\sigma}}^b)^2}{2}$. A
MAP estimator of $\Theta$ is then obtained by minimising:

$$J(\Theta) = \tilde{\vec{d}}^T \tilde{\mathbf{R}}^{-1} \tilde{\vec{d}} + 2 \sum_{l=1}^{N_\Theta} \log(\Theta_l) + \tilde{\boldsymbol{\Delta}}_\Theta^T \tilde{\mathbf{B}}^{-1} \tilde{\boldsymbol{\Delta}}_\Theta \tag{15}$$

"

The optimal solution for Eq. (15) does indeed represent a posterior mode for $\Theta$, and
maximising Eq. (15) without the second term will indeed yield a posterior median esti-
mate for $\Theta$ (and also $\log \Theta$, since quantiles are invariant under scale transformation).

We have modified the text to:

" The MAP or posterior mode estimator of $\log(\Theta)$ is equivalent here to the posterior median estimate and is obtained by maximising $p(\log(\Theta) \mid \vec{y})$. This leads to a cost function given by Eq. (15) without the second term, $2\sum_{l=1}^{N_\Theta}\log(\Theta_l)$ (cf. , Fletcher, 2010)."

We are particularly thankful to Referee #1 for spotting grammatical- and typing errors. All corrections listed by Referee #1 have been applied.

---

## Author Comment (AC2) · 29 Oct 2016

We appreciate the time spent and the comments provided by Referee #2. Major concerns raised by Referee #2 were respected and we introduced considerable changes to our manuscript.

**Comment* 1 *by Referee* #2:**

A major issue of the manuscript is the strongly varying degree of detail in the review of methods and results. While many findings are simply mentioned by citing the corresponding study, others are worked out as examples in more detail. For the examples, for which also figures are included, it is not clear why these cases have been selected. From the text it is not evident that the examples are scientifically

particularly relevant. Given that the chosen examples in the different sections appear to be the scientific work of the lead authors of the respective section, I have the impression that the examples are merely chosen to promote the section lead-authors' work. As such the manuscript leaves an odd impression in that the authors most prominently point out their own work, while keeping results from other studies on a shorter descriptive level or even reducing it to a sole citation of a paper. This varying degree of detail should be corrected. For the readers it will be most helpful, if the examples are clearly chosen to illustrate the scientific most relevant aspects.

*Author's response*: Our aim was to provide examples that describe and stress particularly relevant aspects. In practice, the lead author (M. Schartau) asked each section to include an example figure that both he and the section author(s) considered to be particularly relevant/illustrative. In most cases these examples have come from the sections authors' work or extensions thereof. This is understandable since each section author is a specialist in that area and is most familiar with his or her own work. However, we appreciate that this may have led an unfortunate impression of promoting our own work, and we thank the reviewer for pointing this out. To counteract this impression we have made a number of changes to improve balance in coverage of different approaches and to maximise breadth of relevance in the figures. However, we feel it is acceptable for section authors to use examples from their own work since it is for these that they have best understanding and control. Here is a summary of the figures in the revised manuscript:

1) We have moved the original Fig. (1) with the example of a variable lag fit (VLF) to the Appendix (Fig. A1) and substituted it with a figure from the study of Simon and Bertino (2012, Journal of Marine Systems, 89, 1-18). This figure is a nice example of the improved asymptotic behaviour of a deterministic Ensemble Kalman Filter (DEnKF) when using log-transformed observations and model results to realise the analysis

step. Ehouarn Simon and Laurent Bertino kindly provided their results so that we could redraw the figure. We will send a request to Elsevier for using the redrawn figure. The figure is referred to in Sect. (4) about Error Models. The new Fig. (1) (send upon request) is based on results from a sequential, ensemble based, data assimilation (DA) approach, which should further improve balance with respect to ensemble based DA methods.

2) In Sect. (7) we address space-time variations in model parameter estimates and we find it appropriate to include a figure (Fig. 2 of Losa et al., 2006), based on results from Losa et al. (2004). It is prominent and illustrative example of variable parameter values in the North Atlantic. We will send a request to Elsevier for using their figure. In the text we refer to this figure (Fig. 5) as follows:

Nevertheless, Losa et al. (2004) were able to document the plausibility of their posterior photosynthesis parameter values for the maximum phytoplankton growth rate ($\mu_m$ in Sect. 3.1) and intial slope of the P-I curve ($\alpha_{phot}$ in Sect. 3.3) by comparison with observational estimates of Platt et al. (1991). Six parameters were optimised in all and the posterior parameter fields were cross-validated in a 3D version of their model by comparing the output with an independent SeaWiFS chlorophyll data from 1997-2003 (Losa et al., 2006). The spatially-varying parameter set of Losa et al. (2004), obtained by assimilating Coastal Zone Color Scanner (CZCS) data for the period 1979-1985, was interpolated and extrapolated onto the spatial grid of the 3D model as shown for the two parameters relevant for phytoplankton growth, $\mu_m$ and $\alpha_{phot}$ respectively (Fig. 5).

3) We eliminated the orginal Fig. (7) in Sect. (9) on the probability densities of the climatological phosphate, nitrate and oxygen data. Instead we put more emphasis on explaining the relevance of Fig. (8) with the projections of the parameter-cost function manifold.

***Comment* 2 *by Referee* #2**:
With respect to varying detail and the provision of examples, Section 9 is a particular case. One the one hand, the section sticks out in the manuscript because it is written in a rather lengthly style compared to the other sections. Here, I see a good potential to be more concise.

***Author's response***: Sect. (9) is revised entirely. The text has already been considerably reduced by removing details and by skipping the original Fig. (7) that showed probability densities of nutrient concentrations. The detailed illustrative example in Fig. (8) is kept. This example nicely connects aspects explained in new Sect. (3) (former Sect. 5, Typical parameterisations of plankton models and their parameters), recalled with an example in new Sect. (5) (former Sect. 4, Posterior parameter uncertainties).

***Comment* 3 *by Referee* #2**:
With regard to the methods, the study misses ensemble-based schemes, even though page 4, lines 26-27 state that an "overview of major DA aspects concerning parameter identification" is provided. While several studies that used ensemble-based methods are cited throughout the manuscript, the methodology is only shortly described in Section 7.1.3 on time-varying parameters where the method of "Sequential Importance Resampling (SIR)" (nowadays usually called "particle filter") is shortly explained. A manuscript with the motivation to provide a comprehensive overview of parameter estimation in ecosystem models is clearly incomplete when ensemble schemes are left out.

***Author's response***: Ensemble-based schemes were in fact briefly referred to in

the Theoretical Background section: "Similar concerns apply to parameter estimation for stochastic dynamical models, where fully Bayesian approaches appear to be favoured using computational strategies based on sequential Monte Carlo methods (van Leeuwen, 2009; Jones et al., 2010; Dowd, 2011; Doucet and Robert, 2013; Dowd et al., 2014). "

but we admit that this was not adequate coverage given the importance of these methods. We have therefore replaced this sentence with a new subsection "Sequential methods" (Sect. 2.2.2) in the Theoretical Background, which expands on the methodological material previously included in Sect. (7.1.3). This new subsection (2.2.2) reads as follows:

"In some problems, assimilating all the data at once from all available sampling times can be computationally impractical. This is particularly likely for models with stochastic dynamics ($\eta \neq 0$ in Eq. 1), if the data are clustered in time, or if model states need to be repeatedly updated as new data come in. In such cases a sequential approach can be expedient. The basic idea is to break the large integration problem defined by Eq. (7) into a number of smaller problems by sequentially assimilating observations in subsets defined by sampling time. The method comprises a consecutive sequence of two major steps, a forecast- and an analysis step respectively. If the sequential approximation or 'filter' is accurate, it should approximate the posterior distribution defined by Eqs. (6 and 7), when all data have been assimilated by the end of the assimilation period. To see how this works, suppose we know the probability density $p(\vec{x}_j^t \mid \vec{y}_{1:j}, \Theta)$ of the true state at sampling time $t_j$ (possibly an initial condition) for a given value of the uncertain parameters $\Theta$ and given all the previously assimilated observations $\vec{y}_{1:j}$ (possibly null). The probability density at sampling time $t_{j+1}$ is given by the forecast density:

$$p(\vec{x}_{j+1}^t \mid \vec{y}_{1:j}, \Theta) = \int p(\vec{x}_{j+1}^t \mid \vec{x}_j^t, \Theta) \cdot p(\vec{x}_j^t \mid \vec{y}_{1:j}, \Theta)\, \mathrm{d}\vec{x}_j^t \qquad (8)$$

In general this integral can be approximated by an ensemble of Monte Carlo simulations, sampling an initial condition from $p(\vec{x}_{j+1}^t \mid \vec{y}_{1:j}, \Theta)$ and then running the model to

the next sampling time $t_{j+1}$ (possibly including stochastic dynamical noise, and possibly accounting for kinematic model error). Next, in the analysis step, the new observations are assimilated by applying Bayes' theorem:

$$p(\vec{x}_{j+1}^t \mid \vec{y}_{1:(j+1)}, \Theta) \propto p(\vec{y}_{j+1} \mid \vec{x}_{j+1}^t, \Theta) \cdot p(\vec{x}_{j+1}^t \mid \vec{y}_{1:j}, \Theta), \tag{9}$$

which again can be approximated e.g. by Monte Carlo sampling. The forecast and analysis steps can then be repeated until all the data are assimilated. A seldom-discussed assumption here is the conditional independence of the observations, allowing us to write $p(\vec{y}_{j+1} \mid \vec{x}_{j+1}^t, \Theta)$ instead of $p(\vec{y}_{j+1} \mid \vec{x}_{j+1}^t, \vec{y}_{1:j}, \Theta)$ in Eq. (9). This amounts to assuming that the observational errors are independent between sampling times (Evensen, 2009), which may not be strictly true if sampling is frequent and if there is a noticeable contribution from representativeness/undersampling, or from errors in conversion factors (see Sect. 2.1.3).

Once the predictive filtering densities $p(\vec{x}_{j+1}^t \mid \vec{y}_{1:j}, \Theta)$ have been approximated for all sampling times ($t_j$ with $j = 1, \ldots, N_t$), these can be used to approximate the likelihood in Eq. (7), since:

$$\begin{aligned} p(\vec{y} \mid \Theta) &= \prod_{j=1}^{N_t} p(\vec{y}_j \mid \vec{y}_{1:j-1}, \Theta) \\ &= \prod_{j=1}^{N_t} \int p(\vec{y}_j \mid \vec{x}_j^t, \vec{y}_{1:j-1}, \Theta) \cdot p(\vec{x}_j^t \mid \vec{y}_{1:j-1}, \Theta) \, \mathrm{d}\vec{x}_j^t \\ &= \prod_{j=1}^{N_t} \int p(\vec{y}_j \mid \vec{x}_j^t, \Theta) \cdot p(\vec{x}_j^t \mid \vec{y}_{1:(j-1)}, \Theta) \, \mathrm{d}\vec{x}_j^t \end{aligned} \tag{10}$$

For $j{=}1$ in Eq. (10) we have a set of zero members and $p(\vec{y}_j \mid \vec{y}_{1:j-1}, \Theta) = p(\vec{y}_1 \mid \Theta)$. In the third line of Eq. (10) again some conditional independence of the observations is assumed and the final integral can in general be approximated using the predictive

ensembles (see Jones et al., 2010; Dowd, 2011; Dowd et. al., 2014). This procedure can be repeated for different values of $\Theta$ and combined with Eq. (6) to assess posterior probability. Alternatively, $p(\Theta \mid \vec{y})$ can be calculated from a single application of the filter using a 'state augmentation' approach whereby the parameters $\Theta$ are appended to the vector $\vec{x}$ as additional state variables with zero dynamics. In practice, random parameter noise may need to be added to avoid filter degeneracy, such that this approach may be considered a separate estimation method (Dowd, 2011). However, if such ad hoc noise can be avoided, or if the parameters are in fact assumed to vary stochastically, then the augmented-state filter at the end of the assimilation interval should approximate the theoretical Bayesian posterior for this time. For other times, a 'smoother' algorithm would be required. A further benefit of the augmented-state filter is that the parameter estimates for intermediate time periods may show temporal patterns that expose deficiencies in the model formulation and provide useful information for model development (e.g., Losa et al., 2003).

The various types of filter differ essentially in terms of how the integrals in Eqs. (8) and (9) are approximated. Particle filters (van Leeuwen, 2009) use Monte Carlo sampling for both steps while the Ensemble Kalman Filter (Evensen, 2003; Evensen, 2009) uses Gaussian and linear approximations for the analysis step, enabling the use of smaller ensembles but at the cost of lower accuracy in strongly nonlinear/non-Gaussian problems. The (Extended) Kalman Filter applies when the model dynamics are (quasi-) linear and both model and observational errors are Gaussian. These conditions allow both integrals to be evaluated analytically, but appear to be rarely applicable to parameter estimation in marine ecosystem models. For reviews of sequential approaches the reader is referred to Dowd et al. (2014) for marine biogeochemical modelling and to Bertino et al. (2003) for oceanography in general. "

Note that we do not aim to provide technical details on the various filters, partly because these are already discussed in other reviews (to which we have directed the

interested reader), and partly because we want the Theoretical Background to focus on models and general methods, not algorithms and techniques.

**Comment 4 by Referee #2**:
I have the impression that the authors intentionally left out a methodological description of ensemble-based methods (next to the particle filter also including methods based on the Kalman filters) because the authors do not use these methods and because the methods cannot ensure mass conservation. The mass conservation is already mentioned in lines 20-25 on page. The argumentation in the text that filtering methods are "infringing" mass conservation, that mass conservation is relevant, and that one hence has to use methods that ensure mass conservation in the data assimilation, which "harmonise well" with corresponding methods in ocean state estimation, is part of a very old discussion, which is apparently followed emotionally (which is consistent with the words "infringing" and "harmonise" chosen by the authors). I'm not aware of any study that shows that the change of mass induces errors in the estimation of parameters or issues in the interpretation of the results. Even more, the methods could be used to estimate parameters alone, hence not changing the state directly such that the mass is conserved. My recommendation is that the authors simply avoid this discussion (unless they can provide scientific evidence) and revise the text accordingly.

**Author's response**: This discussion, although old, is still of importance. Whether mass balance is achieved or not is relevant, in particular when communicating model results, other than simulated fields of chlorophyll *a* concentrations, to biological oceanographers and marine biogeochemists. The property of mass conservation is more fundamental than any detail in the parameterisations. It is not our intention to evoke any emotional commotion and have therefore rephrased respective sentences. In the second paragraph of the introduction we introduced the following change:
"So far no fundamental ecophysiological principle has been further exacted beyond

the conservation of mass. Whether a balanced mass budget needs to be achieved depends on the scientific problem addressed. Some ecosystem model applications may not critically depend on mass conservation, e.g. when simulating plankton growth to act as food source in regional simulations of fish stock size and recruitment. In biogeochemical models the conservation of mass can be essential, in particular for large-scale or global ocean simulations. A consistent theme running through most ecosystem models is the determination of mass flux of certain biologically important elements, such as nitrogen, phosphorus, iron and carbon (N, P, Fe and C)."

Furthermore, we revised the subsection (Sect. 1.4, Inferences from data assimilation): "Much of the literature on DA in oceanography is focussed on state estimation (e.g., Allen et al., 2003; Natvik and Evensen, 2003; Dowd 2007; Nerger and Gregg, 2008; van Leeuwen, 2010). In these studies, the primary objective is to improve hindcasts, nowcasts, or forecasts of time-dependent variables such as chlorophyll *a* (Chl*a*). However, many of the DA methods originally developed for state estimation have more recently been adapted to estimate static parameters, especially for stochastic models where random noise is injected into the model dynamics. Stochastic noise offers a plausible way to represent model error, but it should be noted that it can lead to violations of mass conservation unless it is injected in certain ways (e.g. by perturbing growth rate parameters). Deterministic plankton ecosystem models guarantee mass conservation and have a longer tradition in parameter estimation for marine ecosystem models, although they imply a less explicit treatment of model error. To identify and gradually eliminate model deficiencies it can be helpful to analyse model state and flux estimates while mass conservation is imposed as a strong constraint. The optimisation of only parameter values assures that simulation results remain dynamically and ecologically consistent, which is comparable with those DA approaches in physical oceanography that produce dynamically and kinematically consistent solutions of ocean circulation (e.g., Wunsch and Heimbach, 2007; Wunsch et al., 2009). "

***Comment* 5 *by Referee* #2**:
While the main part of Section 10 summarizes the discussions of the manuscript, the sub-sections 10.1. and 10.2 do not fit with the main part. These sub-sections are not summaries a parts of the main text, nor do they show clear perspectives for the manuscripts' topic of parameter estimation. These aspects would better fit into the introduction section in order to discuss the different aspects of model parameterizations and the interplay of measurements, modeling and data assimilation.

***Author's response***: The sections (10.1 and 10.2) would not fit to the introduction section. Section (10.1) clearly refers to perspectives and it should reflect current tendencies in the development of planktonic ecosystem models. We actually complemented Sect. (10.1), in response to comments by Referee 3. In Sect. (10.2) we summerise our major impression after literature search and after reading many papers that covered diverse aspects. One important take home message is that we found, on the one hand, many studies (with DA methods applied for parameter optimisation) where biological aspects (e.g. basic model assumptions) remained undifferentiated (undiscussed). On the other hand, a series of biologically motivated studies did not consider aspects of parameter identifiability, let alone of DA. We think we have expressed this in Sect. (10.2) and stressed the need to find a good balance between the different scientific communities to which we refer to in our manuscript.

—

***Specific comment* 1 *by Referee* #2**:
Abstract; last sentence: I cannot see that the recommendation to find "...a good

balance in the level of sophistication between mechanistic modelling and statistical data assimilation treatment..." is a result of the study. Either the authors should remove the statement or revise the text so that this statement results from reviewing the methods and application studies.

*Author's response*: The implication here is that there is frequently an imbalance in the level of sophistication in these two areas. This was a general impression that we gained from reading the literature, and we feel it could be helpful to report this impression to readers. Probably this imbalancedness is driven by the fact that it is easier to publish a paper and write a successful grant proposal if it purports to be "cutting-edge" and "state-or-the-art" in some particular way, rather than putting a balanced level of effort into *all* methodological aspects. We therefore do not want to follow the Referee's suggestion and leave the statement as it is.

*Specific comment* **2** *by Referee* **#2**:
Page 6, lines 10-12: I have the impression that "weak constraint" and "strong constraint" are not general expressions used "In the geophysical community", but only in connection with data assimilation. Please consider changing the statement (Unfortunately, I cannot check the two cited books, a I don't have an easy access to them).

*Author's response*: Corrected to "In the geophysical data assimilation community". The first citation in this instance is a paper: Sasaki, Y.: Some basic formalisms in numerical variational analysis, Monthly Weather Review, 98, 875–883, 1970. We have added chapter/page ranges to the book references where possible.

*Specific comment* **3** *by Referee* **#2**:

Page 10, line 1: Here "model discrepancy" and "model inadequacy" are mentioned. For readers it would be very helpful if the text could actually explain what these quantities are. The text states that this is part of an "important initiative" (page 9 line 32), but the description is not really more than mentioning the expressions and referring to two papers.

*Author's response*: This paragraph has been expanded to:

"Another important initiative is the estimation of hyperparameters of the kinematic error model along with the ecosystem parameters (Arhonditsis et al., 2008). The posterior of the kinematic model error provides an estimate of the model discrepancy, introduced by Kennedy and O'Hagan (2001) and originally referred to as model inadequacy. The model discrepancy is defined as the model error for the "true" values of the model parameters, i.e. the unknown values of the parameters for which the model best represents $\vec{x}^t$. Estimates of model discrepancies may thus provide useful diagnostics for model skill assessment and development. "

*Specific comment* **4** *by Referee* **#2**:
Page 17, line 31-32: The text states that "The prominance of MCMC methods for data assimilation is described by Rayner et al. (submitted)." Actually, while doing data assimilation for quite a while, I'm not aware of any "prominance" (BTW: this is a typo, it should be "prominence") of MCMC in this field. As data assimilation is usually concerned with high-dimensional models, the application of MCMC is not feasible. Please correct your statement.

*Author's response*: DA is not exclusively concerned with high-dimensional models and MCMC methods have been and still are commonly applied. We admit that the statement is awkward and the reference is not exclusive. We suggest to simply remove the sentence.

***Specific comment*** **5** ***by Referee*** **#2**:
Page 19, lines 1-4: Here, it is mentioned that "Two approaches to point-wise approxi-mation of U are found in ... modeling studies" followed by mentioning the approaches. Unfortunately, references are missing for this statement. While in the following subsections some references are provided for methods based on the Jacobian, no paper is cited for the Hessian-based methods.

***Author's response***: Yes, some references would be helpful. We added two useful references:
" The matrix $\mathcal{H}_\Theta$ is the Hessian whose elements are second derivatives of $J(\Theta)$ with respect to the parameters (e.g., Tziperman and Thacker, 1989; Matear 1995): ... "

***Specific comment*** **6** ***by Referee*** **#2**:
Page 29, lines 27 and 33: Please provide a reference for the "Akaike Information Criterion" as well as for the "weighted AIC" and the "Bayesian Information Criterion"

***Author's response***: We included the original reference (Akaike, 1973) for AIC, refer to the work of Johnson and Omland (2004) for the log-likelihood approximation based on residual sum of squares, and use Burnham and Anderson (2004) as an example reference for the bias-variance trade-off:
"One of the simplest techniques (in terms of its applicability), is the Akaike Information Criterion (AIC, Akaike, 1973). The AIC considers two opposing terms corresponding to the maximum log-likelihood of the parameters given the data ($\ln[L(\widehat{\Theta} \mid \vec{y})]$, measuring model data misfit) and a bias-correction factor, that increases with the number of free parameters ($N_\Theta$).

$$AIC = -2\,\ln\left[L\left(\widehat{\Theta}_p \mid \vec{y}\right)\right] + 2N_\Theta \tag{11}$$

Note that for a model fitted by least-squares, the log-likelihood can be approximated by the residual sum of squares (RSS), following Johnson and Omland (2004): $\ln[L(\widehat{\Theta}_p \mid \vec{y})] \approx -N_y/2 \cdot \ln(RSS/N_y)$, with $N_y$ being the total number of observations. The AIC, and alternative techniques (weighted AIC, or Bayesian Information Criterion, BIC), seek to quantify the trade-off between bias and variance (e.g., Burnham and Anderson, 2004). "

***Specific comment 7 by Referee #2***:
Page 35, line 15: It is stated: "It's flexibility could equally well be increased by increasing the size of the parameter vector, rather than allowing it to vary in time". It is unclear whether this statement is a result of some study (which would require a reference), or whether it is just speculation?

***Author's response***: The statement queried is not the result of a study but was intended simply as a reference to a general concept that is covered in other parts of the text. It is not needed here specifically and has been removed.

***Specific comment 8 by Referee #2***:
Page 42, lines 20-24: Here, the text states that "... three types of data are considered essential for model assessment and calibration" and then lists the data types. I wonder what is the scientific basis for this claim? Unfortunately no paper is cited. Please provide references to support this claim.

***Author's response***: We thank the referee for identifying this mistake. Here, the verbalism is inappropriate and we realised that it does not reflect what we intend to state. There is no such "claim". We suggest to revise the entire Sect. (9) and to correct this statement in a new subsection (9.2 Data availability):

"In regard to the ocean's key role in global carbon cycling and hence for the climate system, three different types of data can be considered for model assessment and calibration: 1) data of dissolved inorganic tracers, e.g. distributions of nutrients, oxygen, alkalinity and dissolved inorganic carbon, 2) measurements or data products of rates, e.g. of planktonic primary- or net community production, and 3) observations of the gravitational flux of organic particles to the ocean interior, transporting particulate organic matter through the water column. "

***Specific comment* 9 *by Referee* #2**:
Page 47, line 20: The text mentions "dynamical and statistical emulators". Given that most readers are not familiar with these emulators, it would be helpful if each type is shortly explained.

***Author's response***: The "dynamical" emulator was/is already described in Sect. (8.1) and the "statistical" emulator was/is explained in Sect. (8.2). We see no need to again explain the differences between the two in Sect. (10).

***Specific comment* 10 *by Referee* #2**:
For completeness of the review, please also consider the recent paper Simon et al. J. Mar. Syst. 152 (2015) 1-17, which is also concerned with parameter estimation in an ecosystem model.

***Author's response***: We thank the referee for alerting us to this useful paper. It is now cited in Sect. (7).

---

## Author Comment (AC3) · 31 Oct 2016

We thank Referee #3 for reviewing our manuscript. According to the comments provided by Referee #3 we have learned that we should mention filter techniques and also describe aspects of sequential DA approaches. We have done so and, in the end, we introduced changes that correspond to a major revision of our manuscript. The comments provided by Referee 3 are appreciated and we think that the revised manuscript has improved considerably.

**Major comments by Referee #3**

*Comment* 1: The focus of the manuscript appears to be quite selective and sometimes arbitrary. Some methodologies are described in great detail, while only brief descriptions are given for others. DA techniques that are commonly used are not mentioned at all, for example Kalman filter-based techniques. If the authors want to provide a synthesis of the current state of the research, these techniques need to be mentioned.

*Author's response*:
During the preparation of our manuscript we concentrated on achieving a balance between biological aspects, problems of parameter identification, and basic DA methodological considerations. It was a difficult and extensive process to elaborate a meaningful structure and work out the content of respective sections of the submitted version of our manuscript. All authors' contributions were mutually revised. We regret that Referee #3 has the impression that the focus of the manuscript is arbitrary and selective. We do not share this view, but we have realised some structural weaknesses and have introduced major changes to the manuscript. Apart from shifting the original Sect. (5) to Sect. (3) (following a suggestion of Referee #4), major changes were done in Sect. (2) and in Sect. (9).

We agree that sequential DA methods, like the ensemble Kalman filter, were underrepresented. We now restructured and extended the Theoretical Background section (Sect. 2) and included two subsections under Sect. (2.2 Estimation methods): one about sequential methods (Sect. 2.2.2) and another about variational methods (Sect. 2.2.3). This way we have introduced an explicit representation of sequential DA methods.

We also included two new figures from the publications of Simon and Bertino (2012) (based on a sequential method) and of Losa et al. (2006) (based on results of a weak constraint variational approach), which should further improve the balance with respect to DA methods. Here is a summary of the figures in the revised manuscript (as also given in the response to Referee #2):

1) We have moved the original Fig. (1) with the example of a variable lag fit (VLF) to the Appendix (Fig. A1) and substituted it with a figure from the study of Simon and Bertino (2012, Journal of Marine Systems, 89, 1-18; top two rows of their Figure 3). This figure is a nice example of the improved asymptotic behaviour of a deterministic Ensemble Kalman Filter (DEnKF) when using log-transformed observations and model results to realise the analysis step. Ehouarn Simon and Laurent Bertino kindly provided their results so that we could redraw the figure. The figure is referred to in Sect. (4) about Error Models. The new Fig. (1) is based on results from a sequential, ensemble based, data assimilation (DA) approach, which should further improve balance with respect to ensemble based DA methods.

2) In Sect. (7) we address space-time variations in model parameter estimates and we find it appropriate to include a plot from Losa et al. (2006; top row of their Figure 2), based on results from Losa et al. (2004). It is prominent and illustrative example of variable parameter values in the North Atlantic.

For both new figures and for Figs. (4 and 6) we still have to request permission from Elsevier.

***Comment* 2**: The authors seem to be very focused on mass conservation (is that the reason for not including many ensemble-based techniques?). I do not agree that this is the "one straw that biogeochemical modelers grasp at". In regional models, river inputs routinely break mass conservation, so why should DA techniques not be allowed to create updates to the mass inside the model domain if the data provides evidence for this? At the very least, the authors need to acknowledge that their view on mass conservation is not shared in the entire modelling community.

*Author's response*:
We did not state that DA techniques are not allowed to introduce sinks and sources of mass. But we wanted to stress that it is important to explicitly clarify whether mass is conserved or not. Scientists not directly involved in DA applications should recognise this. We confess that our formulation is a bit clumsy and we rephrased it: "So far no fundamental ecophysiological principle has been further exacted beyond the conservation of mass. Whether a balanced mass budget needs to be achieved depends on the scientific problem addressed. Some ecosystem model applications may not critically depend on mass conservation, e.g. when simulating plankton growth to act as food source in regional simulations of fish stock size and recruitment. In biogeochemical models the conservation of mass can be essential, in particular for large-scale or global ocean simulations. A consistent theme running through most ecosystem models is the determination of mass flux of certain biologically important elements, such as nitrogen, phosphorus, iron and carbon (N, P, Fe and C). "

*Comment* 3: Given that model complexity and parameter identifiability play an important role in the manuscript (and rightly so), I wonder why there is not more focus on alternatives to the functional group approach. Some approaches, like the "optimal trade-off" are mentioned in section 10.1. Yet there are others which do not require manual parameter selection, like self-selective models (Follows et al., 2007) or the gene-centric approach (Reed et al., 2014), which groups plankton groups based on genetic information. These alternatives to the functional group approach should at least be mentioned, and mentioned earlier than section 10.1.

*Author's response*:
We appreciate this helpful comment. We confess that we should have considered this aspect. However, we do not think that parameter selection is an issue only for the

functional group approach. The models of Follows et al. (2007) and Reed et al. (2014) also depend on parameters to which fixed values have to be assigned. For example, in Follows et al. (2007), fixed values were assigned to the distributional parameters describing the PAR saturation and inhibition constants (mean and standard deviation) and nutrient half saturation constants (upper and lower limit of uniform distribution) for small and large phytoplankton size classes (Follows et al., Table S1, column "Range"), and fixed values were assigned to various non-stochastic parameters (column "Fixed"). The model proposed by Reed et al. (2014) includes more than 40 parameters and they identified 16 parameters that were of importance for determining the biogeochemical dynamics in their example model setup. However, both studies are important contributions and they provide novel approaches. We extensively discussed whether we can refer to the study of Follows et al. (2007) either in Sect. (6, Model performance as a function of model complexity) or at the end of Sect. (7.1.4 Learning from space and time variation in parameter estimates). In the end we found any discussion on the Darwin model approach inappropriate in Sect. (6) or Sect. (7). We concluded that it does fit to Sect. (10.1 Modelling prospects). We therefore suggest a revision of this paragraph (Sect. 10.1) according to: " A commonality of new model formulations is to focus on principles, e.g. by considering the adaptation of traits towards optimal trade-offs (e.g., Wirtz and Pahlow, 2010; Dutkiewicz et al., 2009; Smith et al., 2015), or by accounting for allometric relationships in growth and plankton interaction (e.g., Banas, 2011; Acevedo-Trejos et al., 2015), or by using microbial traits in a functional gene approach (Reed et al., 2014). Recent studies have begun to simulate ecosystem complexity and allow the model to "self-organise" according to a relatively simple set of ecological and physiological rules or "trade-offs" (Bruggeman and Kooijman, 2007; Follows et al., 2007). A major advantage of this approach is that the models are able to resolve greater ecological diversity with fewer specified parameters whose values can be assumed to be spatially invariant. This diversity allows the simulated plankton community to reorganise across broad environmental (e.g. spatial) gradients. But the identification of the most important trade-offs governing competition between

organisms remains a major challenge (Tilman, 1990; Litchman et al., 2007, 2012)."

**Comment** **4**: More focus should be given to the role different data types play in identifying model parameters. Sometimes the manuscript seems to suggest that all is needed is more data in order to identify more parameters, for example in the abstract: "data are often too sparse to constrain all model parameters". Yet more satellite chlorophyll data is probably not helpful in identifying many parameters, other data types and subsurface data are important as well. This is not just true for large-scale models (the issue of different data types is finally discussed in section 9.1 but only in regard to global models). In this context, the authors may also want to discuss the Bio-Argo program which could provide some much needed biogeochemical data products in the near future.

**_Author's response_**:
Yes, we certainly agree. We want to introduce Sect. (10.2 Examples of recent advancements in data availability). In this short Section we want to briefly refer to different types of data and data products that could be potentially used for DA/ parameter estimation. In this paragraph we mention the Bio-Argo program and added Mignot et al., (2014) and Sauzéde et al., (2015) as references. We also mention new remote sensing products (e.g., net community respiration, Tilstone et al., 2015), new flux estimates based on time series observations (e.g. Emerson, 2014) and latest $CO_2$ data products by Roedenbeck et al., (2015) and Bakker et al., (2016).

—

**_Responses to specific comments by Referee #3_**

***Specific comment* 1**: p2 l24: "of fecal pellets" perhaps change to "attached to fecal pellets"

***Author's response***:
Aggregated cells can sink without being attached to sinking fecal pellets. Yes, both, cells and fecal pellets, are often incorporated in aggregates that sink. The proposed addition "attached to" would require further explanations and we therefore prefer to leave the sentence as it is.

***Specific comment* 2**: p3 l4: "trophic levels like fish, which would be subject to changes in biomass on multi-annual rather than seasonal time scales": I would argue that the greater challenge with modelling fish is their behaviour and ability to swim, making it impossible to realistically simulate them by tracer variables.

***Author's response***:
We changed it to: "These closure assumptions ensure mass conservation while neglecting the actual mass loss to higher trophic levels like fish, which would be subject to fish movements and changes in biomass on multi- annual rather than seasonal time scales. "

***Specific comment* 3**: p3 l5: "Every marine planktonic ecosystem model can thus be described as a simplification of the dynamics inherent to a system of nutrients, phytoplankton, zooplankton, detritus, dissolved organic matter, and bacteria". Apart from the fact that some phytoplankton are bacteria, I am wondering why they are listed here if (as stated above) they are often not resolved in models.

***Author's response***:

We actually meant heterotrophic bacteria and changed the text accordingly.

***Specific comment* 4**: p3 l8: "Feedbacks from the ecosystem model ..." Maybe mention that feedback from physical to ecosystem model are essential.

***Author's response***:
We think that the original text is fine. It is obvious from the context of ecosystem models being embedded in physical models that feedback from physics to ecosystem is essential. However, we slightly modified the text: "... but are hardly considered in current marine biogeochemical studies. With such resolved, changes in ecosystem components may induce changes in physical environmental conditions, but so far the physical model remains unaffected by ecosystem states in most studies."

***Specific comment* 5**: p4 l11: "availability of data thus places limitations on the number of model parameters whose values become identifiable." It is not just a numbers game, certain types of parameters may never be fully constrained by certain types of data, even if the model contains just a few parameters.

***Author's response***:
We understand the Referee's concern and refined the statement accordingly: "The availability (type and number) of data thus places limitations on the number of model parameters whose values become identifiable, and values of some parameters may never be fully constrained."

***Specific comment* 6**: p4: l18: "Novel DA methods are predominantly devised for improving forecasts ..." While forecasting skill is often used to judge the quality of an assimilation system, many systems are used for hindcasting and creating reanalyses.

*Author's response*:
We revised the text: "Much of the literature on DA in oceanography is focussed on state estimation (e.g., Allen et al., 2003; Natvik and Evensen, 2003; Dowd, 2007; Nerger and Gregg, 2008; van Leeuwen, 2010). In these studies, the primary objective is to improve hindcasts, nowcasts, or forecasts of time-dependent variables such as chlorophyll *a* (Chl*a*)."

*Specific comment* **7**: p10 l24: "It means that actually the cost function as given by..." this sentence is not very clear.

*Author's response*:
We corrected this formulation and refined the text (see response to Comment 1 by Referee #1): "The MAP or posterior mode estimator of $\log(\Theta)$ is equivalent here to the posterior median estimate and is obtained by maximising $p(\log(\Theta) \mid \vec{y})$. This leads to a cost function given by Eq. (15) without the second term, $2 \sum_{l=1}^{N_\Theta} \log(\Theta_l)$ (cf. , Fletcher, 2010)."

*Specific comment* **8**: p23 l5: "The third approach leads to more complex representations of growth limitation, as they..." Something may be missing here, the third approach is not described well and the "they" should be an "it".

*Author's response*:
For clarity we suggest to slightly rephrase it and add one additional sentence: "The third approach involves more complex representations of growth limitation, as it accounts for interrelations between cell quota, N-uptake and the photoacclimation state of the algae (e.g., Geider et al., 1998; Pahlow, 2005; Armstrong, 2006; Wirtz and

Pahlow, 2010). Here, photoautotrophic growth depends on the cellular C:N (or N:C) ratio and the mass distribution of phytoplankton C and N has to be explicitly resolved in the model. Whether the first, second or third approach is considered can be expected to affect estimates of the associated parameter values."

Some more information is given in the subsections that follow, e.g. in Section (3.5 Algal growth and intracellular acclimation):
" More complex interdependencies between light and nutrient limitation are resolved by models that account for intracellular acclimation dynamics (e.g., Geider et al., 1998; Pahlow, 2005; Armstrong, 2008; Wirtz and Pahlow, 2010). In these models growth rates become dependent on cell quota, e.g. usually normalised to carbon biomass (N:C), and the amount of synthesized Chl*a* per cell. These approaches involve physiological parameters that are related but not identical to those of classical N- or P-based growth models, which impedes a direct comparison of older estimates of growth parameters with values currently used in models with acclimation processes resolved."

***Specific comment 9***: p25 l3: the summary of the loss terms here "Cell lysis, excretion and leakage are usually espressed ..." does not agree very well with the summary previously (p2 l22) "... removed by natural mortality (cell lysis due to starvation, senescence, and viral attack)..."

***Author's response***:
This formulation is imprecise and we thank Referee #3 for the notice. Cell lysis is associated with cell death whereas exudation and leakage induce a loss of organic mass while the cell is alive and its physiology is fully functional. We rephrased the few sentences for clarification: "Parameterisations of phytoplankton cell losses involve lysis (starvation and/or viral infection), the aggregation of cells together with all other

suspended matter, and grazing by zooplankton. Exudation and leakage are processes of organic matter loss that occur while the physiology of the algae is functional. Cell lysis, exudation and leakage are usually expressed as a single rate parameter and this mass loss of organic matter is assumed to be proportional to the phytoplankton biomass."

**Specific comment 10**: p27 l8: "right and left sides of top and bottom row" it would be useful to have labels (a) - (f) in Figure 2.

**Author's response**:
Additional labels (1a/b, 2a/b, 3a/b, and 4a/b) were added to the figure.

**Specific comment 11**: p27 l8: "It means that $g_m$ can only be estimated in combination with". It means that estimates of $g_m$ are dependent on the values of the other parameters. If we are certain what the values of $\phi_{agg}$ and $\gamma_C$ are, we can still estimate $g_m$.

**Author's response**:
We added the following sentence for clarification: "Only if $\Phi_{agg}$ and $\gamma_C$ were known, then $g_m$ could be identified in this mesocosm model setup with these available data types."

**Specific comment 12**: p27 l9: "If $g_m$ remains fixed, we do not find such strong collinearity expressed between $\gamma_C$ and $\phi_{agg}$": I would rephrase this, since in this particular experi- ment only two parameters are varied, i.e. remove the "If $g_m$ remains fixed".

***Author's response***:
We have corrected (removed) it as proposed.

***Specific comment* 13**: p40 l29: "To account for the lateral flux information was helpful
contributed strongly to the emulator accuracy.": something is missing here.

***Author's response***:
We corrected the sentence: "Accounting for the lateral flux information was helpful,
contributing strongly to the emulator accuracy."

***Specific comment* 14**: Section 9: After discussing methodology, why are global
biogeochemical ocean models introduced now? I would move most of this section to
the introduction.

***Author's response***:
The introduction is already extensive and we do not want to further extend it. Although
not explicitly highlighted, the underlying structure of the manuscript gradually extends
from simulations of algal physiology of simulations of laboratory experiments to global
biogeochemical modelling. We have started to revise (and condense) Sect. (9) in
order to make it more concise. Some of the methodological aspects addressed before
are needed in the text of Sect. (9). It would not be meaningful to consider aspects of
parameter identification in global BGC models if they had not been explained before.

We thank Referee #3 for the support. All technical corrections are included in the
revised version of our manuscript.

---

## Author Comment (AC4) · 31 Oct 2016

We greatly appreciate the many meaningful suggestions provided by Referee #4. According to some of the Referee's comments we changed parts of the manuscript's structure, which we think has improved the readability. We thank Referee #4 for taking the time to read our manuscript and particularly for some constructive and very helpful comments.

**General comments by Referee #4**

*Comment* 1: To make the manuscript more understandable by novices in the field, the authors may want to begin the review with Section 5 "Typical parameterisations of

plankton models and their parameters" rather than starting with error models. Given that the title of this article is parameter identification in plankton modeling, it seems strange that theoretical discussion of error models comes before any discussion of actual ecosystem parameterization attempts.

***Author's response***:
This good suggestion is much appreciated. We followed Referee #4's suggestion, although it required some additional changes of the original structure of (now Sect. 3, former Sect. 5) and of (now Sect. 5, former Sect. 4). The suggested switching of the Sections works well, as it puts more emphasis on biological aspects earlier in the manuscript.

The revised manuscript structure is:
"The paper starts with some theoretical background information (Sect. 2), introducing mathematical notation and depicting prevalent assumptions that are typically made for parameter identification analyses and model calibration (Sect. 2.1). We then branch off from DA theory and discuss the parameters typically dealt with in plankton ecosystem models. In Sect. (3) we disentangle major differences between approaches to parameterising photoautotrophic growth and briefly discuss simple but common parameterisations of plankton loss rates. In this context we also address the utilisation of data from laboratory and mesocosm experiments. Error models are described in order to elucidate error assumptions made in previous ecosystem modeling studies (Sect. 4). This is followed by a description of different approaches to specify uncertainties in parameter values (Sect. 5). An example of parameter estimation with simulations of a mesocosm experiment connects aspects of Sect. (3) with the theoretical considerations of Sect. (5). Thereafter, model complexity is jointly addressed together with cross-validation in Sect. (6), followed by a review of space-time variations in marine ecosystem model parameters (Sect. 7). Emulator, or surrogate-based, approaches are briefly explained and exemplified (Sect. 8) before

we discuss parameter estimation of large-scale and global biogeochemical ocean circulation models (Sect. 9). Finally, we summarise the insights that we gained on parameter identification in Sect. (10), and we will briefly address prospects of some marine ecosystem model approaches that could improve parameter identification."

*Comment* 2: The manuscript is quite long and certain sections contain significantly more detail than others. As a result, the manuscript would be improved if the sections could be made more consistent in terms of their degree of detail.

*Author's response*:
To elaborate a meaningful structure was an extensive task and during the preparation of the manuscript we repeatedly discussed the content of respective sections of the submitted version of our manuscript. To achieve a balance that satisfies section authors as well as all other coauthors is somewhat difficult and will always be partially subjective. We think we have found some good trade-off between accessibility and discussion of a wide range of topics. It is extremely difficult to find a balance that could possibly please all readers.

According to the Referee's comment we re-assessed the content and found few places where we could skip details. We decided to skip the two paragraphs that extended on the discussion about Eigenvalue decomposition of the Hessian (now Sect. 5, former Sect. 4). Since we introduced a section that describes the basic theoretical background of the sequential DA approach (Sect. 2.2.2) we could remove the descriptions and equations in Sect. (7.1.3 Time-varying parameters). We also started to revise text and condense the content in Sect. (9) considerably, which will be finalised if the editors invite us to upload a fully revised version of our manuscript. See also response to comment 9.

**Comment 3**: In certain places, e.g. section 2, the article assumes a familiarity of terms, which may benefit from greater introductory explanation. In particular, I found the description of "kinematic model errors" and "dynamical model errors" on page 6 somewhat lacking.

**Author's response**:
We have added an additional sentence to the paragraph following the kinematic model error equation. The relevant text now reads (with extra sentence highlighted in red color): "A general relationship between the true state and model state can be expressed as:

$$\vec{x}^t = T\left[\vec{x}, \boldsymbol{\zeta}(\theta_\zeta)\right] \tag{1}$$

where $T$ is a truth operator, and $\zeta$ is a set of random variables described by distributional parameters $\theta_\zeta$. We will refer to the $\zeta$ as *kinematic* model errors because they are associated with the model state, while the *dynamical* model errors $\boldsymbol{\eta}$ in Eq. (1) act to perturb the model dynamics. The true values of the kinematic model errors therefore define the potential discrepancy between the target true state and a hypothetical "ideal" model output (i.e. with the "true" values of the parameters and, if applicable, the "true" values of the dynamical model errors).

How we interpret and specify Eq. (2) depends on the spatio-temporal averaging scales chosen to define the true state $\vec{x}^t$, which in turn depends on the objectives of the modelling study. One approach is to define these averaging scales as equal to or larger than the shortest space and time scales that are fully resolved by the model. Kinematic model errors $\zeta$ may then represent the integrated effects of the various dynamical sources of model error, if these are not already accounted for by dynamical model errors $\eta$ in Eq. (1). Alternatively, the true state can be defined over scales smaller than those resolved by the model, possibly at the scales of the observations. This may lead to a simpler model for observational error (see below), but now the $\zeta$

must account for the unresolved scales, in addition to any error effects in the model dynamics otherwise not accounted for. With stochastic dynamical models ($\eta \neq 0$), the true state is usually defined on the scales of the model and assumed to coincide with the model output for some ($\theta_e, \eta$), such that no kinematic error model is needed."

**Comment 4**: : Likewise, at the beginning of section 4.1, a few introductory remarks describing the authors' specific intended meanings of "confidence" and "credible" would be helpful when transitioning into this section.

*Author's response*:
We suggest to use the following revised text:
"Uncertainty regions in parameter space can be determined basically in two different ways, either based on a Bayesian- or frequentist interpretations. Depending on the estimator, uncertainties in the combination of parameter values may either disclose a credible region of a random distribution of parameter values (Bayesian interpretation) or they mark a confidence region that should include the true value with a certain nominal probability of e.g. 95% (frequentist interpretation). The latter means that different data sets (of same type) may yield different confidence regions and e.g. 95% of those regions are expected to include the true "fixed" value, which does not imply that the true value falls into a confidence region with 95% probability. According to the Bayesian interpretation a credible region is specified by the probability of including the true value. For maximisations of the likelihood $p(\vec{y} \mid \Theta)$ it is often stated that credible and confidence regions are practically identical. Such interpretation is imprecise since the methods to confine either regions can be very different with respect to the underlying assumptions, e.g. MCMC versus bootstrap approaches."

**Comment 5**: Section 2.2 would likely benefit from being split into a few subsections.

***Author's response***:
A good suggestion, thanks. We have split Section 2.2 into four subsections: "Basic probabilistic approaches", "Sequential methods", "Variational methods", and "Recent approaches".

***Comment 6***: Should section 4.2 be in section 4? Should section 4.3 come before 4.2?

***Author's response***: We follow the Referee's suggestion and the section about profile likelihoods has now been placed before point-wise approximations.

***Comment 7***: Can you mention one or two advantages of mesocosm data to complement your mention of drawbacks in the next sentence?

***Author's response***:
Yes, the text appears more negative than positive with respect to mesocosm experiments. We considered this and have added text, see rearranged Sect. 5 (now Sect. 3). We wrote: "One advantage is that mesocosms are, apart from the surface, closed systems and measurements of inorganic nutrients, dissolved and particulate organic matter should, in principle, add up to approximately constant concentrations of total nitrogen and total phosphorus. Total carbon concentrations may only vary due to air-sea gas exchange. By design these experiments often integrate valuable series of joint and parallel measurements, yielding detailed data from various scientist with different expertise (e.g., Williams and Egge, 1998; Riebesell et al., 2008; Guieu et al., 2014)."

***Comment 8***: Can this section include a mention of where the authors think the use of combined emulators is headed? What are the hurdles to overcome?

*Author's response*:
To Sect. (8.3 Combining dynamical and statistical approaches) we want to add more information in this respect. We suggest the addition of the following paragraph: "The ultimate aim of the two stage procedure would be to use estimates of model output that the dynamic emulator can provide rapidly for 100s of parameter vectors to construct a statistical emulator for a cost function or similar metric that is then used in parameter identification. The dynamical emulator would effectively bridge the gap between a small reference ensemble that is practical to generate with the full 3D model and the statistical emulator that requires a relatively large training set. The metric must incorporate an error model that takes into account all sources of uncertainty in the statistical emulation of the full model. Thus, the uncertainty estimates obtained when training the statistical emulator must be inflated by combining them with the dynamical emulator's own uncertainty estimates. Stage 1 emulation results suggest that it may be important to first extend the latter to include temporal covariance estimates for the parameter-dependent variation. Another important consideration is that longer spin-up times for creating the 3D model reference ensemble would be required in a practical application to truly represent the effects of varying parameters in a global circulation model. The application of dynamical emulation techniques for accelerated spin-up, such as the TM method (Khatiwala, 2007) mentioned in Sect. (8.1), could help to provide a better representation."

*Comment* **9**: After stating that an appropriate treatment of uncertainties for Earth system models is critical, it seems as if this section is going to go into depth on that subject, but it is a cursory overview. I would suggest mentioning here that a full treatment of uncertainties in Earth system models is beyond the scope of this article.

*Author's response*:

Section (9) is currently revised and will be condensed. So far, we have decided to leave the Sect. (9.4 Impact of parameter uncertainties on climate model projections into the future) as it is but added to the first paragraph the suggested statement: "An appropriate treatment of the uncertainties contained in the applied scenarios and employed models is crucial for correctly interpreting model projections, informing the societal debate about climate policies and thus strengthening the base for developing relevant measures. A full treatment of uncertainties in the projections of EMS is beyond the scope of our review and we can only address this topic here briefly."

**Comment 10**: It seems that this paragraph is not focused on what the section title states. Half of the paragraph is about a novel thermodynamically inspired ecosystem model. Can the title be reworded, or the text reworded as to relate more to "keeping number of free parameters low". Also, in the title, what does "grasping" mean in reference to complexity?

**Author's response**:
We suggest to change the clumsy title of Sect. (10.1) simply to "Modelling prospects", which catches the actual content.

We thank Referee #4 for the closer inspection. All technical errors identified and listed by the Referee were corrected.

---

## Author Response (AR2)

**Responses to referee comments of second (revised) manuscript version**

Markus Schartau et al.

*Correspondence to:* Markus Schartau (mschartau@geomar.de) and Phil Wallhead (philip.wallhead@niva.no)

***General comments by the referee***:

The authors performed a very good work in revising the manuscript. The big issue of the neglected sequential data assimilation schemes has been widely resolved. Also, the manuscript appears more balanced with regard to presenting examples of the authors' own work and those by others.

5    I still see some minor issues, which should be resolved before I can recommend the publication of the manuscript. Unfortunately, the manuscript still shows a clear preference against sequential data assimilation methods. The major disadvantage of the manuscript seems to lie in the fact that the group of authors does not include any scientist who applies sequential data assimilation. To this end, the apparent goal of being a fully comprehensive review and synthesis is not reached. This, however should not prevent the publication, because the manuscript is otherwise a very comprehensive, but also extremely long, review.

*Author's response*: We very much appreciate the time and effort that the referee has put into the evaluation of our revised manuscript. We regret that she/he has still the impression of a bias against sequential methods, but we do feel that the manuscript has achieved a level of balance that is reasonable for a review paper. Each of the referee's comments below have been addressed and we have made modifications where considered necessary.

***Main comments***

***Comment* 1**:

Abstract, last sentence: As I already commented in my first review, the last sentence of the abstract - the recommendation to
20   balance level of sophistication of the model and the data assimilation treatment - is not a clear result of the study. The authors only responded that from literature review they had the 'general impression' and hence 'felt' that it would be helpful to report their impression to the readers. For me, this is pretty surprising response, because, as the statement itself, the response is not clearly based on scientific insight but rather represents an opinion. On the other hand, Section 6.2 does in fact discuss the model complexity in a scientific way. Thus, I can still only recommend to reformulate the last sentence in a way that it is not just a
25   statement of opinion (which is not suitable for an abstract of a scientific paper), but shows that it is a result of literature review and studies on model complexity.

*Author's response*: Perhaps we were too careful (in our response) in expressing what we have learned from reading a large number of papers and the phrase "this was a general impression" does not actually reflect the insight we gained. We apologise for being vague in our response text.

The last sentence in the abstract includes a recommendation that is hardly controversial. The sentence follows after stating: "Our review discloses many facets of parameter identification, as we found many commonalities between the objectives of different approaches, but scientific insight differed between studies." In this context (logical sequence) a reader will understand that this recommendation is an overall outcome of our extensive literature review and not just an opinion. It is partially backed up with citations in Sect. (6.2). It is confirmed by our finding that latest studies that involve highly sophisticated data assimilation methods for parameter estimation (also along with state estimation) do not comply with latest developments in modeling phytoplankton growth and plankton interaction, e.g. as described in Sect. (3). Conversely, studies that focused on improving mathematical representations of plankton dynamics have often neglected aspects of parameter identification.

The above described development within the scientific community is evident and can be tested by screening through the literature we refer to in our review. The problem is addressed and discussed in our last paragraph (Sect. 10.3). In conclusion, inference with respect to model development and parameter identification could be further improved if knowledge and expertise of the different scientific communities would be better merged (i.e. communicated). It is for these above reasons that we do not want to follow the referee's request to rephrase the last sentence of the abstract.

*Comment* 2:

In the new section 2.2.2 on sequential methods, the authors state that sequential data assimilation approached can be expedient in cases that assimilating all data at different times is 'computationally impractical'. This statement is misleading, because sequential schemes can be used (and actually have been used) also in cases where a variational scheme could still be applied (There are various cases in the literature). Even more, a sequential scheme can also have a 4D-component, when a smoother method is applied. Rather, it depends on the scientific question whether a sequential data assimilation method results in the required result and the manuscript itself provides examples for these cases. In this respect also the statement that sequential data assimilation method '...break the large integration problem ... into a number of smaller problems...' (page 6, lines 69-70) is incorrect. The sequential methods reformulate the data assimilation problem into a sequential form. This, however, does not necessarily 'break' it. Furthermore, the sequential methods are not necessarily a 'sequential approximation' (page 6, line 74). Please revise the first paragraph of Section 2.2.2 accordingly.

*Author's response*: Section (2.2.2) begins with: "In some problems, assimilating all the data at once from all available sampling times can be computationally impractical". This is a natural introductory sentence given the previous section, which introduces the posterior density (Eq. 6) and the likelihood (Eq. 7) as the basis of theoretically optimal estimates. If we had an "ideal" computational algorithm we would simply apply it to evaluate the posterior density or the likelihood over the space of model parameters.

As we see it, from a theoretical viewpoint (appropriate to a Theoretical Background section), the best motivation for sequential methods is to make computationally practical approximations of the likelihood or posterior for cases involving large and/or stochastic models. When we consider a stochastic model, Eq. (7) becomes a very high dimensional integral over the dynamical model state at all grid points and all times. But even if the integral in Eq. (7) is feasible (e.g. for a deterministic model) it may

5     still be desirable to avoid having to redo it every time new data come in, or to avoid having to rerun over the entire simulation period for each new trial parameter value. So we go on to say: "This is particularly likely for models with stochastic dynamics ($\eta \neq 0$ in Eq. 1), if the data are clustered in time, or if model states need to be repeatedly updated as new data come in. In such cases a sequential approach can be expedient. "

The wording is concise, but it is carefully chosen. Nowhere in the manuscript do we state or imply that sequential methods are

10     a kind of "second choice" option for when variational methods become computationally impractical. That is not our opinion, and we have been careful to avoid such implications in the manuscript. Indeed, we do not consider "variational methods" to be the sole alternative to sequential methods (for example, MCMC sampling of the posterior probability is neither a sequential nor a variational method). We define "variational DA" in a broad sense in Sect. (2.2.3) ( "Within the marine ecosystem modelling community, the term "variational DA" is often used more broadly to refer to all non-sequential methods that involve the min-

15     imisation of a cost function, whether or not this is based on a probability model" ). Our theoretical ideal (probabilistic) method requires accurate evaluation of the posterior probability (Eq. 6) and/or the likelihood (Eq. 7). Minimization of a cost function can in general provide only a posterior mode estimate, and in practice only an approximation of this. Therefore, variational DA is not a theoretically ideal method. To be clear, neither sequential nor variational methods are ideal, nor do they cover all approaches. They both involve approximations that may be more or less appropriate depending on the particular problem. We

20     do not imply that sequential and variational methods cannot both be applied to the same problem (indeed, this may be a useful approach to ensure robust scientific conclusions).

It is true that sequential methods can have a 4D component if a sequential smoother algorithm is applied. However, a sequential smoother is still functioning as a computational aid, breaking the large integration/sampling problem defined by Eqs. (6) and (7) into a set of smaller, more feasible integration/sampling problems by subsetting observations with respect

25     to sampling time (e.g. see Evensen and van Leeuwen, 2000, section 2a). If we could accurately and efficiently evaluate Eqs. (6) and (7) by some other means, we would not need a sequential smoother algorithm. And if a smoother algorithm is not decomposing the problem by subsetting with respect to sampling time, then it would not be a "sequential" smoother. Our next sentence reads: "The basic idea is to break the large integration problem defined by Eq. (7) into a number of smaller problems by sequentially assimilating observations in subsets defined by sampling time."

30     The reviewer seems to give a negative connotation to the word "break" here, which was certainly not our intention. For us, the strategy of breaking a large, intractable problem into a set of smaller, tractable smaller problems is surely an intelligent one with positive connotations. The large integral in Eq. (7) is broken into smaller pieces in the sense that $x$ in Eq. (7) has dimension $N_g \times N_s \times N_t$ where $N_g$ is the number of spatial grid points, $N_s$, is the number of model state variables, and $N_t$ is the number of time steps (between each of which a stochastic perturbation may be applied). The dimension of $x_j^t$ in Eq. (8) is

35     only $N_g \times N_s$. So the problem is far from solved, but it has certainly been broken into smaller pieces. We avoid saying that the

problem is "reformulated", because for us this word may imply some fundamental change in the statistical inference problem, either in the error model or the dynamical (process) model, or in the estimation method.

In our view it is essential that readers appreciate the fundamental statistical basis of data assimilation algorithms (which is not necessarily their historical development). We do this by linking the sequential forecast/analysis equations (Eqs. 8 and 9) and the application of Bayes' Theorem to the model parameters (Eq. 6) (cf. Evensen and van Leeuwen, 2000, sections 2a,b, also see Evensen, 2009, and Jazwinksi, 2007). We want readers to understand that by adopting a sequential method they do not necessarily need to change their statistical model or estimation method (e.g. posterior mode vs. posterior mean, Bayesian vs. maximum likelihood). In our view, the models (dynamical and statistical), and estimation methods should be selected to suit the scientific problem, then the data assimilation algorithm should be selected to suit the models and estimation methods. We should not in general choose a model/estimation method to suit an algorithm, just as we should not choose a problem to suit a model/estimation method (of course there are exceptions and compromise is necessary in the real world).

Regarding the final sentence: "If the sequential approximation or 'filter' is accurate, it should approximate the posterior distribution defined by Eqs. (6 and 7), when all data have been assimilated by the end of the assimilation period."
Here we have modified it to: "If the sequential algorithm is accurate, it should approximate the posterior parameter distribution defined by Eqs. (6 and 7) at times where all available data have been assimilated." This allows us to avoid specialising to the case of a filter, which as the reviewer noted is not appropriate for an introductory paragraph.

*Comment* **3**:
Page 7, lines 49-53: Here the (Extended) Kalman filter is discussed. Actually, these filters are not relevant for large-scale systems like in ocean biogeochemical modeling, because of model nonlinearities and because of the high-dimension of the models, which makes it impossible to store the full state error covariance matrix. This is know to most researchers for more than a decade and accordingly studies use variants of the ensemble Kalman filter, which can cope with high-dimension and partly with the nonlinearities. I recommend to remove the lines discussing the (extended) Kalman filter.

*Author's response*: We stress that this is not a review of data assimilation techniques applicable to large-scale systems. It is a review of parameter identification methods for marine ecosystem models of all scales, from laboratory through mesocosm to regional and global scale models, including box models and 1D vertical models of station time series. We think it is appropriate to keep these two sentences because: i) they help orient a reader in the literature and complete the overview by linking back to the historical origins of sequential methods, ii) given that we are not concerned only with large-scale models, we should not assume that the covariance matrix will exceed storage capacities, and iii) it is not clear that ensemble algorithms are the only reasonable approaches to limit storage requirements. There may be problems where a non-ensemble Kalman filter may be applicable to parameter estimation, perhaps after applying suitable transformations to deal with nonlinearity (e.g. using a SEEK approach to reducing the rank of the covariance matrix, see Nerger et al., 2005).

*Comment* **4**:

Page 7, lines 74-81: The text mentions that there are 'powerful mathematical tools' for variational DA and that the adjoint methods are 'extremely efficient'. Here, the authors should be more specific as the statements are too superficial. What mathematical tools are meant; why are they 'powerful'? Further, what is 'extremely efficient' and compared to which methods is this the case? Actually, given the importance of the adjoint method, I would recommend that the authors state here in the main text, what the adjoint method actually is.

*Author's response*: All relevant information is given in the text already (Sect. 5.3.3 and Appendix C,). Section (2.2.3) informs about the advantages and drawbacks. However, we slightly rearranged and modified the second paragraph, so that it should become clearer: "In any case, there are some powerful mathematical tools developed for variational DA that can be applied to minimise cost functions. Adjoint methods allow the gradient of the cost function with respect to all fitted parameters to be computed in an extremely efficient manner, see Lawson et al. (1995), and Appendix (C). This is particularly useful when dealing with a large number of fitted parameters (high-dimensional $\Theta$) of a computationally expensive model (e.g. Tjiputra et al., 2007). The application of the adjoint method helps reducing the number of model runs to provide access to joint posterior mode and maximum likelihood estimates."

*Comment* **5**:

Sections 2.3 and 2.4: These sections actually describe further aspects of the theoretical data assimilation background. I recommend to change them into sub-subsections of Section 2.2. Section 2.3 could also be merged with Section 2.2.3, because the cost function discussed here is only relevant for variational methods, which are discussed in Section 2.2.3. When Section 2.4 is changed in to sub-sub-section of 2.2 also its first sentence 'We close this section' would be reasonable since 2.2 is concerned with DA methods.

*Author's response*: The detailed structure was refined many times while being profoundly discussed among the lead authors of Sect. (2). On balance we realise that the present structure is entirely consistent:

Sect. (2.1) reviews **Statistical model formulation**

Sect. (2.2) reviews **Estimation Methods**

Sect. (2.3) is a simple **worked example**

Sect. (2.4) considers **terminology** (and closes Sect. 2 **Theoretical Background**)

Section (2.3) does not actually provide further methodology, but rather provides a worked example. The cost functions in Sect. (2.3) are not quite only relevant to variational methods; they are also relevant to other non-sequential methods such as MCMC sampling.

*Comment* **6**:

Last lines of Section 3.2 and Section 3.5: Section 3.5 seems to repeat some aspects on model formulations that account for acclimation dynamics which are already mentioned in Section 3.2. I recommend to focus 3.2 clearly on the aspect of limitation to avoid the redundancy.

*Author's response*: We thank the referee for spotting this. To avoid a repetition we now focus in Sect. (3.2) on the light and nutrient limitation aspect and modified Sect. (3.5) accordingly, with focus on the dependency between growth and cellular acclimation. The respective part in Sect. (3.2) now reads: "... .The third approach involves combinations of light- and nutrient limitation that resolve interrelations between cell quota, N-uptake and the photoacclimation state of the algae (e.g., Armstrong, 2006, see Sect. 3.5). Whether the first, second or third approach is considered can be expected to affect estimates of the associated parameter values."

And the modified Sect. (3.5): "More complex growth dependencies are described with models that consider intracellular acclimation dynamics (e.g., Geider et al., 1998; Pahlow, 2005; Armstrong, 2008; Wirtz and Pahlow, 2010). In these models, photoautotrophic growth rates become dependent on cell quota, e.g. usually normalised to carbon biomass (N:C), and the amount of synthesised Chl*a* per cell. With such approaches, the changes of the mass distribution of phytoplankton C and N, as well as the cellular Chl*a* content, have to be explicitly resolved in the model. One advantage is that these models are more sensitive to variations in light conditions and nutrient availability. The respective equations involve physiological parameters that are related but not identical to those of classical N- or P-based growth models, which impedes a direct comparison of older estimates of growth parameters with values currently used in models with acclimation processes resolved."

*Comment* **7**:

Page 12, line 96: The texts cites '(Simon and Bertino, 2012, Fig. 1)'. I can only guess that 'Fig. 1' does actually refer to Figure 1 of the manuscript and not Fig. 1 of Simon and Bertino (2012). This guess is based on the fact that I didn't find any other place where the manuscript refers to Fig. 1. Actually, in the current form, the figure is nothing more than a pure illustration because the only connection with the main text is that it refers to the figure. As for the other figures, please describe in the text what is the particular result shown in Fig. 1.

*Author's response*: To avoid ambiguity we have modified the sentence to: "It is yet unclear whether such extra flexibility is generally necessary, but it has been demonstrated that the choice of transformation can strongly affect estimates of plankton ecosystem fluxes (Evans, 2003) and that a good choice can improve parameter estimation in twin experiments (see Fig. 1 and Simon and Bertino, 2012)."

We acknowledge that there is not much discussion of Fig. (1), but it is not a pure illustration because it is clearly cited as an example of how a good choice of parameter transformation can improve parameter estimation in twin experiments. The particular result should be already clear from the above sentence. Note that other figures from variational studies (e.g. Figs. 3

and 4) receive a similarly terse treatment in the main text, as is necessary to limit length given the breadth of the manuscript's scope (treating studies at all scales).

*Comment* **8**:

Page 13, lines 14-16: It is stated 'Neglected correlation may result in parameter estimates that are less efficient... and more strongly correlated'. Please provide a reference for this statement.

*Author's response*: The sentence has been modified to: "Neglected correlation may result in parameter estimates that are less efficient (higher variance) and more strongly correlated (e.g. see example in Sect. 5.4)."
Note that such results may be well anticipated from classical results of linear regression analysis (where the generalised least squares estimator is more efficient than the ordinary least squares estimator when the true errors are correlated).

*Comment* **9**:

Page 13, lines 67-68: It is stated 'To our knowledge no application has yet incorporated prior correlations between parame-ters'. This statement does actually ignore that ensemble-based sequential DA schemes naturally include correlations between parameters, if the model dynamics yield them.

*Author's response*: Both sequential and non-sequential approaches can yield correlations between posterior parameter val-ues (Bayesian paradigm) and between parameter estimates (frequentist paradigm). The statement clearly concerns the **prior** correlations between parameters. An ensemble of stochastic model simulations will certainly generate correlations between **state variables** (whether or not the parameters are estimated by sequential or non-sequential methods). But we are not aware of any study that has assumed **a priori** correlations between **parameters**.

*Comment* **10**:

page 16, lines 17-19: 'The determination of parameter uncertainties has many facets, getting to the core of discussions of Bayesian and frequentists approaches...'. This statement cannot be understood unless the reader already knows what it actually means. One needs to read the full section 5 to get the idea which facets are meant and what the Bayesian and frequentist approaches are. Please revise the text so that readers don't need to speculate what the authors actually mean to say.

*Author's response*: We understand the referee's concern and moved sentences from the first paragraph of Sect. (5.1) to this introduction part. The first paragraph of Sect. (5) now reads: "The determination of parameter uncertainties has many facets, getting to the core of discussions between Bayesian and frequentist approaches and interpretations (e.g., Efron, 1986; Cox, 2005; Lele and Dennis, 2009). Depending on the estimator, uncertainties in the combination of parameter values may either disclose a credible region of a random distribution of parameter values (Bayesian interpretation) or they mark a confidence region that should include the true value with a certain nominal probability of e.g. 95% (frequentist interpretation). The latter

means that different data sets would yield different confidence regions and e.g. 95% of those regions are expected to include the true "fixed" value."

***Comment* 11**:

Section 5 in general: Unfortunately, the authors missed to include uncertainty estimates from ensembles methods. Please also discuss it for completeness.

*Author's response*: This is a valuable comment. How the posterior distribution, in principle, can be resolved has already been mentioned in Sect. (2.2.2). However, we agree with the referee and think that it should be picked up again here. In this respect we also find the work of Weir et al. (2013) to provide good information with examples of a twin experiment. We have modified the paragraph: "In the case of classical BEs no tolerance limit $\Delta_J$ is explicitly prescribed. Instead, some efficient sampling of $(\Theta, J(\Theta))$, or directly of the posterior $p(\Theta \mid y)$, is applied. Sequential methods provide approximations of the posterior parameter distribution once all data have been assimilated. These approximations differ, depending on how Eqs. (6) and (7) are sampled and evaluated, as discussed in Sect. (2.2.2). A helpful overview with some comprehensible examples (of four different methods and three different ensemble sizes) is given by Weir et al. (2013). BE methods that do not rely on sequential approaches may also be applied and credible regions are then simply inferred from selective (acceptance/rejection) sampling schemes in a MCMC approach, e.g. Metropolis-Hastings algorithm (Metropolis et al. (1953; Hastings 1970). MCMC methods for the derivation of credible regions are also used for ML estimation problems (e.g., Smith and Yamanaka, 2007a). The main point is that here the data are assumed fixed.

A fundamentally different approach to the BE methods is to repeat parameter optimisations many times but with data sub-samples or resample data sets. Large data sets are split up into... "

***Comment* 12a**:

Section 6.2: The section contains the statements: 'the appropriate degree of model complexity in any given situation is both one of the most important, and one of the least well defined' (p 20, lines 49-50) 'there exists a fundamental trade-off between simplicity and complexity' (p20, l54-55) 'the extra degrees of freedom can lead to the introduction of compensatory errors at the assimilation site' (p20, l67-68) 'an extra flexibility may lead to very different model solutions with only small variations in the assimilated data' (p20, l69-71) All these claims appear to be results from scientific studies rather than the authors' opinion. Accordingly supporting references are required to make the claims valid for a scientific paper.

*Author's response*: As a matter of style, we feel that excessive citation can in some cases detract from readability without providing any real additioinal help to the reader. Considering each of the instances in question:

1) "Of the many factors that affect the ability of a biogeochemical model to reproduce and predict observations, the appropriate degree of model complexity in any given situation is both one of the most important, and one of the least well defined."

This is a general and uncontroversial statement which serves to introduce Sect. (6.2). It does not require a supporting reference, in our view. It is followed by:

**2)** "This is because there exists a fundamental trade-off between simplicity and complexity."

Again, a general and uncontroversial statement, which is immediately explained in the two following sentences: "Simple models have the advantage of being easier to understand, and with fewer parameters they should also be better constrained (both before and after optimisation). Nonetheless, simplification requires a degree of abstraction, and it can sometimes be difficult to draw parallels with the complexities of the observed system."

Regarding the next two

**3)** "If insufficient observations are available, the extra degrees of freedom can lead to the introduction of compensatory errors at the assimilation site, which could then increase uncertainty at other locations."

**4)** "Similarly, an extra flexibility may lead to very different model solutions with only small variations in the assimilated data, also leading to increased uncertainty in model predictions."

Here we agree that references will be helpful; we have modified to: "If insufficient observations are available, the extra degrees of freedom can lead to the introduction of compensatory errors at the assimilation site, which could then increase uncertainty at other locations, as illustrated by Xiao and Friedrichs (2014b). Similarly, for small changes in the assimilated data an extra flexibility may lead to very different model solutions, also leading to increased uncertainty in model predictions (e.g. Xiao and Friedrichs, 2014a)."

**Comment 12b**:
page 20, line 75-76: What makes the review by Johnson and Omland (2004) 'useful'?

*Author's response*: It is our opinion that the review is "useful". The word was included because it helps to communicate our thinking to the reader.

**Comment 12c**:
Why is cross-validation 'most practical' and perhaps 'most general'? The particular expressions are opinions of the authors. It would be preferable for a scientific paper if the authors focus on facts.

*Author's response*: The full sentence reads: "One of the most practical (if not the most general) techniques is cross-validation, as described in the previous section." This is not such a strong statement (because we say "One of..."). It also directs the reader to the previous section where we give a description: "This is the principle of cross-validation, in which an optimised

model is tested in terms of its ability to reproduce data that were not included in the calibration phase. This is achieved by excluding a subset of the original calibration dataset, for later use in model evaluation." In our view, the fact that the method can be described in plain language in two sentences itself supports the statement that this is a practical and general method. In fact the practicality and generality of cross-validation is not controversial. Hastie et al. (2009) begin their section on 7.10 Cross-Validation with the sentence: "Probably the simplest and most widely used method for estimating prediction error is cross-validation." Given that this important text is not cited elsewhere and provides an excellent discussion of cross-validation (in our opinion) we have decided to cite it here. Our sentence is modified to: "One of the most practical and general techniques is cross-validation, as described in the previous section (see also Hastie et al., 2009, section (7.10) for an excellent discussion in a general statistical context). By looking at ..."

*Comment* **13**:

page 21, line 1: I recommend to start a new paragraph at 'A perhaps more intuitive....'

*Author's response*: This is a good suggestion. We changed it accordingly.

*Comment* **14**:

page 21, lines 68-69: It is stated 'Among models with a similar score, the simplest should be favoured'. Here, I again recommend to rephrase the statement to be scientific, which excludes 'favours'. Scientifically, the optimal model choice seems to be that one with the least parameters and minimum score or score within a certain threshold from the minimum.

*Author's response*: Actually this sentence is not correct, and begs the question of a threshold for defining "similar". The simplest standard approach is to choose the model with the lowest AIC (or AICc or BIC). However, this is not a robust approach when we have multiple models that achieve "similar" AIC scores; a better approach in general is to weight the models according to their AIC and perform multimodel inference (Burnham and Anderson, 2002). We have replaced this sentence and the preceding one with the following: "The AIC and BIC can be used to select a single model with the lowest score, or preferably to provide individual model weightings for multimodel inference (Burnham and Anderson, 2002), although it appears that this latter has so far seen little application to planktonic ecosystem models."

*Comment* **15**:

Section 7.1.4: Given the fact that Section 7.1.3 already discussed studies considering time- or space-varying parameters, the first sentence of the section reads quite odd. Please rephrase it.

*Author's response*: Section (7.1.4) (now Sect. 7.5, following a correction to the section numbering) is intended to be a concluding discussion for Sect. (7) as a whole, with the first sentence acting as a summary. This was not clear, so the sentence which reads "A variety of approaches have been explored for DA with parameters varying in space or time or both."

is now replaced with

"As shown in this section, a variety of approaches have been explored for DA with parameters varying in space or time or both. We conclude the section by considering what might be learnt from these types of studies."

*Comment* **16**:

Section 8: This sections appears to be overly detailed compared to the treatment of others aspects in the manuscript. At the same time it is too short to really understand details. E.g. from reading the text, I could not really understand the meaning of he 'alignment operator' (pages 25/26). How can the emulator equal the model (page 26, line 2)? this seems to ignore the presence of the alignment operator matrices $A_l i$. However, given the small number of references in Section 8, its length is not consistent with the current relevance of the methods. Thus, I recommend to shorten the section to a concise overview of the dynamic and statistical emulators. One clear possibility for shortening is also the example in Section 8.2 (page 26, right column), which is too detailed and incomplete at once. While for the model itself it is referred to a publication, the manuscript lists explicit parameter values, which is of no use without knowing the model equations. Further, the example is only concerned with a 0D case, which appears to be trivial in particular as the authors intent to discuss emulators in the context of high-dimensional models. The application in 3D cases of higher dimension, where the method could be most useful, appears to be extremely difficult.

*Author's response*: Emulator approaches are a relatively new development and thus the number of available publications is still limited. It seems, however, that emulator approaches will gain more and more importance in the future - particularly as model complexity increases due to increased computer power, which hinders systematic data assimilation with full models. We thus believe that this upcoming development needs a detailed elaboration. We again went critically through the section and decided to add few sentences to Sect. (8.1) for further clarification of the alignment operator. Along with this, we reformulated page 26 line 2 and we agree that the original formulation was misleading. We also found places where we could shorten text in Sect. (8.2).

Although simple, the example of the statistical emulator is not at all trivial. The example is very illustrative, because it gives an idea of the similarity between the contours (here of RMS error) of the model and the emulators, for a given training set. Whether a 0D- or a (computationally expensive) complex 3D model is applied is irrelevant in this context. We agree that the level of detail provided in the example description may not be appropriate. We therefore decided to shorten the description of the example considerably, and we have moved relevant details to the caption of Fig. (7).

In Sect. (8.3) we revised text, mainly to come up with a slightly shortened version.

*Comment* **17**:

Section 9: The authors stated in their response that they shortened this section. However, this doesn't seem true as in the original version the Section spread over about 4.5 pages and now (in the document version 'author response version 1', it's again 4.5 pages. Particularly long is subsection 9.3. While its title suggests that the section discusses 'Parameters relevant for global ocean BGC modelling', the section does almost exclusively discuss the parameter 'b' of the power law for particle flux. Here also an example from Kriest et al. (2016) is included, which is described in quite some detail. This again seems to be too detailed, even more as the conclusion appears to be that the value of 'b' is 'well identifiable' (page 39, line 76) (similarly in line 62 for a different case). To this end, the main result appears to be that the value of 'b' can be determined but is specific for each model. I wonder, why so much space is used to describe this result.

*Author's response*: Following the attempts of our first response we removed original paragraphs, then restructured and rewrote paragraphs with less detail. However, we then learned that we should consider examples with sequential approaches as well (Ridgwell et al., 2007 and Simon et al., 2015). Eventually, we realised that we had forgotten to refer to Tjiputra et al. (2007), a study where an adjoint model had been applied. These new paragraphs (with new information included) turned out to compensate for the reduction of text elsewhere in Sect. (9).

We screened the individual sections and removed sentences or phrases where possible. But in general, we think that the level of detail is appropriate for Sects. (9.1), (9.2), and (9.4). Sect. (9.3) is of central importance, since therein we address those parameters that are relevant for determining dissolved inorganic tracer distributions on long time scales and large spatial scales (sinking velocity of detritus and remineralisation rate) and we discuss how they are linked to parameterisations of particle export flux (e.g. when described with power law of depth). Note that the focus is on dissolved inorganic tracer distributions for model calibration, which is justified by the data availability discussed in Sect. (9.2). Further, this section, and its emphasis on "b" is motivated by the importance of this parameter for atmospheric $pCO_2$ (Kwon et al., 2009; see first paragraph of this subsection). This parameter has therefore received much attention in the global biogeochemical modelling community. Because it has been popular in global models, we think it is justified to have a closer look at its implicit assumptions. The fourth paragraph of Sect. (9.3) describes an example of how climatological data impose different constraints on parameters that act on the surface and those of particle export, and thus provides a link to Sect. (5) of the paper; this time from a global perspective.

To summarise, in the first paragraph of Sect. (9.3) it is clarified that not all ecosystem model parameters that are important in a local or mesocosm context, or on short time scales, are also relevant for tracer distribution in the global ocean. We address computational constraints, and implications for parameter identification in Sect. (9.1), data constraints (Sect. 9.2), and state-of-the-art of identifying the biogeochemical parameters in global models, run on long time scales (Sect. 9.3). Section (9.4) finally closes this review by summarising the relevance of model sensitivities for future projections. As such, we think the Sect. (9) is comprehensive and concise.

*Comment* **18**:

Page 31, line 17: Please add a reference to the assessment report of the IPCC.

*Author's response*: We added a reference.

*Comment* **19**:

Page 31, lines 94-95: It is stated: 'In BGC models the conservation of mass can be essential, in particular for large-scale or global ocean simulation'. Actually, the conservation requirement is not resulting from a model being large-scale or global, but it results from the scientific question to be considered. In the current form, the text implies that all data assimilation applications of sequential methods with large-scale or global models are wrong because they don't conserve the mass. This is certainly not true and the authors contradict their own statements in Section 2.2. Please reformulate the statement.

*Author's response*: The sentence in question concludes a sequence of three sentences: "Whether the DA approach conserves mass and/or energy is relevant in this respect, depending on the scientific problem addressed. Some ecosystem model applications may not critically depend on mass conservation, e.g. when simulating plankton growth to act as food source in regional simulations of fish stock size and recruitment. In BGC models the conservation of mass can be essential, in particular for large-scale or global ocean simulations."

No mention is made here of sequential methods, and nowhere in the present manuscript do we state that sequential methods necessarily violate mass conservation. We only state in Sect. (1.4) that stochastic models (which do not necessarily imply sequential methods) may violate mass conservation, and then in Sect. (4.5) we discuss how stochastic noise may be injected without violating mass conservation. Further, the second sentence in the above explicitly acknowledges that there may be problems where mass conservation may not be essential, and the final sentence only states that mass conservation can be essential. However we agree with the reviewer regarding the model vs. scientific question, and have therefore modified the final sentence to read: "In BGC applications the conservation of mass can be essential, in particular for large-scale or global ocean applications."

*Comment* **20**:

Page 32, right column, lines 2-3: I already recommended in my first review to mentioned here what the dynamics and statistical emulators are. Unfortunately, the authors just replied that they don't see a need for this, because they defined these emulators in Sections 8.1 and 8.2. To this end, I like to remind the authors about the fact that this is the summary section. Usually, one doesn't expect that readers will read the whole paper but many readers will focus on the introduction and summary (which are already quite long in this manuscript). Thus, it would just help readers if the authors would add one or two sentences shortly mentioning that a dynamical emulator is a computationally cheap approximation of the model operator, while a statistical emulator simulates the output from inputs in a statistical way based on a prior training with independent input/output sets. This should be possible in a very short way so that the overall length of the manuscript is not significantly changed.

*Author's response*: We understand and follow the referee's suggestion. We like the referee's phrase but are not sure whether

we could/should adopt it directly. However, we added three sentences: "The dynamical emulator is a simpler representation of a full model operator that is computationally expensive, thereby approximating the underlying model dynamics. A statistical emulator interpolates model output from a set of training runs with different values assigned to the parameter vector. Based on the derived statistics it can be applied to approximate unknown model output for other input parameters."

***Comment* 21**:

Fig. 6: It's written 'Geographic extent of the two sub-domains'. I can only guess that the colors in the plot distinguish the two sub-domains. Unfortunately, this is never described.

10 *Author's response*: The beginning of the caption is now clarified to read: "Geographic extent of the two sub-domains giving the optimal calibration in the split-domain calibration study of Hemmings et al. (2004), shown here in yellow and green. Also shown are the distributions of the sites used from the calibration set to obtain the parameter vectors for each sub-domain and the sites used for cross-validation. Biogeochemical provinces defined by Longhurst ...".

15 ***Comment* 22**: Typos

*Author's response*: We thank the referee for spotting those typos. We corrected all six of them.

5    eter optimisation. However, because of local minima or flat regions in the cost function one associated problem is that optimal estimates may then depend on the initial guess of parameter values, as discussed in Sect. (2.2.3).

[revised manuscript text omitted]

**9.3 Parameters relevant for global ocean biogeochemical modelling**

~~Parameters of phytoplankton growth and of organic matter remineralisation determine simulated primary production rates. But model calibration against primary production data can be ambiguous, because simulated rates can be tuned to some desired magnitude, by adjusting those parameters that regulate nutrient turnover within the upper ocean layers (e.g. algal exudation rate or assimilation efficiency of zooplankton grazing) without affecting organic matter flux to the ocean interior (Oschlies, 2001). Thus, simulated tracer concentrations at great depth (> 1000 m) may not critically depend on parameters that specify seasonal variations in primary production. The large scale distributions of dissolved nutrients and of oxygen are sensitive to changes in~~

~~ocean circulation dynamics. However, deep global tracer concentrations are also sensitive to biogeochemical parameters that describe global particle export at depth (e.g. between 500 and 2000 m). This sensitivity suggests that global nutrient and oxygen data may help to identify credible values for parameters of particle flux and remineralisation (Kwon and Primeau, 2006; Kriest and Oschlies, 2013). In this manner the model s ability to adequately simulate organic matter flux and carbon storage in the ocean can be improved, e.g. for refining future projections.~~

The joint effect of particle flux and remineralisation is often described by one or two parameters in global models. Early models referred to an exponential function of remineralisation with depth (Bacastow and Maier-Reimer, 1991), which - in equilibrium - would correspond to a constant particle sinking velocity and constant remineralisation. Another, common description of particle flux (and hence of subsequent remineralisation) is the consideration of a power law of depth: $F(z) \, z^{-b}$, where $b$ is usually set to $b = 0.858$, representing the open-ocean composite value derived by Martin et al. (1987) from sediment traps (e.g., Maier-Reimer, 1993). Empirical fits to various observations of particle flux suggest that $b$ may vary between 0.3-1.4 (Martin et al., 1987; Berelson, 2001; Van Mooy et al., 2002; Buesseler et al., 2007). This typical range of variation of $b$ has been used and tested in global biogeochemical models e.g. analysing how its value affects dissolved tracer concentrations in the ocean (Kwon and Primeau, 2006, 2008; Kriest and Oschlies, 2013). Kwon et al. (2009) coupled a simple global biogeochemical model with a one-box atmosphere and found a large effect of this parameter on atmospheric pCO$_2$, highlighting the relevance of this parameterisation in ESM simulations. Since this parameterisation is widely used (e.g., Kwon and Primeau, 2006, 2008; Najjar et al., 2007; Parekh et al., 2005) we will have a closer look at its implicit assumptions in the following and discuss potential constraints for the estimation of respective parameters.

Under steady state conditions $b$ can be interpreted as being equal to a constant remineralisation rate $r$ divided by a particle sinking speed $a$ that increases with depth: $b = r/a$ (Kriest and Oschlies, 2008). The associated potential mechanisms that may lead to a vertical increase in sinking speed are selective export of large and fast particles to deeper layers, or repackaging of small particles into larger ones by zooplankton egestion. An alternative interpretation is to assume the sinking speed to be constant while the remineralisation rate decreases with depth. This implies that particles may become more refractory and less susceptible to bacterial degradation, or that bacterial activity is reduced by the decrease in temperature at depth. Other parameterisations of particle flux profiles have been applied in global models, e.g., constant sinking and remineralisation (leading to an exponential flux curve; e.g., Bacastow and Maier-Reimer, 1991), or models that explicitly simulate different groups of particles with different size and properties (e.g., Gehlen et al., 2006; Schwinger et al., 2016). Cabre et al. (2015) provide an excellent overview about different parameterisations for models applied in CMIP5.

So far few attempts have been made to systematically calibrate parameterisations of particle export and remineralisation in global biogeochemical models. Kwon and Primeau (2006) assimilated annual mean phosphate data into a simple global ocean biogeochemical circulation model to optimise globally uniform $b$. Their study shows that the value of of $b \approx 1$ can be well identified for their model when using global climatological data. According to their approach, the tracer distributions are dynamically consistent with their solution of ocean circulation.

uncertainties in the circulation-driven transport and in observations. Furthermore, it may also be associated with 
[revised manuscript text omitted]